# Chromosome-level baobab genome illuminates its evolutionary trajectory and environmental adaptation

Justine K. Kitony [1], Kelly Colt [1], Bradley W. Abramson[1,7], Nolan T. Hartwick[1], Semar Petrus[1,8], Emadeldin H. E. Konozy[2], Nisa Karimi [3,4], Levi Yant [5,6] & Todd P. Michael [1] ✉

Baobab (*Adansonia digitata*) is a long-lived tree endemic to Africa with economic, ecological, and cultural importance, yet its genomic features are underexplored. Here, we report a chromosome-level reference genome anchored to 42 chromosomes for *A. digitata*, alongside draft assemblies for a sibling tree, two trees from distinct locations in Africa, and *A. za* from Madagascar. The baobab genome is uniquely rich in DNA transposons, which make up 33%, while LTR retrotransposons account for 10%. *A. digitata* experienced whole genome multiplication (WGM) around 30 million years ago (MYA), followed by a second WGM event 3–11 MYA, likely linked to autotetraploidy. Resequencing of 25 trees identify three subpopulations, with gene flow across West Africa distinct from East Africa. Gene enrichment and fixation index (*Fst*) analyses show baobab retained multiple circadian, flowering, and light-responsive genes, which likely support longevity through the UV RESISTANCE LOCUS 8 (UVR8) pathway. In sum, we provide genomic resources and insights for baobab breeding and conservation.

The African baobab (*Adansonia digitata*) is a deciduous tree belonging to the Malvaceae family, specifically within the Bombacoideae subfamily. Hereafter, it will be simply referred to as "baobab" with other species like the Australian or Malagasy baobab mentioned as needed in the text. One of the earliest references to baobab was made by Ibn Batuta in the 14th century, who described it as a food and a large, long-living tree in Africa[1,2]. Colloquially, the baobab is referred to as the 'upside down tree' since when it loses its leaves, the branches look like roots; in addition, due to the Hollywood blockbuster "The Lion King", baobab is also referred to as the "Tree of Life".

Baobab offers various edible parts, i.e., seeds, leaves, roots, flowers, and powdery fruit pulp. The fruit is particularly rich in vitamin C, antioxidants, anti-inflammatory compounds, minerals, and fiber.

Beyond its dietary benefits, the bark is used in crafting robes and mats, adding to the economic importance of the baobab tree. Furthermore, the seeds yield oil used in cosmetics[3]. Baobab seeds contain phytic acids, just like legume seeds; however, proper processing can reduce these acids[4]. The recent approval of baobab as a food ingredient by the European Commission and the United States Food and Drug Administration (FDA) has significantly increased demand for baobab products outside of Africa. The estimated value of baobab products was US$8.2 billion in 2022 and is anticipated to reach US$12.1 billion by 2030[5]. Thus, there is economic interest and social need for genomic resources to study, preserve, and increase baobab yields[6].

Baobabs are among the oldest and largest non-clonal organisms, living over 2400 years with canopies exceeding 500 m³ and trunks

[1]Plant Molecular and Cellular Biology Laboratory, The Salk Institute for Biological Studies, La Jolla, CA, USA. [2]Biomedical and Clinical Research Centre (BCRC), College of Health and Allied Sciences, University of Cape Coast, Cape Coast, Ghana. [3]Missouri Botanical Garden, Science and Conservation Division, St. Louis, MO, USA. [4]Department of Botany, University of Wisconsin - Madison, Madison, WI, USA. [5]School of Life Sciences, University of Nottingham, Nottingham, UK. [6]Department of Botany, Faculty of Science, Charles University, Prague, Czech Republic. [7]Present address: Noblis, Inc., Washington, DC, USA. [8]Present address: Cepheid, Sunnyvale, CA, USA. ✉e-mail: tmichael@salk.edu

about 35 meters wide[7]. Unlike most large trees, baobabs are succulents, lacking "growth rings" or true wood[8]. Achieving maturity in the wild is challenging due to predation by caterpillars and larger animals[9]. Despite having bisexual flowers, baobabs are mostly self-incompatible and rely on external pollinators like bats, bush babies, and hawkmoths[10–15]. As an obligate outcrosser, *A. digitata* shows high genetic diversity, seen as heterozygosity in the genome (Fig. 1a, b; Table 1)[11,15]. Understanding baobab ploidy is key for breeding, as polyploidy influences growth and adaptability. In the wild, baobabs can take over 100 years to flower[16,17]. Additionally, genome size knowledge guides sequencing projects[18,19]. Despite thriving in harsh, low-rainfall environments, recent reports show rising baobab mortality[7,20]. For instance, the 1400-year-old Chapman baobab died suddenly in 2016[21], with others like Panke, the oldest, dying in recent years[7,9]. Suspected causes include rising temperatures, pathogens, soil compaction, and overexploitation, but limited research and genome resources hinder clear conclusions[7,16,22–24].

The *Adansonia* genus has eight recognized species, with six species endemic to Madagascar: *A. grandidieri*, *A. perrieri*, *A. rubrostipa*, *A. madagascariensis*, *A. za*, and *A. suarezensis*, while *A. digitata* and *A. gregorii* are indigenous to mainland Africa and Australia, respectively[15,20,24]. Madagascar is considered the center of origin for baobabs, with ongoing interspecific hybridization when species are in sympatry[24]. While seven baobab species are widely acknowledged as diploids ($2n = 2x = 88$)[15,25,26], *A. digitata* is a tetraploid[27]. Pettigrew et al. claimed to have found a diploid species in mainland Africa, *A. kilima*, but this hypothesis has not been supported by subsequent research[15,28]. Historical reports for African baobab suggested chromosome numbers of $2n = 96–168$, accompanied by genome sizes ranging from 3 to 7 pg 2 C/holoploid[15,25,29]. However, the ploidy of *A. digitata* remains uncertain, with questions about whether it is diploid, autotetraploid, or allotetraploid[20].

Here, we report genome size estimates for all eight recognized baobab species using a *K*-mer-based method from short-read genomic sequences. This method provides independent estimates comparable to flow cytometry[20,25]. Our primary objectives are to create a reference genome, explore genomic features, and examine evolutionary history. We generate haploid chromosome-scale assembly of *A. digitata* (Ad77271a; originally from Tanzania) alongside draft assemblies for sibling tree (Ad77271b), two trees from distinct locations in Africa [Kord Bao (AdKB) from Sudan and Okahao Heritage Tree from Namibia (AdOHT)], and a related species *A. za* from Madagascar (Aza135). Additionally, we resequence 25 *A. digitata* trees representing different regions of Africa to assess genetic diversity. Our findings reveal: (a) a high proportion of DNA transposons over LTR retrotransposons, indicating unusual genomic structure in baobab; (b) four DNA Mutator-type transposons possibly contributing to centromere formation; (c) retention of nearly complete chromosomes after whole-genome multiplication 30 million years ago, leading to gene expansion for flower development, chromatin regulation, and exocytosis; (d) expansion of *UV RESISTANCE LOCUS 8* (*UVR8*) genes, suggesting a unique genome protection mechanism in long-lived trees; and (e) genetically distinct Namibian baobab populations, highlighting the need for targeted conservation. These insights could facilitate baobab breeding, conservation, and domestication.

## Results

### Adansonia species genome sizes and heterozygosity

Historically, Feulgen staining and flow cytometry have been used to estimate genome size[30]. Cytological methods suggested *A. digitata* has 42 chromosomes, a haploid genome size of 920 Mb, and a 2C-DNA value of 3.8 pg[25,30]. We skim-sequenced all eight recognized baobab species and used *K*-mer frequency analysis to estimate genome sizes, which range from 646 Mb in *A. perrieri* to 1.5 Gb in *A. grandidieri*.

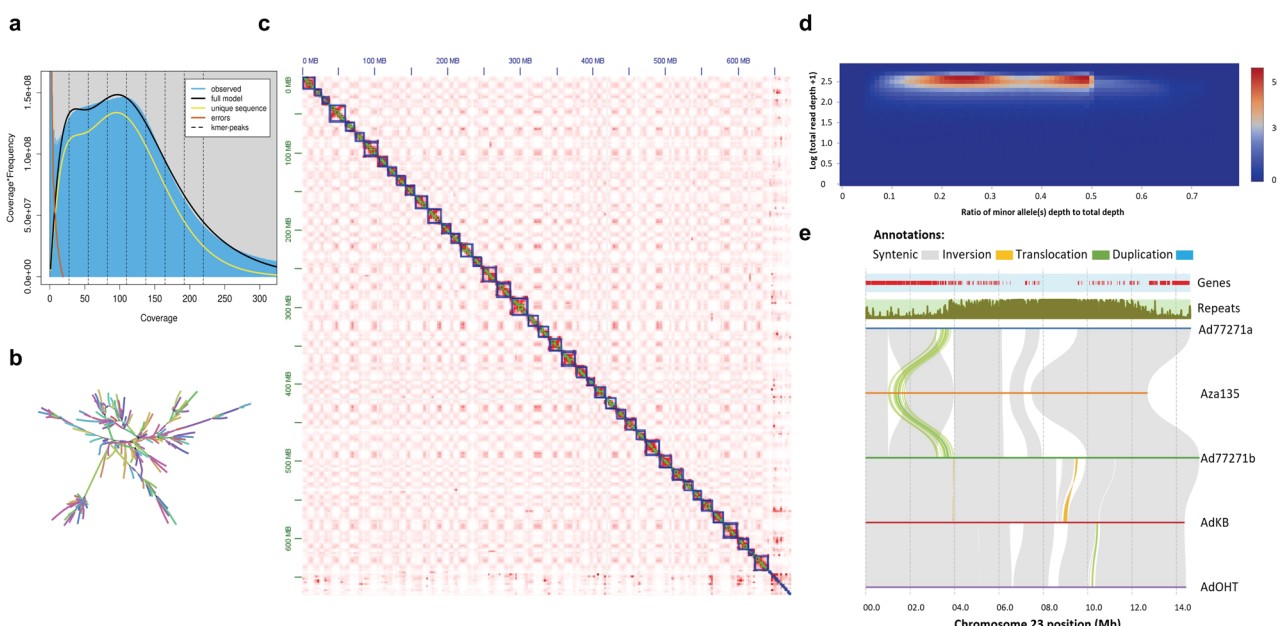

**Fig. 1 | Characteristics of baobab genomes. a** GenomeScope estimation of *A. digitata* genome size using 19-mer sequence counts and ploidy set to 4, the *K*-mer frequency depicts a unimodal pattern suggesting a diploid homozygous genome with a size of 659 Mb. **b** Assembly graph of Ad77271a suggests that there are both stretches of heterozygosity as well as transposable elements (TEs) that impact genome assembly. **c** Hi-C contact map of *A. digitata* (Ad77271a) shows the 42 chromosomes and the shared centromere sequence across the chromosomes. The bottom right corner was unscaffolded centromere sequence. **d** Two-dimensional histogram depicts tetraploid based on minor allele frequency coverage. For diploid organisms, a single peak is expected. However, for tetraploid organisms, the histogram should exhibit two peaks, approximately located at 0.25 and 0.5, respectively. **e** Structural rearrangements and synteny between *A. digitata* (Ad77271a, Ad77271b, AdKB, and AdOHT) and *A. za* (Aza135). A translocation on chromosome 23 distinguishes *A. digitata* from *A. za* species. Gray, orange, green, and blue-ribbon colors represent syntenic, inversion, translocation, and duplication structural variations, respectively. The tracks above the structural variant ribbons in the panel depict the distribution of genes and repeat sequences along chromosome 23.

**Table 1 | *K*-mer-based genome size estimates across the eight *Adansonia* species**

|  | *A. digitata* | *A. madagascariensis* | *A. perrieri* | *A. za* | *A. gregorii* | *A. grandidieri* | *A. rubrostipa* | *A. suarezensis* |
|---|---|---|---|---|---|---|---|---|
| Genome size (bp) | 659,249,037 | 647,683,220 | 645,677,030 | 647,031,624 | 701,880,489 | 1,528,554,306 | 1,336,192,477 | 752,433,438 |
| Unique genome (bp) | 376,813,370 | 480,642,287 | 470,932,522 | 471,700,912 | 507,860,224 | 1,043,221,753 | 574,147,066 | 526,812,479 |
| Repeat genome (bp) | 282,435,668 | 167,040,933 | 174,744,508 | 175,330,713 | 194,020,266 | 485,332,552 | 762,045,411 | 225,620,959 |
| Repeat fraction (%) | 42 | 26 | 27 | 27 | 28 | 32 | 57 | 30 |
| Heterozygosity (%) | 2 | 2 | 2 | 2 | 1 | 1 | 2 | 1 |

Genome sizes of *A. grandidieri* and *A. rubrostipa* were double the predicted haploid size, with *K*-mer peaks indicating potential hybridization, as has been reported between *A. rubrostipa* and *A. za*[24]. However, sampling or methodological artifacts may affect these estimates, requiring further validation. The *K*-mer-based estimates were largely consistent with the previous data[23–25,31]. Repeat content varied from 25.8% to 57%, with *A. rubrostipa* having the highest repeats. Heterozygosity ranged from 1.1% to 1.8%, with *A. digitata* showing one of the highest levels (1.6%), likely due to its outcrossing nature[11,15] and autotetraploidy[27] (Table 1).

**Baobab genome assembly, annotation, and ploidy estimation**

The GRIN conserves *A. digitata* trees named "PI77271" (Supplementary Fig. 1), which we chose because it's publicly accessible for research. Using long-read Oxford Nanopore Technologies (ONT), we sequenced two sibling seedlings, "Ad77271a" and "Ad77271b", whose genome assemblies produced 1780 and 2430 contigs, respectively (Supplementary Data 1, Supplementary Fig. 2). Ad77271a was scaffolded into 42 chromosomes using Hi-C (Fig. 1c; Fig. 1d), consistent with the haploid chromosome number ($2n = 4x = 168$) in *A. digitata*[25]. The final haploid genome size for Ad77271a had a higher N50 of 15 Mb (Supplementary Data 1). This assembly aligns with two recently published *A. digitata* genomes: one at 686 Mb using short reads[23] and another at 668 Mb using PacBio CLR, which had an N50 of only 1 Mb[24].

Additionally, using ONT reads, we assembled draft genomes for two *A. digitata* trees: Kord Bao (AdKB) from Sudan and Okahao Heritage Tree (AdOHT) from Namibia, as well as *A. za* (Aza135) from Madagascar (23°12'44.0"S, 44°02'32.2"E) (Supplementary Data 1 and 2; Supplementary Fig. 3). These assemblies had modest N50 lengths (600–800 kb) compared to Ad77271a and Ad77271b genomes (Supplementary Data 1), but sufficient for analyzing structural variations, nucleotide diversity, and polyploidy (Supplementary Fig. 4). BUSCO analysis showed ~98% completeness for Ad77271a, Ad77271b, and AdKB, with slightly lower scores for AdOHT (94%) and Aza135 (93%) (Supplementary Fig. 3a). The LTR Assembly Index (LAI) further confirmed assembly completeness (Supplementary Fig. 3).

Several chromosome rearrangements were found between *Adansonia* species, with the predominant differences occurring in the centromere region (Fig. 1e; Supplementary Figs. 5 and 6). These unaligned regions may result from unassembled repeat sequences or reflect true centromere differences, especially between *A. digitata* and *A. za*. The inversion count varied from 28 between Ad77271a and Aza135 to 135 between Ad77271b and AdKB. The highest translocation count was between Ad77271b and AdKB (359), followed by AdKB and AdOHT (333) (Supplementary Data 3). These structural differences were observed despite the high nucleotide similarity (Supplementary Fig. 7a), as seen in other autotetraploid genomes[32].

The only large (~1 Mb) translocation identified was between *A. digitata* and Aza135 on chromosome 23, moving approximately 120 genes from near the centromere to positions closer to the telomere (Fig. 1e). Gene ontology (GO) enrichment linked this region to RNA splicing and included genes like *HISTONE-FOLD COMPLEX 2* (*MHF2*) and *REPLICATION PROTEIN A 1B* (*RPA1B*), which are involved with DNA replication, repair, recombination, and transcription, as well as the latter being involved in telomere length[33–36]. In *Arabidopsis thaliana*,

loss of the *RPA1B* increases sensitivity to UV-B light, causing DNA damage and inhibiting root growth[35]. While telomere protection[37,38] and DNA repair are key in long-lived organisms[39,40], the translocation itself doesn't directly link to longevity, as *A. za* can live for 900 years[41]. Additionally, gene flow among baobab species indicates ancient introgression followed by WGM in *A. digitata*[24].

*K*-mer frequency analysis estimated a haploid genome size of around 659 Mb (Table 1) and high heterozygosity (Fig. 1a, b), though the single *K*-mer peak suggested a homozygous diploid genome. This single peak, often seen in autopolyploids like coast redwood[42,43], led us to test the tetraploid designation of *A. digitata*[27]. A heatmap of heterozygous loci coverage showed two peaks, typical of autotetraploids (Fig. 1d; Supplementary Fig. 4). SNP analysis between Ad77271a and Ad77271b identified 3.9 million SNPs (0.65%) (Figshare raw datasets), reflecting low per-haplotype heterozygosity and underscoring the challenge of fully resolving haplotypes. Additional methods will be required to achieve a complete haplotype-resolved autotetraploid assembly, as elucidated in potato[32].

**Baobab genome organization: centromere, telomere, rDNA, and DNA methylation**

The 42 chromosomes of baobab were small, ranging from 9 to 23 Mb. The Hi-C contact map suggested the centromere sequence was highly conserved across the chromosomes, and consistent with this, we found a putative centromere repeat with a base unit of 158 bp, and a higher order repeat (HORs) of 314 and 468 bp (Fig. 2a–e). In general, the centromere arrays assembled well into 1–2 Mb regions that were both metacentric as well as acrocentric (chr12, chr24, chr26, chr28, chr35, chr38, chr39 are acrocentric; Fig. 2f). Unlike the allotetraploid *Eragrostis tef*, which has distinct centromere repeats per subgenome[44], we identified one centromere repeat in the *A. digitata* genome consistent with autotetraploidy. In addition, telomere sequences (AAACCT) were assembled on the ends of most chromosomes, and these telomeres were longer compared to other plants with maximum sequences spanning 30 kb as compared to the model plants *Arabidopsis thaliana* and *Zea mays* that have 3–5 kb telomeres (Fig. 2b; Supplementary Data 4)[37]. In the Ad77271a genome, we assembled a total of 300 Kb of the ribosomal DNA (rDNA) with only one 26S array that included 35 copies on chr38, and one 5S array that included 442 copies on Ad77271a chr01 (Supplementary Table 1).

A consistent marker of age and longevity in animals is DNA methylation[45,46]. In plants, the situation is more complex, but it has been shown that DNA methylation is linked to genome stability and epimutation[47]. DNA methylation is sometimes referred to as the fifth base because it is a chemical modification to the cytosine base of DNA that is known in plants to specify cell fate, silence transposable elements, and mark environmental interactions[48]. ONT reads enable the direct detection of 5-methylcytosine (5-mC)[49]. We leveraged these reads to look for global DNA methylation in baobab, finding average levels of 54.74% in Ad77271a, 54.94% in Ad77271b, and a higher level of 62.52% in AdKB (Supplementary Fig. 8). Consistent with previous findings[50], we observed increased DNA methylation in the putative centromere arrays (Fig. 2c–f). Additionally, we found that TEs exhibit hypermethylation, while genes display hypomethylation

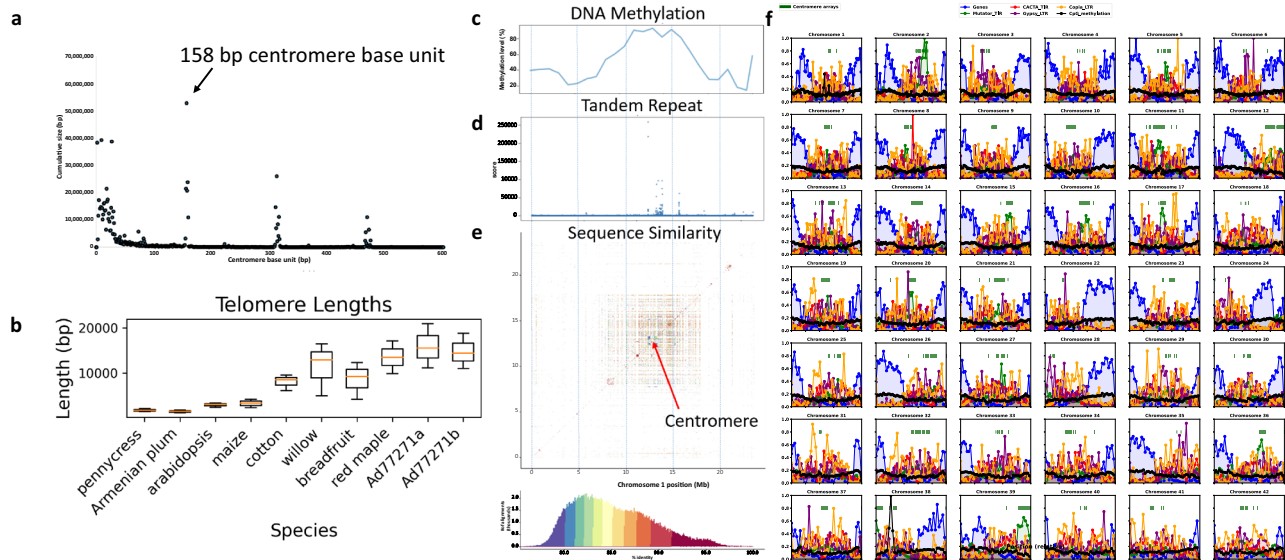

**Fig. 2 | Genomic organization of baobab. a** Predominant and large centromere array with a base unit of 158 bp with higher order repeats (HORs) of 314 and 468 bp. **b** Comparative box plots showing telomere length distribution between baobab (Ad77271a and Ad77271b) and eight other randomly selected plant species. Baobabs have long telomeres (AAACCCT), reaching a maximum length of 30 kb, which aligns with the renowned longevity of baobabs[81]. The box plots display the median (line inside the box), the 25th and 75th percentiles (bounds of the box), and the minimum and maximum values (whiskers). Sample sizes (n) represent the number of biological replicates (reads) for each species: *Thlaspi arvense* (n = 562), *Prunus armeniaca* (n = 590), *Arabidopsis thaliana* (n = 16), *Zea mays* (n = 62), *Gossypium* (n = 1366), *Salix dunnii* (n = 1917), *Artocarpus nanchuanensis* (n = 605), *Acer rubrum* (n = 1650), *A. digitata* AdPI77271a (n = 344), *A digitata* AdPI77271b (n = 578). Data represent biological replicates, and no technical replicates were used. **c** Plot showing elevated DNA methylation (5-methylcytosine) levels in centromeric regions of chromosome 1. **d** Increased occurrence of tandem repeats within the centromeric region. **e** The meta-centromeric region of chromosome 1 is shown on a pairwise sequence identity heatmap. The scale below shows the percentages of sequence similarity. The blue dotted lines show regions along the chromosome. **f** Global distribution of genome features in *A. digitata*. Centromere arrays are plotted as dark green bars at y = 0.8. Genes are represented in blue, Mutator transposons in green, CACTA transposons in red, Gypsy retrotransposons in purple, Copia retrotransposons in orange, and CpG methylation in black. Transposable elements, such as Mutator transposons, tend to be located opposite to genes, and in the centromeric regions, Methylation is generally present throughout the genome. Plot window positions are divided by the chromosome total length to create a relative scale between 0 and 1. Source data are provided as a Source Data file.

(Supplementary Fig. 9). These observations align with expected methylation patterns seen in other angiosperms[50].

## High accumulation of DNA transposons and the potential role of mutators in centromere creation or repositioning

We performed an ab initio TE prediction for Ad77271a, identifying 378,634 TEs spanning 296 Mb (~43% of the genome) (Fig. 3a; Supplementary Data 1). In most plant genomes, long terminal repeat retrotransposons (LTR-RTs) constitute the largest fraction of TEs due to their copy-and-paste mechanism of proliferation, which leads to genome bloating[51]. However, in the Ad77271a genome, LTR-RTs comprised only 10% of the TEs complement, while DNA TEs, which proliferate by a cut-and-paste mechanism, made up 33% of the genome (Supplementary Data 1). Among DNA TEs, 'Mutator-type' elements were predominant, a pattern observed across all baobab genomes (Fig. 3a; Supplementary Data 5). Moreover, Copia LTR and CACTA elements showed signs of recent bursts. Investigation of TE insertion times using synonymous substitution (Ks) values showed that most TEs were inserted earlier than 10 million years ago (MYA). We also found more recent TE insertions, particularly in the autotetraploid *A. digitata*, peaking around 2–3 MYA (Fig. 3b; Supplementary Fig. 6).

Further analysis of the relationship between TEs and centromeres across the 42 chromosomes of *A. digitata* revealed that most centromeric regions were flanked by four Mutator elements (TE_00000631, TE_00000845, TE_00000927, and TE_00000967) (Fig. 3c and Supplementary Fig. 5). A secondary analysis of the chromosomes and genes after WGM suggested that Mutator element expansion likely occurred approximately 11 MYA (Fig. 3b). While it is more common for LTR-RTs to nucleate centromere arrays[52], here the proliferation of the Mutator element is likely associated with the

establishment of centromeres. Conversely, LTR-RTs exhibited a high ratio of solo to intact elements in Ad77271a chromosomes (Fig. 3d), indicating that genome-purging mechanisms are adopted to counterbalance the genome size amplification[51].

## Gene expansion, contraction, and comparative orthogroup analysis

Baobab is unlike most large, long-lived trees because it is succulent, and when it dies, its "wood" seems to "deflate" or mush; wood soaked in water will completely disintegrate after several days, leaving only fibers that are used as packing materials[8]. We predicted and annotated genes across the baobab genome assemblies to compare against other tree genomes and closely related species like cotton, cacao, and bombax. Leveraging both ab initio and protein homology gene prediction, we estimated an average of 44,000 genes in *A. digitata* (Ad77271a, Ad7721b, AdKB, and AdOHT), and a slightly higher number in Aza135 (46,890).

Gene family comparisons between baobab and fifteen plants identified a total of 40,312 orthogroups (OG) (Fig. 4a–c). In the Malvaceae, including baobab, cotton, cacao, durio, and cotton tree, alongside representatives from various plant families, identified 3,851 expanded gene families shared between baobab and cacao, 799 gene families expanded in baobab only, and 232 gene families expanded in cacao only (Fig. 4d). Among contracted gene families, 596 were shared between baobab and cacao, with 1465 specific to baobab and 5678 specific to cacao (Fig. 4d). While there were 107 and 283, Ad77271a and Ad77271b specific OG respectively (Supplementary Table 2), the largest shared specific OGs (4626) were baobab-specific (both Ad77271a and Ad77271b), confirming a shared evolution event between the baobab siblings (Fig. 4a). The second largest shared OG

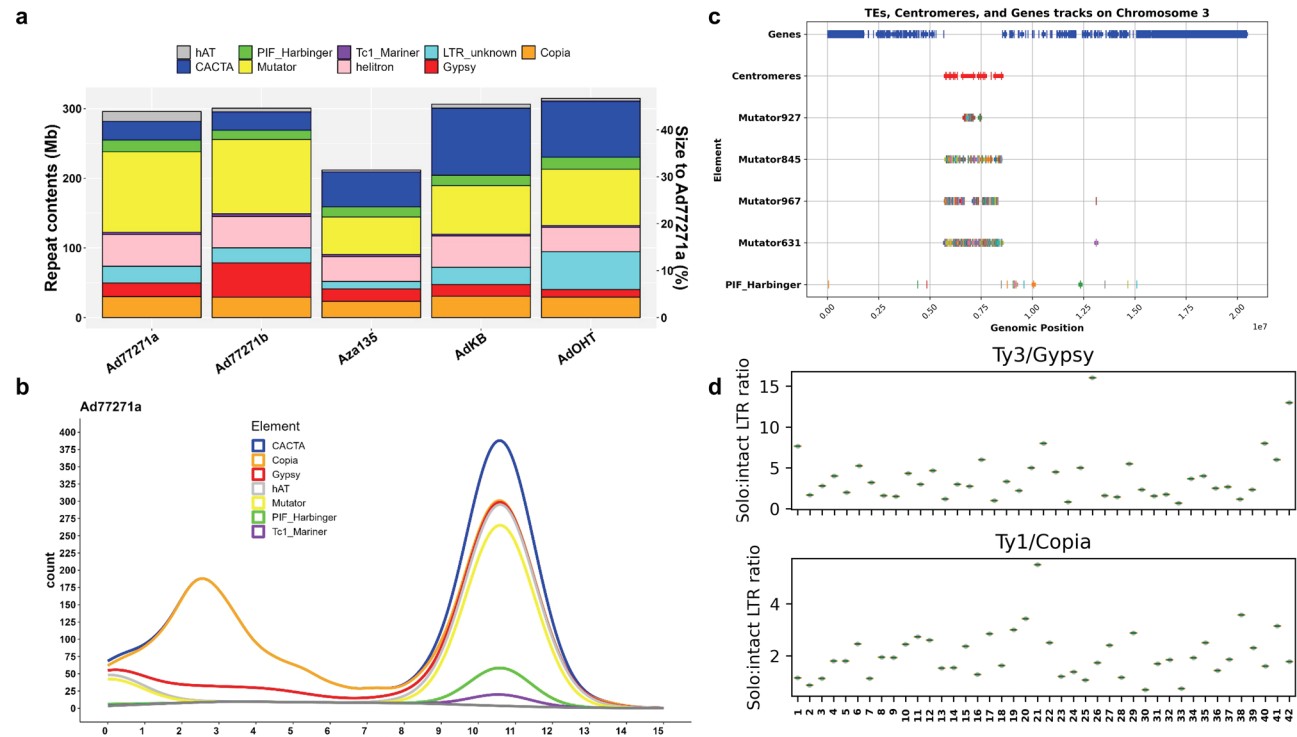

**Fig. 3 | Insights into transposable elements (TEs) in baobab. a** Bar plot comparison of TEs classification and sizes in five baobab genomes: Ad77271a, Ad77271b, Aza135, AdKB, and AdOHT. Aza135 is highlighted in the center to emphasize TE divergence. Except for Aza135 (*A. za*), the other four genomes belong to *A. digitata*. Distinct colors in the legend denote TE classes. The TE proportion to genome size (Mb) and percentage are displayed on the left and right y-axes, respectively. **b** Density plots of intact TEs, displaying the distribution of different types of TEs. Mutator transposons showed a burst around 11 million years ago. **c** Accumulation of

DNA mutators in the centromeric regions of the Ad77271a chromosome 3. Four mutator Terminal Inverted Repeat (TIR) transposons (TE_00000631: Mutator631, TE_00000927: Mutator927, TE_00000845: Mutator845, and TE_00000967: Mutator967) with a potential role in centromere creation or repositioning are displayed. **d** High solo to intact LTR ratio in Ad77271a chromosomes shows an aggressive purging mechanism following WGM, contributing to a smaller genome size (659 Mb). Source data are provided as a Source Data file.

was between monocots, while cotton genomes were the third largest shared OG group with 1723 and 1263 respectively (Fig. 4a). Further gene family comparisons between *A. digitata* (Ad77271a, Ad77271b, AdKB, AdOHT) and *A.za* (Aza135) with 15 other plant species revealed 93% of genes in orthogroups and 6399 orthogroups with all the twenty species (Supplementary Fig. 10)

Leveraging GO enrichment analysis, we asked which biological categories were specific to baobab. We identified 490 significant (FDR < 0.01) GO terms that could be clustered into three broad categories of metabolic processes (GO:0008152; right top), response to stimulus (GO:0050896; left top), and a more dispersed group that included growth (GO:0040007), chromatin (GO:0006325) immune system process (GO:0002376) and circadian rhythm (GO:0007623) (Fig. 4b; Supplementary Data 6; Supplementary Fig. 11). The third grouping represents genes associated with the long-lived nature of baobab. For instance, there are six baobab *UV RESISTANCE LOCUS 8* (*UVR8*) genes, represented in four OG families, three of which are only found in baobab genomes. *UVR8* is a UV-A/B photoreceptor that interacts with the E3 ubiquitin-ligase *CONSTITUTIVELY PHOTO-MORPHOGENIC1* (*COP1*) and *SUPPRESSOR OF PHYA-105* (*SPA*) to stabilize and destabilize two central growth-regular transcription factors *ELONGATED HYPOCOTYL 5* (*HY5*) and *PHYTOCHROME INTERACTING FACTOR 5* (*PIF5*) respectively[53–56] (Fig. 4c). UV-B radiation has the potential to damage macromolecules such as DNA and impair cellular processes, suggesting the baobab UVR8 proteins may play a broad signaling role to protect the genome of this long-lived tree. It has been shown that UVR8 may interact directly with chromatin based on its orthology to *REGULATOR OF CHROMATIN CONDENSATION 1* (*RCC1*) and nucleosome binding assays[57]. Consistent with this, we observe

enriched OG for DNA repair (GO:0006281), DNA damage response (GO:0006974), chromatin organization (GO:0006325), and remodeling (GO:0006338), consistent with baobab actively protecting its genome through UVR8-chromatin regulation (Fig. 4c).

## Evidence of an ancient WGM event in baobab genome

We performed comparative genomics of the baobab genomes Ad77271a, Ad77271b, AdKB, AdOHT and Aza135 with three closely related Malvaceae species, cotton (*G. raimondii*), bombax (*B. ceiba*) and cacao (*T. cacao*), as well as grape (*V. vinifera*) that only has one whole-genome triplication (WGT) and amborella (*A. trichopoda*) that is sister to the eudicot lineage and lacks WGM[58,59]. Synteny-based and rates of synonymous substitutions (Ks) were used to estimate WGM as well as to understand its relationship with other species, such as *A.za* (Aza135). The Ks analysis revealed a consistent timing of the separation of baobab-amborella and baobab-grape at 128 and 96 MYA. In contrast, bombax and Aza135 genomes diverged from Ad77271a around 20 MYA and 17 MYA (Fig. 5a–c)[58,59]. Following the split, Aza135 experienced chromosome rearrangements, resulting in smaller chromosomes (Supplementary Fig. 12). The Aza135 genome also showed a second peak around 30 MYA, indicating remnants of WGM. Further alternative analysis using a calibrated gene-tree species-tree confirmed that the WGM for bombax and Aza135 occurred after their divergence from cacao (Supplementary Fig. 13). Additionally, the calibrated time tree revealed that cotton species diverged ~5.0 MYA, congruent with other studies[60,61].

The self-self alignments (paralogs) of genomes provided insights into the timing of WGM events. This analysis clarified that all of the baobab, bombax, and cotton genomes experienced WGM around 30

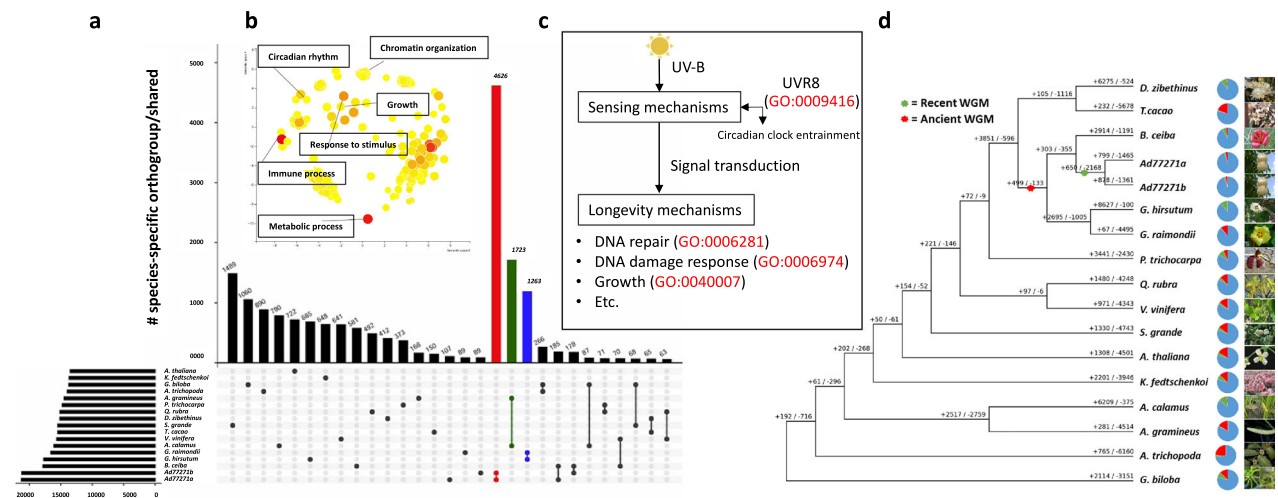

**Fig. 4 | Orthogroups, gene ontology, longevity pathway, and evolutionary dynamics of African baobab tree. a** An UpSet plot shows elevated species-specific orthogroups; red, green, and blue bars correspond to baobabs, acorus (monocots), and cotton, respectively. **b** Enriched gene ontology terms in baobab include chromatin organization and circadian rhythms. **c** Ultraviolet-B (UV-B) reception pathway via *UV RESISTANCE LOCUS 8* (*UVR8*) followed by clock gene entrainment, signal transduction, and acclimation responses[53]. **d** Comparative gene evolution in baobab alongside 15 other plant species is shown in the phylogenetic tree. Positive and negative numbers represent gene family expansion and contraction, respectively. Notable enriched GO terms include signal transduction in response to DNA

damage (GO:0042770) and circadian rhythm (GO:0007623), highlighting key biological processes that contribute to the long-term survival of baobab. Conversely, contracted genes are generally associated with plant growth, such as the negative regulation of stomatal complex development (GO:2000122), which imposes limitations on growth, nutrient uptake, and overall adaptability (Supplementary Data 6, 8, and 13). The pie chart's blue color indicates no evident change, while green and red denote expansions and contractions in gene families. Colored asterisks represent the point of whole-genome multiplication, while images on the right show the plant's inflorescence.

MYA[62], while cacao underwent its last WGM of approximately 118 MYA, consistent with it only experiencing the WGT-γ event, as previously reported (Fig. 5c)[63]. Focusing on the baobab complex, we observed a minor Ks peak around 30 MYA across all genomes, along with additional peaks at 4, 6, 11, and 17 MYA for Ad77271b, AdKB, AdOHT, and Aza135 respectively (Supplementary Fig. 6d). We hypothesize that these peaks represent speciation and polyploidization events. The recent WGM in *A. digitata* resulted in an autotetraploid genome, as further evidenced by minor allele frequency coverage showing two peaks, indicating a chromosomal cluster from the two pairs of homologous chromosomes (four subgenomes) (Fig. 1d). This WGM in *A. digitata* is estimated to have occurred between 4 AND 11 MYA, with additional evidence provided by a burst of TEs and time-calibrated gene tree (Fig. 3b; Supplementary Fig. 13).

The self-self (paralog) and pairwise (ortholog) alignment of the haploid baobab genomes revealed a 4:4 syntenic depth like the pairwise alignments of cotton and bombax consistent with them sharing the WGM (Fig. 5a, c; Supplementary Fig. 14). In contrast, cacao displayed a 4:1 syntenic depth with the baobab and cotton genomes, and a 5:1 syntenic depth with bombax, confirming it did not share the recent WGM with these genomes as seen by Ks and time tree analysis (Fig. 5a, d; Supplementary Fig. 14). It was postulated that the cotton genome experienced a WGM that was a decaploidization[64,65], and this WGM history has been clarified with a high-quality balsa (*Ochroma pyramidale*) genome to suggest an allohexaploid lineage hybridized to form an allodecaploid lineage sister to the Bombacoideae and Malvoideae[66]. Recently, a higher quality chromosome-scale bombax genome as well as the closely related kapok (*Ceiba pentandra*) genome were found to have a syntenic depth of 5:1 with cacao consistent and a shared WGM of about 30 MYA[67]. While we find a syntenic depth of 4:1 between cacao and cotton/baobab, 8.2 and 6.4% of genes are found in a 5:1 syntenic depth, respectively, which is similar to the 14.6% of genes having a syntenic depth of 5:1 between cacao and bombax, suggesting the underlying WGM in baobab could be a decaploidization event[66].

Plants generally return to diploidy over time through a process called fractionation, which involves gene loss and chromosomal fusions[68,69]. However, in all baobab genomes, as well as bombax, kapok, and balsa, the last WGM remains mostly unfractionated, with up to five chromosomes retained compared to one in cacao (Fig. 5a; Supplementary Fig. 15; Supplementary Data 7). In Ad77271a, we found that 15%, 24%, 30%, 13%, and 6% of genes were retained in 1, 2, 3, 4, and 5 copies respectively post-WGM. Since the predominant syntenic depth was 4:1 in baobab, we looked at GO enrichment analysis of quadruplicated genes, which revealed a significant (bonferroni FDR < 0.05) set of overlapping GO terms that were predominantly associated with gene regulation/chromatin, exocytosis, and flower timing/development (Fig. 5e). In contrast, cotton has undergone extensive chromosomal rearrangements, fractionation, and reshuffling, resulting in the loss of the Malvaceae or Malvadendrina protochromosomes[66].

Across most plant genomes analyzed, circadian, flowering, and light-related genes are reduced back to one or two copies in the genome during fractionation, presumably to ensure the correct gene dosage[68–70]. However, in Bombacoideae subfamily, with baobab as an example, we found that among circadian, flowering, and light orthologs[71], only six (out of 35) orthogroups had one gene copy, while more than half (18/35) contained three or four copies (Supplementary Data 7; Supplementary Data 8). In contrast to baobab, a grape that has only the WGT-γ and cotton that has both the WGT-γ and WGM have 50% and 75% of the circadian, flowering, and light orthologs respectively, while bombax has retained orthologs similar to baobab (Supplementary Data 7). One pair of genes retained in four copies in the baobab genomes were an evolutionarily conserved syntenic gene pair *ZEITLUPE (ZTL)* and *VERNALIZATION 3 (VIN3)*, which are involved with flowering time, thermomorphogenesis, photomorphogenesis, and the circadian clock, and are linked in the genome across the eudicot lineage back to amborella (Fig. 5d)[70]. Plants that leverage Crassulacean Acid Metabolism (CAM) photosynthesis to ensure stomata are open at the correct time of day (TOD)[72–75], as well as the crop soybean that highly specific latitude maturity groups[76], have retained multiple

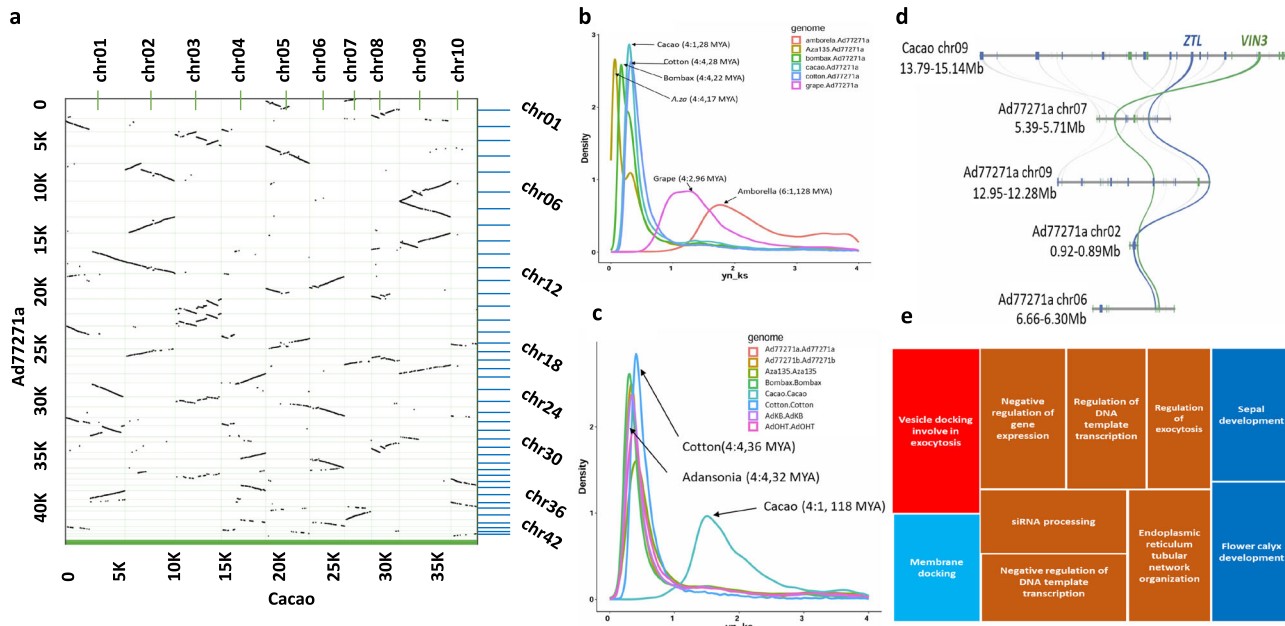

**Fig. 5 | Syntenic relationships and the Ks distribution suggest an ancient whole-genome multiplication (WGM) event in baobab. a** Dot plot illustrating that *A. digitata* genome retained many almost full chromosomes (Chr) after WGM. The Ad77271a (y-axis) genome was aligned to the cacao (*Theobroma cacao*; x-axis) genome at the protein level with MCscan (Python version with default parameters). **b** Multi-species pairwise comparison of substitution per synonymous site (Ks) showing speciation events. **c** The self-self alignments (paralogs) showing WGM events. **d** Microsynteny plot between *A. digitata* and *T. cacao* showing a chromosomal region in the *T. cacao* chromosome 9 bearing gene *ZEITLUPE* (*ZTL*) and *VERNALIZATION 3* (*VIN3*) genes, which can be tracked to four regions in *A. digitata* (blue and green lines, respectively). **e** Enriched Gene Ontology (GO) terms related to gene regulation for exocytosis and flower development in *A. digitata*. The syntenic depth ratio between genomes and the evolution event age million years ago (MYA) is indicated inside the parenthesis. Source data are provided as a Source Data file.

copies of circadian, light, and growth genes[70]; these results indicate that baobab may leverage CAM photosynthesis and a tightly regulated TOD process for flowering. These genes offer insight into how baobab has adapted to various environments across Africa (see variation analysis below).

### Diversity in baobab across Africa

*A. digitata* is endemic to mainland Africa and little is known about the genetic diversity across the continent. We resequenced 25 *A. digitata* trees from across Africa using Illumina short reads, yielding an average sequencing depth of 20x per accession (Supplementary Data 2). Mapping these sequences to Ad77271a reference produced 58.9 million SNPs and 446 thousand INDELs, accessible at https://resources. michael.salk.edu/baobab/index.html. Analysis of ploidy using these genotypes confirmed that *A. digitata* is an autotetraploid (Supplementary Data 9). Principal component analysis (PCA) of 25 accessions revealed that more than 30% of the genetic variance was due to geographical origins, mostly east and north/west (Fig. 6a; Supplementary Data 9). Population 1 encompasses germplasms from north of the equator up to approximately 16 degrees north (and mostly west), while populations 2 and 3 span from the equator to 26 degrees south. The trees from the northern desert region of Namibia, clustered into population 3 (Fig. 6a; top left; green circles), were distinct from the trees collected closer to Botswana (population 2). These results indicate there are geographical or environmental factors that have limited gene flow between these populations.

We leveraged the fixation index (*Fst*) across the three populations to determine their relatedness. The analysis revealed that population 1 and 3 were more related (lower *Fst*; average <0.1), while populations 2 and 3, as well as population 1 and 2, were less related (higher average *Fst* > 0.4) (Fig. 6b). These results were consistent with Namibian population 3 being closer to population 1, in contrast to population 2, despite their relative geographic closeness. We looked at the genes with high *Fst* SNPs (>0.8) across the three population comparisons to identify genes that are under selection for their environment, and only found enriched GO terms between populations 2v3 and 1v3 that could be summarized into pollination, organelle localization, and chromatin (Supplementary Data 10 and 11). Consistent with our findings that baobab-specific genes were related to longevity through the UVR8-chromatin connection, these genes also played a role in an environment-specific fashion. The identification of pollination genes having high *Fst* suggests there may be some selective pressure on synchronizing flowering and pollination, which is thought to be one of the factors that led to the 'rise of angiosperms'[77,78]. Similar to CAM plants that have expanded circadian clock genes to ensure TOD-specific stomatal opening, it has been shown that coordination of flower development and opening must be highly TOD-specific to pollinators[70]. This explains why many circadian clock genes have been retained in three or four copies after fractionation (Fig. 5d; Supplementary Data 7), and we find that many of these genes have high *Fst*. For instance, the circadian-linked gene pair *ZTL* and *VIN3* that were retained in four copies also had high *Fst* (Fig. 5d; Supplementary Data 12). Baobab flowers open in a tightly regulated manner, starting to bloom at sunset. They remain in bloom and receptive to pollinators throughout the night but begin to wilt by early the next morning. In East and West Africa, bats have been identified as the primary pollinators, while in Southern Africa, hawkmoths also contribute to pollination[13,79]. Taken together, the regulation of chromatin and pollination play important roles in shaping the diverse populations of baobab in Africa.

Variation and selection can be specific to regions of the chromosome; therefore, we employed a local PCA technique to identify patterns of relatedness from SNP frequencies across the genome[80]. Along the chromosomes, we observed striking regions of shared relatedness that contrasted greatly from surrounding regions. One region on chromosome 23 between 6 and 7 Mb overlaps with the

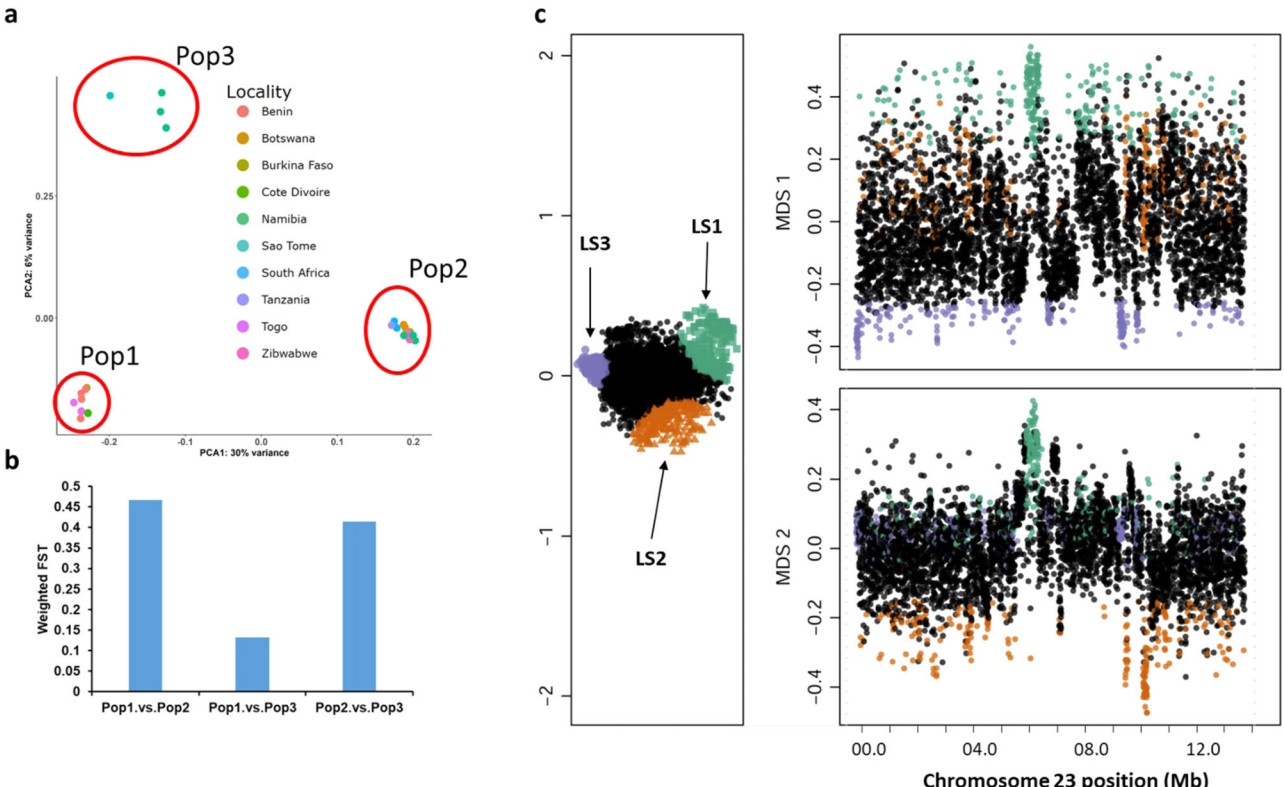

**Fig. 6 | Diversity and structural variations in *A. digitata*. a** Principal Component Analysis (PCA) of 25 *A. digitata* accessions collected across Africa using 6490 SNPs. Three clusters are present: Cluster 1 (Pop1: *n* = 8), Cluster 2 (Pop2: *n* = 13), Cluster 3 (Pop3: *n* = 4). **b** Weighted *Fst* (fixation index) values between populations: Pop1 vs. Pop2 (*Fst* = 0.47), Pop1 vs. Pop3 (*Fst* = 0.13), and Pop2 vs. Pop3 (*Fst* = 0.41). The high *Fst* values support population differentiation due to genetic structure, primarily associated with longitudinal variation (east-west). **c** Lostruct partitions vary across chromosomes. The multidimensional scaling shows the coordinates of the 25 *A. digitata* populations on the x-axis (MDS 1) and the y-axis (MDS 2). Points that are not outliers are shown in black, while points that are outliers are shown in green (LS1), orange (LS2), and purple (LS3). Each point on the graph corresponds to a specific genomic window of 3 kb. MDS stands for multidimensional scaling.

predicted centromere arrays (6–9 Mb) (Fig. 1e; Fig. 2f; Fig. 6c; Supplementary Fig. 16). This region is present in Ad77271a, Ad77271b, and AdKB (all East Africa), but lost in AdOHT (Northwest Namibia), suggesting that region could be important for the diversification of baobab across Africa (Fig. 1e). Many of the putative centromere regions in *A. digitata* had translocation, inversions, and duplications, or regions that didn't map (Supplementary Fig. 17; Supplementary Data 3), as well as high *Fst* with GO enrichment of chromatin, consistent with these regions being under active change in the baobab genomes. Baobab has a large number ($2n = 4x = 168$) of small chromosomes (9–23 Mb), suggesting that selection on chromatin stability and arrangement may play a role in longevity and environmental specificity. In a nutshell, baobab has adapted to specific environments across Africa and achieved extreme longevity by actively protecting and regulating its genetic material while retaining accurate TOD acuity through the retention of circadian, light, and flowering time genes to ensure environment-specific pollination.

## Discussion

Baobab, an iconic succulent tree emblematic of Africa's savannas, bears nutritious fruits fueling its global demand and bolstering income for rural communities across the continent. However, some of the oldest known baobabs are dying across Africa[7,81] which makes it imperative that we better understand the baobab genome to initiate conservation and breeding. Baobab belongs to the Malvaceae family, which comprises 4,225 species across 243 genera and nine subfamilies, including economically and ecologically important plants such as cotton (*Gossypium spp.*), the cotton tree (*Bombax ceiba*), cacao (*Theobroma cacao*), and baobab (*A. digitata*)[66]. WGM has played a crucial role in the evolution of these plants. For example, following the ancient core eudicot whole-genome triplication (γ) event, cotton plants have undergone repeated polyploidization and hybridization, resulting in polyploid offspring with combined genomes that diverge significantly from their diploid parents, thereby initiating new evolutionary trajectories. The decaploid ancestor of cotton (a 5× multiplication) significantly contributed to the development of spinnable fibers[24,65,66]. Similarly, polyploidy in *Bombax ceiba* has facilitated its robust growth and vibrant blooms in tropical regions[67], while ancient WGM in cacao has led to the emergence of diverse and flavorful varieties. Here, we demonstrate that baobab has a unique proportion of DNA transposons, experienced WGM, and presents a high-quality chromosome-level haploid assembly of *A. digitata* (Ad77271a) alongside draft genomes from a sibling tree (Ad77271b), two geographically diverse *A. digitata* (AdOHT and AdKB) and distinct species *A. za* (Aza135) from Madagascar. We also show a distinct baobab population in Namibia.

Gene dosage and gene balance hypotheses are well-established concepts in plant genetics, emphasizing the conservation of the functionality of specific gene pathways[68,82,83]. Baobab has undergone several rounds of WGM starting with the WGT-γ followed by what is thought to be a decaploidization[64–67], the latter of which has remained largely unfractionated with between four and five full chromosomes retained as compared to cacao (Fig. 5a; Supplementary Figs. 14g and 18). Consequently, baobab has retained evolutionarily conserved linkages between the circadian/temperature genes *LHY/PRR9* (3 copies) and *ZTL/VIN3* (4 copies) (Fig. 5d; Supplementary Data 7)[70,84,85], similar to patterns observed in soybean (*Glycine max*)[70,86], which provide environmental specificity and are correlated with maturity[76]. Our

analysis also identified unique *UV RESISTANCE LOCUS 8* (*UVR8*) genes, suggesting mechanisms for UV-B light reception and DNA repair, crucial for baobab's adaptation to arid conditions. Upon UV-B absorption, UVR8 undergoes monomerization, leading to a structural change that initiates downstream signaling events through the circadian clock and CONSTITUTIVELY PHOTOMORPHOGENIC1 (COP1) to regulate DNA damage response and repair[53,87]. Further, genes with a high fixation index (*Fst*) showed enriched terms in flower development, regulation/chromatin, immunity, and exocytosis (Supplemental Fig. 11; Supplementary Data 6). We note that *Fst* is a population genetic estimation based on a diploid model. Thus we interpret our polyploid *Fst* results cautiously[88]. Baobab flowers bloom late in the afternoon and wilt within 24 h, indicating tightly controlled timing for flower development, opening, and pollinator schedules[70,84,85,89,90]. These findings posit that the retention of gene copies in baobab, especially in circadian, light, and flowering time pathways, contributes to its adaptation and longevity.

This study further confirms baobab's autotetraploid nature through analysis of the sibling genome (Ad77271b) and the 25 resequenced trees (Figs. 1d and 3; Supplementary Fig. 5; Supplementary Data 9). Initially, the *K*-mer analysis profiled a single peak with an estimated haploid genome assembly size of 659 Mb, which is in line with what was found through Feulgen staining and flow cytometry[25], suggesting a diploid genome. However, since chromosome counting indicated that it is tetraploid[25], we looked more closely at the allele frequencies. Leveraging both the Ad77271b and the resequenced 25 tree reads, we analyzed coverage at heterozygous sites, which provided evidence that *A. digitata* possesses an autotetraploid genome. In addition, we identified only one centromeric repeat (base unit = 158 bp) supporting autotetraploidy, since allotetraploids like *Eragrostis tef* have been shown to have distinct centromere arrays representing the different subgenome origins[44]. Finally, our Ks analysis found a recent WGM 3–11 MYA consistent with an autotetraploidy event. However, it is possible that autotetraploidy didn't immediately lead to disomic inheritance, instead, there may have been a long period of tetrasomic inheritances that could still be happening now, as has been seen in the coast redwoods[42].

Our study also revealed baobab genome is unlike any published plant genome to date in that DNA TEs are dominant (3x) compared to LTR-RT that typically result in the bloating of plant genomes[51]. Compared to other plant genomes, baobab has an average total TE content of *A. digitata* (~45%) having more than *A. za* (35%), which was also in line with the 50-60% DNA methylation levels (Supplementary Data 1; Supplementary Fig. 8). The *A. digitata* TE composition was unusual compared to other plant genomes in that the proportion of DNA TEs was 33% while the LTR-RTs were 10% (Supplementary Data 1). Typically, LTR-RTs are the predominant transposon in a plant genome since they proliferate in a "copy-and-paste" mechanism, while in contrast, DNA TEs accumulate through a "cut-and-paste" mechanism[51]. In terms of LTR-RT TEs, a big rise in Copia and CACTA TEs was seen around 10 MYA, which sets baobabs apart from its relative in the Malvaceae family, cacao[59,63]. In contrast, the DNA TEs burst around 3–11 MYA, coinciding with the putative tetraploidy event in baobab, suggesting that the tetraploidy event may have played a role in the accumulation of the cut-and-paste DNA TEs that have shaped the baobab genome. This unusual genomic composition raises the possibility that DNA transposons, particularly the four sets of mutator-type elements (Fig. 3c), play roles in recruiting centromeric histones and potentially contributing to new centromere formation, suggesting an adaptive mechanism[91].

Finally, the resequencing of 25 *A. digitata* accessions from across Africa revealed that *Adansonia* trees situated north of the equator exhibited some divergence from those located southward, with approximately 6% of the variation attributed to this distinction. However, the most partitioning, about 30% of the variation, occurred along

an east-west axis, as exemplified by the distinctiveness between populations 1 and 2 (Fig. 6; Supplementary Fig. 7). Intriguingly, Namibian baobab trees clustered into two populations (2 and 3), which correspond to different watersheds, suggesting limited gene flow due to possible geographic barriers and adaptation to distinct climate regimes[31]. Another study conducted in Niger and Mali supported this notion, indicating variations in baobab species across the continent; specifically, it found that West African germplasms exhibited faster growth and better adaptation to arid environments compared to their East African counterparts[92]. A recent study on the evolutionary history of baobab trees suggests that a diploid progenitor dispersed from Madagascar to Africa, where autopolyploidy occurred in West Africa, creating the current tetraploid *A. digitata*. This neo-polyploidy, with its ecological advantages, likely contributed to *A. digitata's* spread eastwards and southwards[24]. Additionally, the observed complex collinearity between Ad77271a from mainland Africa and Aza135 from Madagascar underscores the extent of chromosomal restructuring and divergence within the species (Supplementary Fig. 12).

In summary, this work illuminates the evolutionary history of baobab, confirming *A. digitata* as an autotetraploid ($2n = 4x = 168$) with 42 chromosomes (Fig. 1c). The retention of multiple copies of circadian, light, and flowering-related genes following WGM highlights their roles in baobab's strategies for longevity and pollination[70]. The identification of *UVR8* as key for stress resilience opens new avenues for breeding programs to enhance adaptation. The baobab genome's high proportion of DNA transposons, compared to LTR retrotransposons, is unique among plant genomes, highlighting its unusual structure. Additionally, the identification of specific Mutator transposons (TE_00000631, TE_00000845, TE_00000927, and TE_00000967) suggests they may play a role in centromere formation or repositioning. The isolation of genetically diverse baobab populations in Namibia underscores the need for targeted conservation efforts. These findings highlight the key role of genomics in conserving baobab, advancing domestication, and optimizing agricultural potential[60,93,94]. Our study also provides a detailed view of the genome's structural organization, including centromeres, telomeres, ribosomal DNA, and patterns of DNA methylation.

## Methods

### Plant materials
Seeds were obtained from the USDA Germplasm Information Resource Network (GRIN) from three trees grown in the USDA-Agriculture Research Service, Subtropical Horticulture Research Station, Miami, FL, USA, under the accession number PI77271. The original seed came from Dar es Salaam, Tanganyika Territory, Tanzania, Africa, in 1928. The PI77271 tree was chosen to generate the reference genome to enable broad access to the baobab germplasm through GRIN. Seed for the *Adansonia* species to estimate genome sizes was ordered from Le Jardin Naturel (https://www.baobabs.com/). N.K. and E.H.E.K. (coauthors of this paper) collected leaf and seed material for the resequencing of 25 baobab species. The seed was cleaned, soaked in boiling water for three days, and then planted in well-drained soil. Seedlings were grown to the first true leaf stage and then dark-adapted for two days for DNA and RNA extraction (Supplementary Fig. 1a).

### DNA and RNA extraction
Baobab has been a difficult species to extract high molecular weight (HMW) DNA due to the large amounts of polysaccharides that it produces. Therefore, we employed two different methods to obtain HMW DNA from baobab: first, seedlings at the two true leaf stages were used for DNA extraction, and second, they were dark-adapted for two days to deplete the polysaccharides. After two days of dark adaptation, two PI77271 seedlings were chosen for genome sequencing and named "Ad77271a" and "Ad77271b"; These sibling seedlings were chosen for sequencing to enable analysis of reported autotetraploidy. HMW DNA

was extracted from Ad77271a and Ad77271b, as well as "AdOHT" from Namibia and "AdKB" from Sudan, along with *A. za* (Aza135) from Madagascar using a modified protocol[95] (Supplementary Fig. 2). For the 25 *A. digitata* resequencing, DNA was extracted from dried leaf samples[60]. For RNA, total RNA from young-fresh leaves was extracted using the QIAGEN RNeasy Plus Kit according to the manufacturer's instructions. The RNA was quantified using the Qubit RNA Assay and TapeStation 4200. Before library preparation, DNase treatment was performed, followed by AMPure bead cleanup and rRNA depletion with QIAGEN FastSelect HMR.

### Genome and transcriptome sequencing

Unsheared HMW DNA (1.5 µg) from Ad77271a, Ad77271b, AdOHT, AdKB and Aza135 was used for ONT ligation-based libraries (SQK-LSK109). Final libraries were loaded on an ONT flowcell (v9.4.1) and run on the PromethION. Illumina 2 × 150 paired-end reads were also generated for genome size estimates and polishing genome sequences. Libraries were prepared from HMW DNA using Illumina NexteraXT library prep kit and sequenced on NextSeq High Output 300 cycle using Illumina 2 × 150 paired-end kit (Illumina, San Diego, CA). The resulting raw sequence was only trimmed for adapters, resulting in >60× coverage. For the transcriptome, NovaSeq6000 platform with a 2 × 150 bp configuration was utilized.

### Hi-C library preparation

Hi-C library was prepared for Ad77271a using Phase Proximo Hi-C (Plant) kit (V.3.0) and run on Illumina NextSeq P3 300 cycle, paired-end 2 × 150 kit.

### Genome assembly analysis

ONT reads were assembled using Flye (v2.9.2)[96] and then polished using Racon (v1.5.0)[97] and Pilon (v1.24)[98] with Illumina reads. Hi-C data and Juicer version 1.6.2, 3ddna (v180419), and JBAT (v1.11.08) built the final assembly. The completeness of the genome assembly was assessed through BUSCO (v. 5.4.3), utilizing the ODB10 eudicots dataset[99].

### K-mer-based genome size estimates

Applying *K*-mer-based techniques to Illumina short reads from Illumina sequencing libraries allowed us to estimate the genome's size, repeat, and heterozygosity. Jellyfish in combination with GenomeScope2[100] were employed to assess parameters such as haploid genome length, repeat content, and heterozygosity. For analysis of ploidy, nQuire Tool[101] in conjunction with statistics of variants from Illumina short reads were utilized (Supplementary Data 9).

### Scaffolding long-read assembly contigs

Ragtag (v2.1.0) was used to scaffold the contigs of Ad77271b, AdOHT, AdKB, and Aza135 with the Hi-C scaffolded Ad77271a.

### Repeats and gene prediction

EDTA (v1.9.6)[102] was used to construct a repeat library and softmask complex repeats. Tandem Repeats Finder (v4.09)[103] was employed to identify centromere and telomere sequences, as well as softmask simple repeats. The ONT cDNA library was mapped against the reference using Minimap2 with splice presets and then assembled using Stringtie (v2.2.1) with the long reads processing flag. Genes in baobab were predicted via the Funannotate (v1.8.2; https://github.com/nextgenusfs/funannotate) pipeline with the Stringtie transcript models included as evidence in addition to the UniProt protein set. Predicted proteins were characterized using Eggnog-mapper v2.0.1[104].

### Solo long terminal repeat (LTR) and intact LTR retrotransposons

We used the EDTA (v1.9.6) 98 pipeline as described by Lynch et al.[105]. Briefly, solo-LTRs were identified by first collecting the set of LTRs that

were not assigned as intact LTR-RTs, which are retrieved on the basis of "method = homology" in the attribute column of the TEanno.gff3 file. We applied thresholds to isolate solo-LTRs from truncated and intact LTRs, as well as internal sequences of LTR-RTs. These thresholds include a minimum sequence length of 100 bp, 0.8 identity relative to the reference LTR, and a minimum alignment score of 300. We also required that the four adjacent LTR-RT annotations not to have the same LTR-RT ID. Further, we required a minimum distance of 5000 bp to the nearest adjacent solo-LTR, intact LTR, or internal sequence. Last, we kept solo-LTR sequences that fell within the 95th percentile for LTR lengths. LTR insertion time was estimated using a substitution rate of $4.72 \times 10^{-9}$ per year[106].

### DNA methylation analysis

LoReMe (Long Read Methylation) (Oxford Nanopore Technologies · GitHub) was used to infer DNA methylation patterns. In brief, the process involved the conversion of ONT FAST5 data to POD5 format using the Loreme Dorado-convert tool v.0.3.1. Subsequently, super-high-accuracy base calling was performed, aligning the sequences to the reference genome of *A. digitata* (Ad77271a). Modkit v0.1.11 was employed to generate a bed file containing comprehensive methylation data, enabling us to create visual representations of methylation profiles for further investigation and interpretation.

### Orthology analysis

OrthoFinder (v2.5.5)[107] was used for comparative genomics of Malvaceae: *A. digitata* (Ad77271a and Ad77271b), cotton (*Gossypium raimondii* and *Gossypium hirsutum*), cacao (*Theobroma cacao*), durio (*Durio Zibethinus*) and cotton tree (*Bombax ceiba*). Additionally, we examined representatives from Vitaceae (*Vitis vinifera*), Brassicaceae (*Arabidopsis thaliana*) Salicaceae (*Populus trichocarpa*), Fagaceae (*Quercus rubra*), Myrtaceae (*Syzygium grande*), Crassulaceae (*Kalanchoe fedtschenkoi*), Acoraceae (tetraploid *Acorus calamus* and diploid *Acorus gramineus*), Amborellaceae (*Amborella trichopoda*) and Ginkgoaceae (*Ginkgo biloba*). Except for the baobab, the primary proteins were downloaded from phytozome (v13)[108] and websites (Supplementary Table 2). Gene family size in the context of phylogeny was analyzed using CAFE v5.0[109]. Orthogroups with lots of genes in one or more species (100 genes) and only present in one species were excluded[110]. Results were then visualized using CafePlotter (https://github.com/moshi4/CafePlotter).

### Synteny analysis

CoGe was used to make syntenic region dot plots for intergenomic and intragenomic alignments (https://genomevolution.org/CoGe/). MCscan (https://github.com/tanghaibao/jcvi/wiki/MCscan; Python version) was used for interspecies syntenic analysis, enabling the identification of homologous gene pairs, gene blocks, and the creation of syntenic plots that depict the relationships between homologous gene pairs between baobab and other species.

### Structural variation and rearrangement identification

Structural variations (SVs) were profiled using Syri v1.6.3[111].

### Variants analysis

In order to compare the sibling baobabs, we performed short variant calling and structural variant calling using two distinct pipelines: a short read (Illumina) based small variant calling workflow, and a long read (i.e. ONT) based structural variant calling workflow. Short reads were used for small variant calling due to their high accuracy at the nucleotide level, permitting high confidence SNP and short indel calls. Long reads were used for structural variant calling because their increased length allows for covering large structural variants and verifying their structure. For each of our sibling baobabs, we ran both variants calling pipelines using both baobabs as a method of sanity-

checking our results. Results were consistently symmetric regardless of which of the two baobabs was used as a reference.

Our short-read-based short variant calling pipeline involves four primary steps: read trimming, read alignment, variant calling, and quality filtering. Read trimming is helpful for filtering out low-quality portions of reads that could reduce variant calling accuracy and run time as erroneous base calls when aligned to the reference have to be processed as potential variants. Read trimming was conducted using Trim Galore (v0.6.6). Read alignment was conducted using minimap2 (v2.20) with short-read-appropriate settings and was then sorted using Samtools (v1.12). Each sample was mapped independently and then processed by freebayes (v1.3.5) collectively using tetraploid settings in order to call variants. Subsequent variant calls were then filtered to Q20 before being manually inspected and summarized with several stats tools: vcftools (v0.1.16) and rtg tools (v3.12). We also used an in-house developed stats tool (available from https://gitlab.com/NolanHartwick/bio_utils), which includes functionality to process coverage stats as output by freebayes in order to verify ploidy.

## Molecular dating and whole-genome multiplication (WGM)

A total of 492 single-copy orthologues, 365,810 genes, and 31,169 orthogroups from nine species: *T. cacao*, *B. ceiba*, *G. arboreum*, *G. raimondii*, *A.za* (Aza135), *A. digitata* (Ad77271a), *Carica papaya*, *Arabidopsis thaliana* and Oryza sativa were used in the analysis. Secondary calibration of molecular clocks was performed using priors from the http://www.timetree.org/ database for monocot (*Oryza sativa*) and dicot (*Arabidopsis thaliana*) (163 MYA). For alternative WGM prediction, synonymous substitution rates (Ks values) were calculated using the MCScan software (Python version) with default parameters. A substitution rate of $6.56 \times 10^{-9}$ was then utilized to estimate the time of duplication events[112].

## Ultraviolet-B radiation (UV-B) photoreceptor (UVR8) gene analysis

Baobab-specific orthologs were subjected to gene ontology enrichment (GO) analysis using Python GOATOOLS[113], and subsequently visualized using REVIGO[114]. The phylogenetic tree was analyzed via GeomeNet (https://www.genome.jp/en/about.html).

## Reporting summary

Further information on research design is available in the Nature Portfolio Reporting Summary linked to this article.

## Data availability

The genome assemblies and annotations for Ad77271a, Ad77271b, AdOHT, AdKB, and Aza135 are available through the Michael Lab genome portal [https://resources.michael.salk.edu/baobab/index.html], and are also uploaded to CoGe under ID 67790 – 67801. The raw sequencing reads can be accessed via BioProject 1022505. Source data are provided with this paper and via Figshare [https://doi.org/10.6084/m9.figshare.26039878][115].

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

## Acknowledgements

We thank members of the Michael Lab for their comments on the genome work and manuscript. We also thank the genome sequencing team at Monsanto for the initial genome size survey of the *Adansonia* species funded by the Illumina Greater Good program awarded to T.P.M. Additionally, we are grateful to Mike Winterstein, USDA, GRIN for sending seed and leaf material from *A. digitata* tree PI77271 for genome sequencing. This work was supported by the Tang Genome Fund to T.P.M., a Global Challenges Research Fund (GCRF), Nottingham Interdisciplinary Research Award, and the European Research Council (ERC) under the European Union's Horizon 2020 research and innovation programme [grant number ERC-StG 679056 HOTSPOT], via a grant to L.Y. The 25 samples used for resequencing were prepared with financial support from the National Science Foundation award DEB-1354268 to N.K. and field collecting from Diana Mayne, Sarah M. Venter, and Achille E. Assogbadjo. Finally, we are very grateful to David Baum for his constructive suggestions during the writing of the manuscript.

## Author contributions

T.P.M. and L.Y. designed the research; J.K.K., T.P.M., K.C., B.W.A., N.T.H., S.P., and N.K. performed research or analyzed data; E.H.E.K. and N.K. contributed materials and/or tools; J.K.K. and T.P.M. wrote the manuscript. All authors revised the manuscript.

## Competing interests

The authors declare no competing interests.
