## [Peer Review File · Nature Communications]

Chromosome-level baobab (*Adansonia digitata*) genome illuminates its evolutionary insights

Corresponding Author: Professor Todd Michael

Version 0:

Reviewer comments:

Reviewer #1

(Remarks to the Author)

This paper describes a haploid chromosome-level reference genome for baobab, *Adansonia digitata*, as well as draft assemblies for a sibling tree, two trees from distinct locations in Africa, and a related species from Madagascar. The paper describes a paleopolyploid origin for baobab, shared with the Malvoideae ~30 million, and a more recent autotetraploidy event 3-4 million mya that coincides with a recent burst of TE insertions. Resequencing of 25 *A. digitata* trees from Africa revealed three subpopulations that suggest gene flow through most of West Africa but separated from East Africa. I have read this paper with great interest, amongst other things because the paper describes new, high-quality genomic data of a truly iconic species. However, I have to say that I was a little disappointed after reading the paper. Please allow me to explain why.

For instance, I found the entire section on whole genome duplication (WGD) quite superficial and to contain some statements that are difficult to understand. For instance, the authors state that "The Ks revealed a consistent timing of the separation of baobab-amborella and baobab-grape at 128 and 96 MYA respectively (Fig. 5b). Both cacao and cotton diverged from the ancestor of *Adansonia* around 30 MYA, which was the first piece of evidence that a baobab-specific WGD occurred around this time." First, the approach to determine the timing of WGDs and speciation events seems only based on estimated Ks values, which is concerning since different species can have very different substitution rates. Also, how are these Ks values calculated? Which software is used to get those Ks value? Which substitution rates are used for the inference of divergence time? There is no information on this unless I have missed it. Furthermore, one must be cautious to compare Ks peak values to determine which event occurred first, a speciation or a WGD event. In my opinion, if there is some concern, dot plot analysis and in particular gene tree – species tree reconciliation methods should be applied to unequivocally (and still then ...) place WGDs in time. The authors state that "The self-alignments of genomes informed the timing of the most recent WGD in a species, which clarified that all of the baobab, bombax and cotton genomes shared the same WGD event at about 30 MYA, while cacao experienced its last WGD around 118 MYA that was consistent with it having only the WGT- γ (Fig. 5c)." What does the self-alignment of genomes mean? From the figure, I assume it refers to age distributions made from collections of paralogous genes (all genes, or only homeologs)? Finding similar Ks peak values for different organisms can indeed suggest a shared WGD, but it could also suggest independent WGDs in different branches at similar times in evolution.

Second, the entire section could do with some more detailed explaining. "Both cacao and cotton diverged from the ancestor of *Adansonia* around 30 MYA, which was the first piece of evidence that a baobab-specific WGD occurred around this time". How is this first evidence for a baobab-specific WGD? It seems problematic to infer a specific WGD event solely based on divergence time without further elucidation or reference to accompanying figures. Additionally, there's a notable absence of a summarizing sentence indicating the presence of a WGD event shared with the Malvoideae but not with cacao. Instead of ambiguously stating "around ~30 Mya," it would be more precise to assert that this WGD event occurred after the divergence with cacao but prior to the divergence within the ancestor of the Malvoideae.

The authors state that "When we zoom into just the baobab complex, we saw that all genomes have a minor Ks peak around 30 MYA in addition to peaks at 4, 6, 11 and 17 MYA for Ad77271b, 1 AdKB, AdOHT and Aza135 respectively (Supplementary Fig. 7). We hypothesize that the autotetraploidy event might be seen as the distance between the Ad77271a and Ad77271b siblings since they most likely represent distinct and random haplotypes, which would place the autotetraploidy event in the 4 MYA range with a similar timing of the last detectable TEs burst (Fig. 2b)." I think this is not

good enough and the authors should try harder to explain what these different peaks (obtained by Ks rates) refer to? For instance, it is not clear to me what e.g., the 17 my old peak could refer to then?

The authors state that “We hypothesize that the genes retained as four copies in the baobab genome may represent specific biology that was important to baobab.” There is a huge literature (none of it cited) about gene balance and gene dosage issues that explain the biased retention of genes of certain functional categories (transcription factors, developmental genes, genes encoding proteins acting in multiprotein complexes) that has not necessarily to do with ‘specific biology of importance to baobab’, especially not when WGDs are relatively recent, or very conserved (see further). All the classes of genes the authors mention are the genes typically found in excess in genomes following WGDs. This has been widely documented, not only for plants but appears to be a general phenomenon.

Related to that, the authors state that “In addition to the WGD 4 MYA, baobab also experienced a WGD 30 MYA that is shared across the Malvaceae; yet unlike other Malvaceae species studied to date, baobab retained almost all four copies resulting from the WGD. While most genomes fractionate gene copies back to a diploid state after WGD, it is thought that some genes may not be fractionated and retained to modify gene dosage in specific pathways important to the organism.” The authors do not seem to be aware that gene loss takes time, and is related to substitution rates, which are a function of generation times, metabolic rates, and population sizes. Therefore, simply assuming that the difference in gene loss or retention following WGDs between two species with significantly different life styles, is due to selection for genes of importance and can be linked to adaptation, is short-sighted, I think and could lead to wrong conclusions.

The authors do discuss some gene families in more detail, but most of it is very speculative. As a matter of fact, the paper is riddled with statements such as ‘we hypothesise’, ‘we speculate’, ‘these results could point’, ‘may provide insight’, ‘provides clues’, etc., which, I have to say, become a bit frustrating after a while.

The authors also conclude that the whole genome duplication unveiled in *A. digitata*, about 4 mya, coincided with an unprecedented amplification of DNA transposable elements (TEs) compared to Long Terminal Repeat Retrotransposons (LTR-RT). Again, this is very speculative given the way the authors have computed divergence times and estimates of the timing of WGDs. Also, why is the WGD inferred at 4 MYA consistent with an autotetraploidy event. Could this not have been an allopolyploidy event?

Other comments:

Page 4 line 29:

“We long read Oxford Nanopore Technologies (ONT) sequenced...” sounds awkward, I think.

Page 4, line 36:

The flow cytometry result shows that the genome size is around 920Mb, while the Kmer analysis result suggests a genome size of 749Mb. However, the final haploid genome size is estimated at around 675Mb, so how to explain the 70Mb size difference between the genome survey result and the final assembly? A 1.9% heterozygosity level (for *A. digitata*) is high, could this be part of the explanation (if the assembly is not completely phased)?

Page 5, lines 23~32:

This large translocation could hold significance for the baobab tree. Nevertheless, the absence of RPA1B does not correlate with the translocation of these genes. We cannot attribute the function of losing this gene to the function of gene translocation. They do not have a direct relationship.

Page 5 line 34:

The Kmer analysis result shown in Fig 1a looks strange. With a heterozygosity level of 1.9%, the result should show a peak indicating Kmers from heterozygous loci and a main peak. Why not just provide the result from GenomeScope?

Page 5 line 35:

Is Fig. s 1a,b the same as Fig. 1a;b? This causes confusion (throughout the manuscript).

Page 6 Table 2:

Some information in Table 2 is not clear. Does the N50 shown in the table refers to the Contig N50 or the Scaffold N50? Does the BUSCO assessment refers to the genome assembly or to (the completeness of) the annotation? The table would be better divided into three subsections, I think, namely genome assembly, repeat content, and annotation.

Page 7 line 25-28:

This information is already shown in Table 2, there is no need to reiterate it here.

Page 8 Fig 2:

Is it possible to use the same color palette for the same TE element?

Page 8 line 19:

Fig.s 3 -> Fig. 3?

Page 9 line 6:

Fig.s 3 -> Fig. 3?

Page 9 line 7:

The methylation part is very descriptive. It would be interesting to investigate whether the methylation links to any trait or adaptation of baobab.

Page 9 line 26:

It would be good to add another *Adansonia* (*Aza135*) in the gene family analysis to check whether it is also having these baobab specific orthogroups.

Page 10 line 12-16:

Cacao is not the sister group of baobab based on Fig. 4d. Is there something special in the gene families shared by baobab and cacao as the authors only list the result between these two species?

Page 12 line 9:

Fig.s 5 -> Fig. 5; Fig 5a is a syntenic dot plot, not syntenic depth plot.

In Supplementary Fig. 9, please highlight some syntenic regions.

Page 12 line 11:

Fig.s 5 -> Fig. 5

Page 12 line 13-14:

Should it be "amborella and grape..."? The ratio also does not seem correct, could the author please also provide the syntenic depth plot for those comparisons.

Page 12 line 16:

Fig.s 4 -> Fig.4

Page 14 line 1:

"as well 1 as 1 and 3 were less related"?

Page 19 line 10:

Could the authors kindly provide the parameters which they provided to Funannotate? What evidence are used for gene prediction?

Page 19 line 14:

Substitution rate varied a lot between species, Malvaceae and Salicaceae are not that closely related, using the substitution rate from *Populus* is probably not accurate enough for the calculation of the LTR insertion time.

In conclusion, although the paper discusses the genomes of an iconic species, I found the paper to be very descriptive and to be honest, seems to reveal very little about the evolution and/or adaptation of baobab, as nevertheless suggested by the title. The authors state that 'this research not only unravels baobab genomic evolution mysteries but also provides a crucial sequence for expediting gene discovery, enhancing breeding efforts, and aiding baobab species conservation.' I'm not sure what 'genomic evolution mysteries' the authors refer to or solved? The paper lists TEs, describes a WGD, there is some information on methylation, and some discussion on the genetic diversity in baobab, none of which provides, in my opinion, exciting novel discoveries that shed light on the evolution, diversification, or adaptation and specific lifestyle of baobab. Again, what is the significance of statements such as 'these results indicate there are geographical or environmental factors that have limited gene flow between these populations.'?!

Reviewer #2

(Remarks to the Author)

The manuscript presents a comprehensive genomic study of the African baobab tree, *Adansonia digitata*, a long-lived species with significant ecological and economic importance. The authors constructed a chromosome-level reference genome for *A. digitata* using long-read Oxford Nanopore sequencing and HiC data and draft assemblies for related species, uncovering unique genomic features. Unlike most plant genomes dominated by LTR retrotransposons, the baobab genome has a high proportion of DNA transposons.

The authors report that a whole-genome duplication event occurred 30 million years ago, shared with other members of the Malvaceae family. In addition, an autotetraploid event 4 million years ago that correlates with a burst of DNA transposon insertions was identified.

Moreover, the authors report that baobab retains multiple copies of circadian, light, and flowering-related genes, which may contribute to its longevity and unique pollination strategies. They resequenced 25 *A. digitata* trees across Africa, finding geographic variation that divided the species into three main subpopulations.

The baobab genome is undoubtedly fascinating. The genomic resources and insights offered by this study could form the basis for conservation efforts focused on preserving the diversity of the baobab. This research significantly enhances our understanding of the baobab genome and its evolutionary history. Perhaps clearer conclusions on the practical implications

for breeding and conservation strategies could be provided.

In addition, here are my detailed comments:

1. Baobab is autotetraploid. It went through a whole genome duplication and has $2n=4x=168$ chromosomes. The reported reference genome has 42. So not only the homologs are collapsed as typical for reference genome assemblies but also the homeologs. While I recognize the challenges in assembling sequences to individual chromosomes, I am concerned about the potential impact on the study's conclusions. 4 million years is a lot of time for mutations to happen, hence the sequences wouldn't necessarily collapse but lead to assembly artefacts and problems especially if the subgenomes don't recombine. Do they? If they don't, they must have evolved and changed in 4 million years. Collapsing them can lead to a lot of wrong conclusions. Was this considered when running the expansion/contraction analyses of gene clusters and copy number variation analyses? The expansion in copy number of specific genes mentioned in the manuscript can also be due to this. Also the translocations. These might be present in one subgenome and not in the other. Have you tried to separate the subgenomes with your long reads?
2. Are you sure about "buhibab"? It happens that I speak Arabic and I don't recognize the word at all. In standard Arabic, "fruit with many seeds" would typically be expressed as "فاكهة ذات بذور كثيرة" - fākihah dhāt bidhūr kathīrah or "فاكهة ذات حبوب كثيرة" - fākihah dhāt ḥubūb kathīrah, where "بذور" - bidhūr and "حبوب" - ḥubūb both mean seeds. You are citing a citation, that is citing this: <https://www.sciencedirect.com/science/article/pii/S0254629911001189#bb0160> and this: https://agritrop.cirad.fr/531802/1/document_531802.pdf and the first is citing the second. So it all comes down to Diop et al. 2005 which mentions that the word "baobab" originates from the Arabic term "bu hibab," which is claimed to mean "fruit with many seeds." However, the text notes that this origin is controversial, and various explorers and researchers have used different terms to describe the baobab tree and its fruit over time. The concept that "bu hibab" might mean "fruit with many seeds" could either be an adaptation or interpretation of the term in historical texts. Hibab sounds close to ḥubūb/ huboob but anyway, obviously this is not a solid fact. So please, if you don't mind and unless you can be sure about the origin, please rephrase your sentence "The word "baobab" comes from the Arabic name 'buhibab,' which means a fruit with many seeds1."
3. There is a translocation between the *A. digitata* species and Aza135 on Chr23 that moved ~120 genes including genes related to longevity closer to the telomere. The hypothesis is that telomere-associated proteins may be more active or better maintained near telomeric regions, I suppose? E.g. moving DNA repair genes closer to telomeres could facilitate more efficient repair of telomeric DNA and the maintenance of chromosome stability, right? Maybe you can make the hypothesis more clear. But then again this only affects one chromosome. And also is there is a difference in terms of longevity between Aza135 and the other species?
4. How can the variation in repeat content be explained? Does it correlate with the phylogeny?
5. By inter-homeolog recombination you mean that homeologs chromosomes recombine, is that correct? I can't find a description of that in the Lemna minor publication. Maybe I'm overlooking.
6. It is very interesting that there are more DNA than LTR transposons. However, don't you think it is possible that a lot of LTR elements collapsed in the ONT assembly? You can see that looking at depth of coverage and mapping rates when mapping the reads back to the genome.
7. Why do AdOHT and Aza135 have such a low BUSCO score?
8. Fig. 4c: did you find this enriched in the gene family expansions of baobab? Can you please provide the tables of enriched terms?
9. Did you only use two of the baobab genomes for Orthofinder? Why?
10. The cells in your suppl. table 7, column E are being formatted to dates from row 96 onwards.
11. Please add the URL of the Baobab database <https://resources.michael.salk.edu/baobab/index.html> to the manuscript. The link in the text is not working.
12. Page 13, line 28: (population 3; top left; green circles). You need to refer to the figure. Fig 6a, I assume.
13. "Genes in baobab were predicted via the Funannotate (v1.8.2) pipeline with modifications" What modifications? And were all the baobab genomes annotated like this?
14. Would it be possible to name the chromosomes chr1 instead of just "1"?

Reviewer #3

(Remarks to the Author)

I was happy to read this manuscript presenting very interesting research and very much needed if the domestication of

African baobab should be started. The introduction summarizes the state of our knowledge of African baobab but in the end I miss much more clearly explained problem statement and also the general and specific objectives should be expressed in better way. E.g. why we need to know the ploidy of *A. digitata*, and how it can help us in further domestication of this species. The results from this research should be of the interest of wider scientific community, however, sometimes I felt lost such a high number presented results in Table and six figures. Would it be possible to consolidate result explain them specially in much understandable manner for wider audience even without the deep experience in this specific topic. I just feel overwhelmed with such a high number of highly specific data results. However the implication of the results are very well enlighten in the discussion part.

Version 1:

Reviewer comments:

Reviewer #2

(Remarks to the Author)

The manuscript was revised thoroughly and all the points and questions I had were answered to my satisfaction. I only disagree that BUSCO scores of 94 and 93% are considered high nowadays. I do agree with the authors, though, that this is probably sufficient for the analyses they are conducting.

There is no further criticism from my side.

Reviewer #3

(Remarks to the Author)

My comments were successfully incorporated into the final version of the ms.

Reviewer #4

(Remarks to the Author)

The authors have made relatively good responses to the concerns raised by Referee #1, point by point. However, some results still look highly confusing which need more illustrations.

1. As shown in Responding Fig. 1, two WGD events were inferred in Ad77271a, one is ancient and the other is autotetraploidy which belongs to baobab lineage. If Ad77271a genome is a haploid (i.e. subgenome), it would be surprise to observe a syntenic depth ratio between Ad77271a and cacao as 4:1, comparing to a ratio between Ad77271a and grape as 4:2, as shown in Responding Fig. 2 & 10. Does the ancient "WGD" here represent an octoploidization event?

2. I also concern about the issue of gene loss as Referee 1# raised. The authors have not fixed it very well. Why is the original gene copy number four in baobabs? If there is only one ancient tetraploidization event, the original gene copy number in a haploid genome should be two but not four. The authors did not provide additional evidence for the claim about "most genomes fractionate gene copies back to a diploid state after WGD" in the main text or "some genes are lost" in the rebuttal. If the authors want to highlight the functional importance of "clock/flowering/temperature genes are retained in four copies", I would suggest that they should provide comparisons of gene copy with more species (e.g. cottons and *B. ceiba*). Recent studies suggested a decaploidization event in Malvaceae. If so, the syntenic depth and gene loss here need more discussions.

3. The authors claimed that "the haploid representation most likely contains a combination of non-phased variation from the four subgenomes" (I think they are indeed be "two subgenomes") in the rebuttal. If the genome is non-phased, it is extremely hard to assign the Ks rates to the speciation peaks. The authors' response is ok for a normal diploid genome. But it could be a different situation for a non-phased haploid genome of an autotetraploid species.

My essential argument is that the genome of the two siblings can't be simply treated as two subgenomes. The siblings only represent combinations of random chromosomes or contigs, and the recombination between homeologs can't be overlooked. The potential artefacts caused by autopolyploid can lead to complex result of assembly. Currently, the assembly of the autopolyploid baobab genome is still technologically treated as a normal one of a diploid organism.

Several conclusions from this manuscript remained highly controversial to me,

1. Line329. Are there any impacts of assembly artefacts on the number and distribution of TEs? Similar concerns are remained on the number of genes, gene expansion, and patterns of centromere and telomere.

2. Line335. Is it possible that the two peaks of TEs represent homologs (later) and homeologs (earlier) respectively?

3. Line436. It may not be a good idea to infer the ancient WGD history in Malvaceae by using of the autopolyploid genomes.

4. Line461. The timing of autotetraploidy event seems considerably underestimated given only parts of the sibling genomes are homeologs.

5. Line533. It is worth noting that *Fst* is a population genetic estimation based on the model for diploids. Polyploid *Fst* should be cautiously interpreted.

I do agree with that assembly of the "true subgenome" is of great challenges. However, the relevant conclusions in the manuscript are doubtful before new analyses specific to autopolyploid (not affected by polyploidy or homeologs) are introduced.

Besides, I am struggling about some conflicting interpretation of results here,

Basing on resequencing data, the author claimed that some genomic signals, such as gene dosage of some pathway and centromere regions are "highly dynamic and location specific" among populations within the baobab species. If so, the previous conclusions on these genomic signals by comparing the baobab genome with other species seem unreliable. For example, if the centromere regions are highly dynamic, is it possible that the conclusion on centromeres (Line274-288) is

just a description on one population or individual, instead of the species? Does the retention of gene copies in several pathways contribute to local adaptation of populations or the common adaptation of the species? Similarly, I found that the Gypsy number in Table 2 is highly various among samples, even between the two siblings. There are two possible reasons: one is due to the highly dynamics, and the other is assembly artefacts.

Other comments:

Line168. Why is the genome size of *A. grandidieri* and *A. rubrostipa* were much larger than the others given they are all diploid?

Line174. Why is the heterozygosity of *A. digitata* accurate to two decimal places, but others are accurate to one decimal place? Is the conclusion "one of the highest heterozygosity" still correct if the data of *A. perrieri* and *A. za* were accurate to two decimal places as *A. digitata*?

Line220-228. I'm not so confident about the comparison initiated by the lower-quality assembly of Aza135. Is there any phenotypic difference between *A. digitata* and Aza135, which can be explained by this translocation? The authors highlighted genes associated with DNA repair and longevity, isn't *A. za* a long-lived organism?

Line297. What is the meaning of the number 54.74%, 54.94% and 62.52% shown here? Are they significantly larger than the number in other organisms? I realized that the detection method is based on deep learning, so the comparison will be reasonable if the average levels of DNA methylation in other organisms can be detected in the same way.

Line386. "baobab has some unique gene content" sounds meaningless since each species have its own unique gene content.

Line435. Two literatures addressing the genomic evolution in Malvaceae have provided some new insights which deserved consideration since WGD is one of the most important issue dissected in this study.

<https://doi.org/10.1016/j.xplc.2024.100832>

<https://doi.org/10.1073/pnas.2313921121>

Line444. I realized that the divergence time here is not so consistent with the time estimated using the calibrated tree (Supplementary Fig. 10). Which one is more reliable?

Line450. *G. hirsutum* is an allotetraploid cotton with AADD genome. It is not a good idea to use it for rebuilding of the gene-based species tree.

Line488. Is it possible that the high number of gene copies is just due to assembly artefacts?

Line915. Supplementary Fig. 4b is same to Fig. 1d.

Version 2:

Reviewer comments:

Reviewer #4

(Remarks to the Author)

I think most of my concerns have been well addressed by the authors and I'm particularly happy to see that additional citations and discussions are added appropriately. This manuscript deserves to be published yet some remaining concerns need further illustrations.

1. Regarding the WGM history, it becomes much easier to understand the 4:1 ratio between baobab and cacao in Fig. 5 and Supplementary Fig. 8. But I am confused by the 4:2 ratio with grape and the 6:1 ratio with Amborella. I noticed that the authors have removed the related sentences from the main text. Please can the authors provide possible explanations on the ratios?

2. As to Response Table 1, baobab and *B. ceiba* share a quite similar pattern of copy number of the circadian, flowering, and light genes. I think this could be appropriately interpreted as the feature of subfamily Bombacoideae rather than specific to baobab lineage (Line 491).

3. The authors have clarified the relationship of the two sibling haploids. However, I am still not so confident about the conclusions basing on comparison between these two siblings, especially the estimated time of the autotetraploid event. I would prefer a clear definition about the Ks peak at 4 MYA between the two siblings, which may be neither a speciation peak nor an autotetraploidization peak. Despite the claims on dating of the autotetraploid event have been currently toned down, I do hope some words could be spend on highlighting the probable limitation or potential bias in the main text.

4. The authors' explanation on the large estimated genome size of *A. grandidieri* seems probable but not so reliable given the relatively low heterozygosity of *A. grandidieri*. Moreover, it is surprise to see that historical hybridization can lead to such a various genome size. Is it possibly attributed to a special sample, some unexpected methodological problems or some unique genomic features? I suggest that the present results of genome size of diploid baobabs here should be interpreted very cautiously.

5. In Supplementary Fig. 11, it is a big surprise to see such a "complex" collinearity between Ad77271a and Aza135.

Version 3:

Reviewer comments:

Reviewer #4

(Remarks to the Author)

Thanks for the positive responses to my concerns. I have no more issues with this manuscript and recommend the publication of it.

Reviewer #1 (Remarks to the Author):

This paper describes a haploid chromosome-level reference genome for baobab, *Adansonia digitata*, as well as draft assemblies for a sibling tree, two trees from distinct locations in Africa, and a related species from Madagascar. The paper describes a paleopolyploid origin for baobab, shared with the Malvoideae ~30 million, and a more recent autotetraploidy event 3-4 million mya that coincides with a recent burst of TE insertions. Resequencing of 25 *A. digitata* trees from Africa revealed three subpopulations that suggest gene flow through most of West Africa but separated from East Africa.

I have read this paper with great interest, amongst other things because the paper describes new, high-quality genomic data of a truly iconic species. However, I have to say that I was a little disappointed after reading the paper. Please allow me to explain why.

Response: Thank you for your valuable feedback on our manuscript. Your thoughtful evaluation and suggestions are greatly appreciated, and we are grateful for your support throughout this process. We are fully dedicated to addressing all the concerns you raised and have made revisions to enhance the quality of the manuscript.

For instance, I found the entire section on whole genome duplication (WGD) quite superficial and to contain some statements that are difficult to understand. For instance, the authors state that “The Ks revealed a consistent timing of the separation of baobab-amborella and baobab-grape at 128 and 96 MYA respectively (Fig. 5b). Both cacao and cotton diverged from the ancestor of *Adansonia* around 30 MYA, which was the first piece of evidence that a baobab-specific WGD occurred around this time.” First, the approach to determine the timing of WGDs and speciation events seems only based on estimated Ks values, which is concerning since different species can have very different substitution rates. Also, how are these Ks values calculated? Which software is used to get those Ks values? Which substitution rates are used for the inference of divergence time? There is no information on this unless I have missed it. Furthermore, one must be cautious to compare Ks peak values to determine which event occurred first, a speciation or a WGD event. In my opinion, if there is some concern, dot plot analysis and in particular gene tree – species tree reconciliation methods should be applied to unequivocally (and still then ...) place WGDs in time.

Response: Generally, we start WGD analyses with the amborella and grape comparisons to determine if the general WGD/WGT history is consistent using a specific substitution rate; in this case we found that the a substitution rate of 6.56×10^{-9} (Gaut et al., 1996) recapitulated the timing of baobab-amborella and baobab-grape as has been described by many papers as noted in the text. Of course, these are estimates on the order of millions of years and should be taken as directional just like dotplots and gene trees. All the genome comparisons were first visualized as dotplots, and syntenic depth was calculated. The syntenic depth plots as well as dotplots were added to the supplemental figures (Supplementary Fig. 8 and Supplementary Fig. 9). In addition, we have added time-calibrated phylogeny based on 1,296 single-copy orthologue genes to the revised manuscript (Supplementary Fig. 10; Response Fig. 1). The resulting timing slightly shifts the split of Ad77271a to 21 MYA from the previously estimated 17 MYA. However, the overall pattern remains consistent with the Ks analysis.

Response Fig. 1: Time-calibrated phylogenetic tree of *Theobroma cacao*, *Bombax ceiba*, *Gossypium hirsutum*, *Gossypium raimondii*, *Adansonia za* (Aza135), and *Adansonia digitata* (Ad77271a) based on 1,296 single-copy orthologous genes. Sky-blue horizontal bars indicate the 95% Highest Posterior Density (HPD) intervals of estimated divergence dates. Median node ages are shown beside the HPD bars, and timescale in million years ago (MYA) is provided at the bottom.

We have provided a more detailed explanation of the methodology used for calculating Ks values and inferences of divergence times in the revised manuscript. The supplemented “methods” section is as follows:

Page 21 lines 32~46:

Molecular dating and whole genome duplication (WGD): A total of 216 single-copy orthologues from *T. cacao*, *B. ceiba*, *G. hirsutum*, *G. raimondii*, *A.za* (Aza135), and *A. digitata* (Ad77271a), encompassing 1,296 single-copy orthologous genes, were analyzed. Protein sequences were aligned using MUSCLE (v.3.8.31) and converted to corresponding coding sequences (CDS) by substituting each amino acid with its triplet bases. The CDS for each single-copy gene family were concatenated to form a super gene for each species. The second nucleotide of each codon from the aligned sequences was extracted to construct a phylogenetic tree using BEAST (v2.7.6) under the HKY model with a Gamma category count of 4. The tree prior was set to the calibrated Yule model, with birth-rate parameters set to an Alpha (shape) parameter of 0.001 and a Beta (scale) parameter of 1000. Secondary calibration of molecular clocks was performed using priors from the <http://www.timetree.org/> database for the cotton split from bombacoid/malvoid (50 MYA)^{107,108}. For WGD, synonymous substitution rates (Ks values) were calculated using the MScan (Python version) with default parameters. A substitution rate of 6.56×10^{-9} was then utilized to estimate the time of duplication events¹⁰⁹

Reference:

Gaut, B. S., Morton, B. R., McCaig, B. C. & Clegg, M. T. Substitution rate comparisons between grasses and palms: synonymous rate differences at the nuclear gene *Adh* parallel rate differences at the plastid gene *rbcl*. *Proceedings of the National Academy of Sciences* 93, 10274–10279 (1996).

The authors state that “The self-alignments of genomes informed the timing of the most recent WGD in a species, which clarified that all of the baobab, bombax and cotton genomes shared the same WGD event at about 30 MYA, while cacao experienced its last WGD around 118 MYA that was consistent with it having only the WGT- γ (Fig. 5c).” What does the self-alignment of genomes mean? From the figure, I assume it refers to age distributions made from collections of paralogous genes (all genes, or only homeologs)? Finding similar Ks peak values for different organisms can indeed suggest a shared WGD, but it could also suggest independent WGDs in different branches at similar times in evolution.

Response: The “self-alignment of genomes” (Ad77271a vs. Ad77271a, etc.) refers to estimating the Ks based on paralog comparisons where paralogs result from past WGD events. We edited it to say “self-self alignments” with paralogs mentioned. We agree that they don’t have to share the same WGD, so we changed this wording. Based on the timing of the time calibrated tree, it implies independent WGDs. The text has been updated to make this clearer.

Second, the entire section could do with some more detailed explaining. “Both cacao and cotton diverged from the ancestor of Adansonia around 30 MYA, which was the first piece of evidence that a baobab-specific WGD occurred around this time”. How is this first evidence for a baobab-specific WGD? It seems problematic to infer a specific WGD event solely based on divergence time without further elucidation or reference to accompanying figures. Additionally, there’s a notable absence of a summarizing sentence indicating the presence of a WGD event shared with the Malvoideae but not with cacao. Instead of ambiguously stating “around ~30 Mya,” it would be more precise to assert that this WGD event occurred after the divergence with cacao but prior to the divergence within the ancestor of the Malvoideae.

Response: We have revised the text and asserted that the WGD event happened after the split from cacao, as shown in **Response Fig. 1**.

The authors state that “When we zoom into just the baobab complex, we saw that all genomes have a minor Ks peak around 30 MYA in addition to peaks at 4, 6, 11 and 17 MYA for Ad77271b, 1 AdKB, AdOHT and Aza135 respectively (Supplementary Fig. 7). We hypothesize that the autotetraploidy event might be seen as the distance between the Ad77271a and Ad77271b siblings since they most likely represent distinct and random haplotypes, which would place the autotetraploidy event in the 4 MYA range with a similar timing of the last detectable TEs burst (Fig. 2b).” I think this is not good enough and the authors should try harder to explain what these different peaks (obtained by Ks rates) refer to? For instance, it is not clear to me what e.g., the 17 my old peak could refer to then?

Response: We have expanded the explanation of Ks peak values to offer a clearer interpretation, particularly regarding peaks like the one at 17 MYA as a speciation event, being in the range of the Ad77271a-Aza135 split depicted by the time-calibrated tree approach (21 MYA). While we have noted the observed elevated Ks values, confirming the precise events (speciation/duplication) and their timing requires additional datasets, such as fossil records, which are beyond the scope of this paper. Nonetheless, we believe readers will appreciate the complexity of the baobab’s evolutionary journey and recognize the need for more targeted experiments to validate the hypotheses raised in this study.

The authors state that “We hypothesize that the genes retained as four copies in the baobab genome may represent specific biology that was important to baobab.” There is a huge literature

(none of it cited) about gene balance and gene dosage issues that explain the biased retention of genes of certain functional categories (transcription factors, developmental genes, genes encoding proteins acting in multiprotein complexes) that has not necessarily to do with 'specific biology of importance to baobab', especially not when WGDs are relatively recent, or very conserved (see further). All the classes of genes the authors mention are the genes typically found in excess in genomes following WGDs. This has been widely documented, not only for plants but appears to be a general phenomenon.

Related to that, the authors state that "In addition to the WGD 4 MYA, baobab also experienced a WGD 30 MYA that is shared across the Malvaceae; yet unlike other Malvaceae species studied to date, baobab retained almost all four copies resulting from the WGD. While most genomes fractionate gene copies back to a diploid state after WGD, it is thought that some genes may not be fractionated and retained to modify gene dosage in specific pathways important to the organism." The authors do not seem to be aware that gene loss takes time, and is related to substitution rates, which are a function of generation times, metabolic rates, and population sizes. Therefore, simply assuming that the difference in gene loss or retention following WGDs between two species with significantly different life styles, is due to selection for genes of importance and can be linked to adaptation, is short-sighted, I think and could lead to wrong conclusions.

Response: The haploid baobab genome is unique amongst plant genomes sequenced to date in that it has retained almost four full chromosomes after the WGD 30 MYA; if the base chromosome number is 10 based on cacao, then the 42 chromosomes represent mostly a retention and some splitting/fusions (**Response Fig. 2**). Soybean is similar and has retained almost all four copies after the last WGD 15 MYA. While these processes take time, based on the current high-quality genomes most fractionate back to two copies in the time frames analyzed. This assessment is based on an analysis of 123 high quality plant genomes (**Zhao et al., 2021**) that were analyzed for WGD, and we further analyzed to understand how often this happens in reference to the circadian, light and flowering time genes (**Michael 2021**). We do cite what we think is the most relevant background on fractionation and dosage (**Cheng et al., 2018**) and now we have added one other (**Ruju et al., 2020**). However, *A. digitata* does not retain all four copies suggesting that some genes are lost while the overall organization and structure of the chromosomes are retained. We agree that the language that these are "baobab specific" may be too strong so we have removed this wording. These concepts are consistent with current thinking in the genomics field regarding polyploidy and WGD (**Van de peer et al., 2017**). We also point out cases in the text where genes have been retained and play functions specific to a species such as in soybean and CAM plants; we highlight this rare event in Fig. 5d where syntenic copies of clock/flowering/temperature genes are retained in four copies. We are aware that transcription factors, developmental genes, and genes encoding proteins acting in multiprotein complexes are often retained after WGD and this is why we have highlighted gene classes that are distinct: chromatin, exocytosis, and flower timing/development; the latter category we further develop later.

Response Fig. 2: The *A. digitata* genome retained many almost full chromosomes (Chr) after whole genome duplication (WGD). The Ad77271a (blue, bottom) genome was aligned to the cacao (*Theobroma cacao*; red, top) genome at the protein level with MCscan (Python version with default parameters). The cacao genome was used since it clearly displays the retention of full Chr structure (length) with the Ad77271a genome (Fig. 5a main text), which also highlights the continuity and accuracy of the Ad77271 assembly. The syntenic regions were visualized using the `jcvi.graphics.karyotype` tool with the settings `--minspan=30`. The resulting anchor file was analyzed for the number of syntenic connections between cacao and Ad77271a Chrs, and the syntenic blocks greater than 10 or 300 genes were retained for plotting. Each cacao Chr was plotted separately to highlight the complete Ad77271a Chr, and colors were randomly assigned to syntenic blocks by Ad77271a Chr. Cacao Chr 1, 2, 3, 4, 6, and 9 had all four copies retained in the Ad77271a genome with different levels and areas of fractionation. Chr 1 and 4 were fused before the Ad77271a WGD resulting in the Chr 1 arms being split into 8 copies with connections to Chr4. A similar yet more complicated fusion/breaking occurred between Chr 5, 7, 8, and 10, which is why the four Chr retention pattern is less obvious in these plots compared to the dotplot (Fig. 5a main text).

References:

Cheng, F. *et al.* Gene retention, fractionation and subgenome differences in polyploid plants. *Nat Plants* **4**, 258–268 (2018).

Raju, S. K. K. Gene Dosage Balance Immediately following Whole-Genome Duplication in Arabidopsis. *The Plant cell* vol. 32 1344–1345 (2020).

Zhao T, Zwaenepoel A, Xue JY, Kao SM, Li Z, Schranz ME, Van de Peer Y. Whole- genome microsynteny-based phylogeny of angiosperms. *Nat Commun.* 2021 Jun 9;12(1):3498. doi: 10.1038/s41467-021-23665-0. PMID: 34108452; PMCID: PMC8190143.

Michael TP. Core circadian clock and light signaling genes brought into genetic linkage across the green lineage. *Plant Physiol.* 2022 Sep 28;190(2):1037-1056. doi: 10.1093/plphys/kiac276. PMID: 35674369; PMCID: PMC9516744.

Van de Peer Y, Mizrachi E, Marchal K. The evolutionary significance of polyploidy. *Nat Rev Genet.* 2017 Jul;18(7):411-424. doi: 10.1038/nrg.2017.26. Epub 2017 May 15. PMID: 28502977.

The authors do discuss some gene families in more detail, but most of it is very speculative. As a matter of fact, the paper is riddled with statements such as ‘we hypothesise’, ‘we speculate’, ‘these results could point’, ‘may provide insight’, ‘provides clues’, etc., which, I have to say, become a bit frustrating after a while.

Response: We understand your concerns about the speculative nature of some of our statements; however, this is a genome manuscript where many things cannot be tested readily. We have revised the manuscript providing more concrete statements where possible. Additionally, we have clearly distinguished between well-supported findings and hypotheses to enhance the clarity and robustness of our discussion. We would like to note that for many of these “speculative” findings, we provide several lines of evidence supporting them. Our main biological finding concerning circadian, chromatin and flowering genes were supported in both our WGD retention analysis, as well as the variation analyses across the 25 resequenced trees (population Fst and local PCA). We try not to overstate our findings, but we hope readers will be able to make their own conclusions based on the several lines of evidence we present. We have created a FigShare with all the raw data so researchers can check the veracity of our data and analyses.

The authors also conclude that the whole genome duplication unveiled in *A. digitata*, about 4 mya, coincided with an unprecedented amplification of DNA transposable elements (TEs) compared to Long Terminal Repeat Retrotransposons (LTR-RT). Again, this is very speculative given the way the authors have computed divergence times and estimates of the timing of WGDs. Also, why is the WGD inferred at 4 MYA consistent with an autotetraploidy event. Could this not have been an allopolyploidy event?

Response: We have conducted additional analyses using a time-calibrated phylogenetic tree as you suggested, which confirmed that the timing of speciation and WGD events are congruent (**Response Fig. 1**). Therefore, the event 4 MYA that separates the haplotypes (Ad77271a vs. Ad77271b) represents the polyploidy event. It is true that distinguishing between auto- and allo-

polyploidy can be difficult because the hybridization of closely related species can appear as an autotetraploid event. However, we do not detect any evidence for an allopolyploidy event, such as distinct centromere sequences (**Response Fig. 3**) like we have seen in homeologs of *Eragrostis tef* (VanBuren et al., 2020), or a high level of variation in the subgenomes (once again the comparison between Ad77271a vs. Ad77271b). Therefore, based on our genome assemblies and annotations, we can assert that the WGD in *A. digitata* is consistent with an autotetraploid rather than an allotetraploid event. We now provide a clearer rationale for our inferences, supported by additional data and analyses.

Response Fig. 3: The *A. digitata* (Ad77271a) centromere arrays are identical across all 42 chromosomes. The centromere base repeats in arrays greater than 10 kb were extracted and aligned using mafft (--reorder --adjustdirection --maxiterate 1000 --retree 1 --globalpair) and visualized with FigTree. Only arrays from chromosomes 6, 7, 9, and 20 (full chromosomes with

cacao) are highlighted with different colors to show that there is not a specific pattern by chromosome.

Reference:

VanBuren R, Man Wai C, Wang X, Pardo J, Yocca AE, Wang H, Chaluvadi SR, Han G, Bryant D, Edger PP, Messing J, Sorrells ME, Mockler TC, Bennetzen JL, Michael TP. Exceptional subgenome stability and functional divergence in the allotetraploid Ethiopian cereal teff. *Nat Commun.* 2020 Feb 14;11(1):884. doi: 10.1038/s41467-020-14724-z. PMID: 32060277; PMCID: PMC7021729.

Other comments:

Page 4 line 29:

“We long read Oxford Nanopore Technologies (ONT) sequenced...” sounds awkward, I think.

Response: Revised.

Page 4, line 36:

The flow cytometry result shows that the genome size is around 920Mb, while the Kmer analysis result suggests a genome size of 749 Mb. However, the final haploid genome size is estimated at around 675Mb, so how to explain the 70 Mb size difference between the genome survey result and the final assembly? A 1.9% heterozygosity level (for *A. digitata*) is high, could this be part of the explanation (if the assembly is not completely phased)?

Response: Genome sizes from either flow cytometry or k-mer frequency are both estimates and can vary based on several factors (Pflug et al., 2020). Flow cytometry can result in estimates that differ from the actual genome size due to the selection of cultivars, staining techniques or the controls used, while k-mer genome size estimates vary due to sequencing reads used (long vs short reads), quality of sequencing data, coverage of sequencing data and the k-mer size chosen. A 70 Mb difference between the assembled genome size and k-mer frequency estimate is in the margin of error; however, we updated Fig. 1 as suggested with the genome scope result and now the k-mer frequency estimate is 659 Mb (different estimation methods even provide slightly different results), which is directly in line with the assembled genome size. Also, the genome scope figure more clearly highlights the broad single peak that we see in some autopolyploids (discussed in the text); the broad peak also causes issues with accurately estimating genome size using k-mer frequency.

Some of the hardest regions of a genome to assemble are the centromere, rDNA and TEs (Michael and Jackson, 2013). In the first wave of plant genomes based on either Sanger or short read sequencing, the centromere, rDNA and TEs were often collapsed leading to an assembled genome size smaller than the actual. Now with long read sequencing we are now rarely finding that regions collapsed and in addition we are assembling full centromeres and TEs while rDNA arrays still are problematic (Naish et al., 2021). While the centromere arrays are large in our assemblies, there is always a chance that we have collapsed the rDNA array, which we quantify in the text.

We state in the text that we have only assembled the haploid genome of *A. digitata*. Therefore, the haploid representation most likely contains a combination of non-phased variation from the four subgenomes. While we have tried to separate the sub genomes, we were unsuccessful (we suspected due to the low level of SVs and SNPs between subgenomes), so we sequenced a sibling to (Ad77271b) to ascertain the level of heterozygosity. While the overall heterozygosity is estimated at 1.9%, which reflects the underlying four subgenomes (haplotypes) of the tetraploid, the comparison of the sibling assemblies Ad77271a and Ad77271b was 0.65%.

References:

Pflug JM, Holmes VR, Burrus C, Johnston JS, Maddison DR. Measuring Genome Sizes Using Read-Depth, k-mers, and Flow Cytometry: Methodological Comparisons in Beetles (Coleoptera). *G3 (Bethesda)*. 2020 Sep 2;10(9):3047-3060. doi: 10.1534/g3.120.401028. PMID: 32601059; PMCID: PMC7466995.

Michael and Jackson. The First 50 Plant Genomes. *The Plant Genome*. 2013 July 01; <https://doi.org/10.3835/plantgenome2013.03.0001in>.

Naish M, Alonge M, Wlodzimierz P, Tock AJ, Abramson BW, Schmücker A, Mandáková T, Jamge B, Lambing C, Kuo P, Yelina N, Hartwick N, Colt K, Smith LM, Ton J, Kakutani T, Martienssen RA, Schneeberger K, Lysak MA, Berger F, Bousios A, Michael TP, Schatz MC, Henderson IR. The genetic and epigenetic landscape of the Arabidopsis centromeres. *Science*. 2021 Nov 12;374(6569):eabi7489. doi: 10.1126/science.abi7489. Epub 2021 Nov 12. PMID: 34762468.

Page 5, lines 23~32:

This large translocation could hold significance for the baobab tree. Nevertheless, the absence of *RPA1B* does not correlate with the translocation of these genes. We cannot attribute the function of losing this gene to the function of gene translocation. They do not have a direct relationship.

Response: We don't say that *RPA1B* is lost. We say that *RPA1B* is part of the 1 Mb translocation along with other genes that have potentially relevant functions and gene ontology (GO) enrichment. We highlight *RPA1B* because it also overlaps with UV-B signaling that is important in baobab due to its unique copies of these genes in the genome (highlighted later in the text).

Page5 line 34:

The Kmer analysis result shown in Fig 1a looks strange. With a heterozygosity level of 1.9%, the result should show a peak indicating Kmers from heterozygous loci and a main peak. Why not just provide the result from GenomeScope?

Response: In the revised version, we have substituted the image with genome scope output as suggested (**Response Fig. 4**). As noted above, genome scope estimates the genome size to be similar in size to the assembled genome (659 vs. 674 Mb), and the heterozygosity is 1.63%, which is similar to our estimate based on read mapping (1.9%).

Response Fig. 4: Characteristics of baobab (*Adansonia digitata*) genome. a GenomScope estimation of *A. digitata* genome size using 19-mer sequence counts and ploidy set to 4, the K-mer frequency depicts a unimodal pattern suggesting a diploid homozygous genome with a size of 659 Mb.

Page5 line 35:

Is Fig. s 1a,b the same as Fig. 1a;b? This causes confusion (throughout the manuscript).

Response: Revised.

Page 6 Table 2:

Some information in Table 2 is not clear. Does the N50 shown in the table refers to the Contig N50 or the Scaffold N50? Does the BUSCO assessment refers to the genome assembly or to (the completeness of) the annotation? The table would be better divided into three subsections, I think, namely genome assembly, repeat content, and annotation.

Response: We have addressed your concerns in the revised version. The table is now divided into three subsections, as suggested.

Page 7 line 25-28:

This information is already shown in Table 2, there is no need to reiterate it here.

Response: We have removed the redundant information.

Page 8 Fig 2:

Is it possible to use the same color palette for the same TE element?

Response: We have implemented as suggested (Response Fig. 5).

Response Fig. 5: Insights into transposable elements (TEs) in baobab. **a** Barplot comparison of TEs classification and sizes in five baobab genomes: Ad77271a, Ad77271b, Aza135, AdKB, and AdOHT. Aza135 is highlighted in the center to emphasize TE divergence. Except for Aza135 (*A. za*), the other four genomes belong to *A. digitata*. Distinct colors in the legend denote TE classes. The TE proportion to genome size (Mb) and percentage are displayed on the left and right y-axes, respectively. **b** Density plots of intact TEs, displaying the distribution of different types of TEs. Mutator transposons showed a burst around 11 million years ago. **c** Accumulation of DNA mutators in the centromeric regions of the Ad77271a chromosome 3. Four mutator Terminal Inverted Repeat (TIR) transposons (TE_00000631: Mutator631, TE_00000927: Mutator927, TE_00000845: Mutator845, and TE_00000967: Mutator967) with a potential role in centromere creation or repositioning are displayed. **d** High solo to intact LTR ratio in Ad77271a chromosomes shows an aggressive purging mechanism following WGD, contributing to a smaller genome size (659 Mb).

Page 8 line 19:
Fig.s 3 -> Fig. 3?

Response: We have corrected the figure reference to "Fig. 3" in the revised version.

Page 9 line 6:
Fig.s 3 -> Fig. 3?

Response: Revised.

Page 9 line 7:
The methylation part is very descriptive. It would be interesting to investigate whether the methylation links to any trait or adaptation of baobab.

Response: Investigating the potential links between methylation and specific traits or adaptations of baobab is indeed an intriguing avenue for future research. We have expanded the analysis to include global methylation patterns in relation to genes and transposable elements. Our observations indicate the presence of methylation across the entire genome, with subtle elevations on DNA mutators or centromeric regions, such as on chromosome 11 (**Response Fig. 6**).

Response Fig. 6: Methylation patterns across *A. digitata* chromosomes. Global distribution of genome features in *A. digitata*. Centromere arrays are plotted as darkgreen bars at $y=0.8$. Genes are represented in blue, Mutator transposons in green, CACTA transposons in red, Gypsy retrotransposons in purple, Copia retrotransposons in orange, and CpG methylation in black. Transposable elements, such as Mutator transposons, tend to be located opposite to genes and in the centromeric regions. Methylation is present throughout the chromosome. The window positions are divided by the chromosome total length to create a relative scale between 0 and 1.

Page 9 line 26:

It would be good to add another *Adansonia* (*Aza135*) in the gene family analysis to check whether it is also having these baobab specific orthogroups.

Response: We have added *Aza135*, *AdOHT* and *AdKB* to the gene family analysis (**Response Fig. 7**) and discussed the gene ontology (GO) terms of baobab-specific orthogroups (**Response Fig. 8**).

Response Fig. 7: An UpSet plot showing orthogroups across 20 plant species: *A. digitata* (Ad77271a, Ad77271b, AdKB, AdOHT), *A. za* (Aza135), *A. calamus*, *A. gramineus*, *A. thaliana*, *A. trichopoda*, *B. ceiba*, *D. zibethinus*, *G. biloba*, *G. hirsutum*, *G. raimondii*, *K. fedtschenkoi*, *P. trichocarpa*, *Q. rubra*, *S. grande*, *T. cacao* and *V. vinifera*. The number of species-specific orthogroups and shared orthogroups is shown on the main bar y-axis; red bars correspond to the baobabs. The x-axis bars correspond to the number of orthogroups containing species.

Response Fig. 8: Subset of the 125 significant GO terms in 817 specific orthogroups unique to baobab (*Adansonia digitata*, including accessions Ad77271a, Ad77271b, AdKB, and AdOHT, and *Adansonia za* Aza135) compared to a diverse set of fifteen plant species: *Acorus calamus*, *Acorus gramineus*, *Arabidopsis thaliana*, *Amborella trichopoda*, *Bombax ceiba*, *Durio zibethinus*, *Ginkgo biloba*, *Gossypium hirsutum*, *Gossypium raimondii*, *Kalanchoe fedtschenkoi*, *Populus trichocarpa*, *Quercus rubra*, *Symphonia grande*, *Theobroma cacao*, and *Vitis vinifera*. The x-axis represents the gene ratio (gene count / 1178 genes). The y-axis lists the GO term names. Dot size indicates the study count, while dot color reflects the p-values, adjusted using the FDR BH method implemented in GOATOOLS v1.3.11. The color gradient corresponds to the range of p-values.

Cacao is not the sister group of baobab based on Fig. 4d. Is there something special in the gene families shared by baobab and cacao as the authors only list the result between these two species?

Response: We selected cacao as the comparison species due to its status as an outgroup of Malvaceae species with extensive evolutionary information available.

Page 12 line 9:

Fig.s 5 -> Fig. 5; Fig 5a is a syntenic dot plot, not syntenic depth plot.

Response: Revised.

In Supplementary Fig. 9, please highlight some syntenic regions.

Response: Implemented as suggested, thank you (**Response Fig. 9**).

Response Fig. 9: Comparative analysis of autotetraploid *A. digitata* and diploid *Theobroma cacao* genomes. Inner to outer tracks depict: a Syntenic genes, b GC content, c Gene density, and d Chromosome information. Prefixes 'Ad' and 'Tc' denote baobab and cacao respectively. The circos plot illustrates 42 pseudomolecules for baobab and 10 for cacao, with a window size of 100 kb. The red asterisk highlights the metacentric and acrocentric centromeres in baobab.

Page 12 line 11:

Fig.s 5 -> Fig. 5

Response: Revised.

Page 12 line 13-14:

Should it be “amborella and grape...”? The ratio also does not seem correct, could the author please also provide the syntenic depth plot for those comparisons.

Response: We have paraphrased the sentence for clarity and provided syntenic depth plots for the comparisons (**Response Fig. 10**).

Response Fig. 10: Syntenic depth ratios between Ad77271a and a Ad77271b b AdKB c AdOHT d Aza135 e cotton f bombax g cacao h grape and i amborella.

Page 12 line 16:
Fig.s 4 -> Fig.4

Response: Revised.

Page 14 line 1:
“as well 1 as 1 and 3 were less related”?

Response: We have revised the sentence to clarify that population 1 is as distantly related to population 2 as population 3 is to population 2.

Page 19 line 10:

Could the authors kindly provide the parameters which they provided to Funannotate? What evidence are used for gene prediction?

Response: The revised manuscript is as follows:

Repeats and gene prediction: EDTA (v1.9.6)⁹⁸ was used to construct a repeat library and softmask complex repeats. Tandem Repeats Finder (v4.09)⁹⁹ was employed to identify centromere and telomere sequences, as well as softmask simple repeats. The ONT cDNA library was mapped against the reference using Minimap2 with splice presets and then assembled using Stringtie (v2.2.1) with the long reads processing flag. Genes in baobab were predicted via the Funannotate (v1.8.2) pipeline with the Stringtie transcript models included as

evidence in addition to the uniprot protein set. Predicted proteins were characterized using EggNOG-mapper v2.0.1¹⁰⁰.

References:

Ou, S. et al. Author Correction: Benchmarking transposable element annotation methods for creation of a streamlined, comprehensive pipeline. *Genome Biol.* 23, 76 (2022).

Benson, G. Tandem repeats finder: a program to analyze DNA sequences. *Nucleic Acids Res.* 27, 573–580 (1999).

Cantalapiedra, C. P., Hernández-Plaza, A., Letunic, I., Bork, P. & Huerta-Cepas, J. eggNOG-mapper v2: Functional Annotation, Orthology Assignments, and Domain Prediction at the Metagenomic Scale. *Mol. Biol. Evol.* 38, 5825–5829 (2021).

Page 19 line 14:

Substitution rate varied a lot between species, Malvaceae and Salicaceae are not that closely related, using the substitution rate from *Populus* is probably not accurate enough for the calculation of the LTR insertion time.

Response: While we recognize that baobabs have distinct evolutionary histories, which may limit the direct applicability of substitution rates from other taxa. As noted above, we leveraged this substitution rate because it recapitulated the published timing of known events.

In conclusion, although the paper discusses the genomes of an iconic species, I found the paper to be very descriptive and to be honest, seems to reveal very little about the evolution and/or adaptation of baobab, as nevertheless suggested by the title. The authors state that ‘this research not only unravels baobab genomic evolution mysteries but also provides a crucial sequence for expediting gene discovery, enhancing breeding efforts, and aiding baobab species conservation.’ I’m not sure what ‘genomic evolution mysteries’ the authors refer to or solved? The paper lists TEs, describes a WGD, there is some information on methylation, and some discussion on the genetic diversity in baobab, none of which provides, in my opinion, exciting novel discoveries that shed light on the evolution, diversification, or adaptation and specific lifestyle of baobab. Again, what is the significance of statements such as ‘these results indicate there are geographical or environmental factors that have limited gene flow between these populations.’?!

Response: We have reevaluated our conclusions to ensure that they effectively highlight the novel insights gained from our research. Additionally, we have refined our language to better articulate the evolutionary implications of our findings, emphasizing their relevance to baobab diversification and adaptation. We appreciate your input and have taken steps to strengthen the manuscript accordingly.

Reviewer #2 (Remarks to the Author):

The manuscript presents a comprehensive genomic study of the African baobab tree, *Adansonia digitata*, a long-lived species with significant ecological and economic importance. The authors constructed a chromosome-level reference genome for *A. digitata* using long-read Oxford Nanopore sequencing and HiC data and draft assemblies for related species,

uncovering unique genomic features. Unlike most plant genomes dominated by LTR retrotransposons, the baobab genome has a high proportion of DNA transposons. The authors report that a whole-genome duplication event occurred 30 million years ago, shared with other members of the Malvaceae family. In addition, an autotetraploid event 4 million years ago that correlates with a burst of DNA transposon insertions was identified. Moreover, the authors report that baobab retains multiple copies of circadian, light, and flowering-related genes, which may contribute to its longevity and unique pollination strategies. They resequenced 25 *A. digitata* trees across Africa, finding geographic variation that divided the species into three main subpopulations.

Response: We appreciate your professional summary of our work.

The baobab genome is undoubtedly fascinating. The genomic resources and insights offered by this study could form the basis for conservation efforts focused on preserving the diversity of the baobab. This research significantly enhances our understanding of the baobab genome and its evolutionary history. Perhaps clearer conclusions on the practical implications for breeding and conservation strategies could be provided.

Response: We have revised the manuscript to include clearer significance of the study

Page 3 line 42 to page 4 line 17:

Here, we report genome size estimates for all eight recognized baobab species using a K-mer-based method from short-read genomic sequences. This method can provide independent estimates to those obtained previously using Feulgen staining and flow cytometry¹. Our primary objectives are to create a reference genome, evaluate genetic diversity, investigate key genomic features, and study evolutionary history. We generated a haploid chromosome-scale assembly of *A. digitata* (Ad77271a; originally from Tanzania) as well as long read draft assemblies of an Ad77271a sibling (Ad77271b), and offspring of the following trees; the Kord Bao Sudan (AdKB), the Okahao Heritage Tree from Namibia (AdOHT), and an additional species, *A. za* (Aza135) from Madagascar. Additionally, we resequenced 25 additional *A. digitata* trees representing different regions of Africa with short reads to assess genetic diversity. The findings from this work revealed: (a) a high proportion of DNA transposons compared to LTR retrotransposons, suggesting an unusual genomic structure; (b) specific set of DNA mutator transposons potentially contributing to new centromere formation; (c) retention of nearly complete chromosomes post-whole genome duplication (WGD), indicating a unique evolutionary path in genome maintenance; (d) a recent WGD event approximately 30 million years ago leading to gene expansion in flower development, chromatin regulation, and exocytosis; (e) expansion of UV RESISTANCE LOCUS 8 (UVR8) genes, which suggests a unique mechanism for genome protection in long-living trees; and (f) discovery of genetically distinct Namibian baobab populations, enhancing our understanding of baobab diversity and suggesting the need for targeted conservation initiatives. These insights will facilitate breeding, conservation and domestication efforts.

In addition, here are my detailed comments:

1. Baobab is autotetraploid. It went through a whole genome duplication and has $2n=4x=168$

chromosomes. The reported reference genome has 42. So not only the homologs are collapsed as typical for reference genome assemblies but also the homeologs. While I recognize the challenges in assembling sequences to individual chromosomes, I am concerned about the potential impact on the study's conclusions. 4 million years is a lot of time for mutations to happen, hence the sequences wouldn't necessarily collapse but lead to assembly artefacts and problems especially if the subgenomes don't recombine. Do they? If they don't, they must have evolved and changed in 4 million years. Collapsing them can lead to a lot of wrong conclusions. Was this considered when running the expansion/contraction analyses of gene clusters and copy number variation analyses? The expansion in copy number of specific genes mentioned in the manuscript can also be due to this. Also the translocations. These might be present in one subgenome and not in the other. Have you tried to separate the subgenomes with your long reads?

Response: We agree; we were hoping to resolve all four haplotypes of the baobab genome. As we note in the text, additional methods (that may or may not be possible in baobab) will be required to assemble the fully phased, haplotype resolved genome. For autotetraploid genomes with an even higher level of heterozygosity (both SV and SNPs/INDEL) such as the potato genome, a pollen approach was required. We did try to resolve the subgenomes with the long reads but were unsuccessful.

The k-mer frequency plot suggests that the sub genomes are very similar since we observe a single peak. As noted in the text, and discussed below, we also have observed this k-mer frequency profile in *Lemna* that has a low level of SV and SNPs/INDEL variation. In figure 1a we include the k-mer frequency plot that has only one peak, which is consistent with a homozygous diploid genome. This is a similar plot you would see for an inbreeding species like *Arabidopsis* which has very low heterozygosity and is diploid. Since our allele frequency analysis confirmed *A. digitata* is tetraploid, as was predicted by other methods, the k-mer frequency suggests that the subgenomes are very similar at the nucleotide level. As noted to reviewer #1, the heterozygosity between the subgenomes is on the order of 0.65%.

We conducted a coverage analysis (in response to DNA TE vs LTR query) and we observed consistent coverage across the genome and repeats suggesting there is not large-scale collapsing. In addition, across the four *A. digitata* genomes we find a high level of synteny, especially with the diploid *A. za* and cacao genomes, which suggests we do not have a high level of collapsing. All four *A. digitata* genomes displayed the same translocation event with *A. za* consistent with this being real and not an artifact of collapsing (it is unlikely all four genomes would collapse in the same way). The four *A. digitata* genomes share similar gene expansions/contractions consistent with these gene families being resolved correctly. All the measures that we have tried (coverage, BUSCO, k-mer, synteny, etc) suggest that the haploid representation is a high quality, non-collapsed version of the genome.

2. Are you sure about "buhibab"? It happens that I speak Arabic and I don't recognize the word at all. In standard Arabic, "fruit with many seeds" would typically be expressed as "فاكهة ذات بذور كثيرة" - *fākihah dhāt bidhūr kathīrah* or "فاكهة ذات حبوب كثيرة" - *fākihah dhāt ḥubūb kathīrah*, where "بذور" - *bidhūr* and "حبوب" - *ḥubūb* both mean seeds.

You are citing a citation, that is citing

this: <https://www.sciencedirect.com/science/article/pii/S0254629911001189#bb0160> and

this: https://agritrop.cirad.fr/531802/1/document_531802.pdf and the first is citing the second.

So it all comes down to Diop et al. 2005 which mentions that the word "baobab" originates from the Arabic term "bu hibab," which is claimed to mean "fruit with many seeds." However, the text

notes that this origin is controversial, and various explorers and researchers have used different terms to describe the baobab tree and its fruit over time. The concept that "bu hibab" might mean "fruit with many seeds" could either be an adaptation or interpretation of the term in historical texts.

Hibab sounds close to ḥubūb/ huboob but anyway, obviously this is not a solid fact. So please, if you don't mind and unless you can be sure about the origin, please rephrase your sentence "The word "baobab" comes from the Arabic name 'buhibab,' which means a fruit with many seeds1."

Response: We appreciate your perspective on this matter. Given the uncertainty surrounding the origin of the word "baobab," we have removed the specific Arabic term "buhibab" to avoid potential inaccuracies.

3. There is a translocation between the *A. digitata* species and Aza135 on Chr23 that moved ~120 genes including genes related to longevity closer to the telomere. The hypothesis is that telomere-associated proteins may be more active or better maintained near telomeric regions, I suppose? E.g. moving DNA repair genes closer to telomeres could facilitate more efficient repair of telomeric DNA and the maintenance of chromosome stability, right? Maybe you can make the hypothesis more clear. But then again this only affects one chromosome. And also is there is a difference in terms of longevity between Aza135 and the other species?

Response: In the section on the translocation, we are just highlighting that the 120 genes in this region share interesting biology (and GO enrichment) related to our finding of unique UV-B genes; see the comment to reviewer#1. It is an interesting hypothesis that moving genes closer to the telomere may play a role in maintenance or chromosome stability. Another hypothesis is that this translocation creates a species barrier between *A. digitata* and *A. za*, but this is probably too speculative to include as well.

4. How can the variation in repeat content be explained? Does it correlate with the phylogeny?

Response: The repeat variation is consistent across the *A. digitata* genomes (44-45%) even though we have seen that the repeat variation can vary substantially across accessions of the same species. The fact that *A. za* has an overall lower repeat content (35%) could reflect a different evolutionary trajectory of repeats (it only takes one LTR to copy and paste to cause genome expansion) or it could be the specific *A. za* tree we sequenced. We have too few genomes and species to draw general conclusions about repeat content and phylogeny. However, the consistency across the *A. digitata* genomes suggests repeat content could be conserved across different trees.

5. By inter-homeolog recombination you mean that homeologs chromosomes recombine, is that correct? I can't find a description of that in the Lemna minor publication. Maybe I'm overlooking.

Response: We have edited this section for clarity. We are trying to make the case that in polyploid genomes with low levels of structural variation (SV) and SNPs/INDELs, a single k-mer peak is sometimes observed.

6. It is very interesting that there are more DNA than LTR transposons. However, don't you think

it is possible that a lot of LTR elements collapsed in the ONT assembly? You can see that looking at depth of coverage and mapping rates when mapping the reads back to the genome.

Response: All five assemblies have more DNA TEs, and we don't see any bias in coverage suggesting the LTRs are not collapsed. Generally, with long read based assemblies we don't see the LTRs collapsing; generally the read length is greater than TEs allowing their proper resolution. However, we conducted a coverage analysis across the genome, as well as the three classes of LTR-RTs, and found that the coverage was consistent across genomes and LTRs, providing an additional data point that these regions are fully assembled. We tested this another way using the LTR Assembly Index (LAI) for the whole genome (calculated using https://github.com/oushujun/LTR_retriever) and found mean LAI was 9 with a standard deviation of 4.87 (sliding window size of 3 Mb with 300 Kb steps). This LAI score categorizes our assembly within the range of a "reference" assembly (Mokhtar et al. 2023).

Reference:

Mokhtar MM, Abd-Elhalim HM, El Allali A. A large-scale assessment of the quality of plant genome assemblies using the LTR assembly index. *AoB Plants*. 2023 Apr 4;15(3):plad015. doi: 10.1093/aobpla/plad015. PMID: 37197714; PMCID: PMC10184434.

7. Why do AdOHT and Aza135 have such a low BUSCO score?

Response: BUSCO scores of 93 and 94% are still quite respectable and appropriate for what we use them for in this manuscript. These BUSCO scores are lower than Ad77271a, Ad77271b and AdKB, which are almost complete (C: 98%), but the AdOHT and Aza135 BUSCO scores are still quite good.

8. Fig. 4c: did you find this enriched in the gene family expansions of baobab? Can you please provide the tables of enriched terms?

Response: Yes, we found genes shown in Fig. 4c enriched and we have included a supplementary table (13 and 14) in revised version of the manuscript

9. Did you only use two of the baobab genomes for Orthofinder? Why?

Response: In the revised manuscript, we show all five genomes for which we have long reads (**Response Fig. 5**). However, for the identification of baobab single-copy genes and the additional time-calibrated phylogenetic tree analysis (**Response Fig. 1**), we focused specifically on the reference genome (Ad77271a and the sibling genome). This decision was made to ensure consistency and accuracy in our comparative genomic analyses, as the reference genome provides the most comprehensive and well-annotated dataset for these specific analyses.

Response Fig. 7: An UpSet plot showing orthogroups across 20 plant species: *A. digitata* (Ad77271a, Ad77271b, AdKB, AdOHT), *A. za* (Aza135), *A. calamus*, *A. gramineus*, *A. thaliana*, *A. trichopoda*, *B. ceiba*, *D. zibethinus*, *G. biloba*, *G. hirsutum*, *G. raimondii*, *K. fedtschenkoii*, *P. trichocarpa*, *Q. rubra*, *S. grande*, *T. cacao*, *V. vinifera*. The number of species-specific orthogroups and shared orthogroups is shown on the main bar y-axis; red bars correspond to the baobabs. The x-axis bars correspond to the number of orthogroups containing species.

10. The cells in your suppl. table 7, column E are being formatted to dates from row 96 onwards.

Response: Revised, thanks.

11. Please add the URL of the Baobab database <https://resources.michael.salk.edu/baobab/index.html> to the manuscript. The link in the text is not working.

Response: We have updated it and ensured it works properly (**Response Fig. 11**).

Baobab Genome Database

Response Fig. 11: Baobab genome portal: Access the baobab genome portal at <https://resources.michael.salk.edu/baobab/index.html> for comprehensive information and resources on baobab genomes.

12. Page 13, line 28: (population 3; top left; green circles). You need to refer to the figure. Fig 6a, I assume.

Response: Yes, you are correct. The sentence now reads: "population 3; top left; green circles (Fig. 6a)." We have made the necessary corrections.

13. "Genes in baobab were predicted via the Funannotate (v1.8.2) pipeline with modifications" What modifications? And were all the baobab genomes annotated like this?

Response: We have added the specific annotation parameters used in the Funannotate pipeline in the revised version of the manuscript; see the comment to reviewer#1. Yes, all the baobab genomes were annotated using the same pipeline to ensure consistency across our analyses.

14. Would it be possible to name the chromosomes chr1 instead of just "1"?

Response: We have updated the manuscript to name the chromosomes as "chr1" instead of just "1."

Reviewer #3 (Remarks to the Author):

I was happy to read this manuscript presenting very interesting research and very much needed if the domestication of African baobab should be started. The introduction summarizes the state of our knowledge of African baobab but in the end, I miss much more clearly explained problem statement and also the general and specific objectives should be expressed in better way. E.g. why we need to know the ploidy of *A. digitata*, and how it can help us in further domestication of this species.

Response: Thank you for your suggestions and encouraging feedback. We have revised the introduction to more clearly articulate the problem statement and explicitly state the general and specific objectives of the study, including the importance of understanding the ploidy of *A. digitata*, and its implications for domestication efforts.

The supplemented words in the "Introduction" section are as follows:

Page 3 lines 1~17

Baobabs are some of the oldest and largest non-clonal living organisms on Earth with trees that can live over 2,400 years with canopy sizes of greater than 500 m³ and trunks reaching diameters of up to 10.8 meters (35 feet)⁸. However, baobabs are unlike most large and long-lived trees; they are succulents characterized by parenchyma-rich tissues that efficiently store

water and therefore, do not form “growth rings” or true wood ⁹. Achieving maturity in the wild presents a considerable challenge for baobab trees since seedlings face predation from caterpillars, goats, and cattle ¹⁰. Despite having bisexual flowers, baobabs are mostly self-incompatible, depending on external pollinators for successful fertilization ^{11,12}. In natural populations, *A. digitata* is primarily pollinated by bats ^{13,14}, with occasional visits by bushbabies ¹⁵, and hawkmoths in southern Africa ¹². Since *A. digitata* is an obligate outcrosser, the populations harbor a high level of diversity, which is observed as heterozygosity in the genome (Fig. 1a; Table 1) ^{12,16}. Understanding the ploidy of *A. digitata* is essential for breeding and genetic improvement programs, as polyploidy affects traits like growth rate and environmental adaptability. This understanding can help overcome the species' slow maturation period of 8 to 23 years and aid in its domestication. Additionally, measuring genome size across different baobab species provides valuable evolutionary insights and aids in designing genomic sequencing projects ^{17,18,19}.

Page 3 line 42 to Page 4 line 17: (see the comment to reviewer#2)

The results from this research should be of the interest of wider scientific community, however, sometimes I felt lost such a high number presented results in Table and six figures. Would it be possible to consolidate result explain them specially in much understandable manner for wider audience even without the deep experience in this specific topic. I just feel overwhelmed with such a high number of highly specific data results. However the implication of the results are very well enlighten in the discussion part

We have revised the results to make them clearer and more accessible to a wider audience, while still including detailed information for experts. Some of the novel discoveries are illustrated in graphical format (**Response Fig. 5**).

Response Fig. 5: Insights into transposable elements (TEs) in baobab. **a** Barplot comparison of TEs classification and sizes in five baobab genomes: Ad77271a, Ad77271b, Aza135, AdKB, and AdOHT. Aza135 is highlighted in the center to emphasize TE divergence. Except for Aza135 (*A. za*), the other four genomes belong to *A. digitata*. Distinct colors in the legend denote TE classes. The TE proportion to genome size (Mb) and percentage are displayed on the left and right y-axes, respectively. **b** Density plots of intact TEs, displaying the distribution of different types of TEs. Mutator transposons showed a burst around 11 million years ago. **c** Accumulation of DNA mutators in the centromeric regions of the Ad77271a chromosome 3. Four mutator Terminal Inverted Repeat (TIR) transposons (TE_00000631: Mutator631, TE_00000927: Mutator927, TE_00000845: Mutator845, and TE_00000967: Mutator967) with a potential role in centromere creation or repositioning are displayed. **d** High solo to intact LTR ratio in Ad77271a chromosomes shows an aggressive purging mechanism following WGD, contributing to a smaller genome size (659 Mb).

Reference:

Klein, S. J. & O'Neill, R. J. Transposable elements: genome innovation, chromosome diversity, and centromere conflict. *Chromosome Res.* **26**, 5–23 (2018).

#####

#####

Reviewer #4 (Remarks to the Author):

The authors have made relatively good responses to the concerns raised by Referee #1, point by point. However, some results still look highly confusing which need more illustrations.

1. As shown in Responding Fig. 1, two WGD events were inferred in Ad77271a, one is ancient and the other is autotetraploidy which belongs to baobab lineage. If Ad77271a genome is a haploid (i.e. subgenome), it would be surprise to observe a syntenic depth ratio between Ad77271a and cacao as 4:1, comparing to a ratio between Ad77271a and grape as 4:2, as shown in Responding Fig. 2 & 10. Does the ancient “WGD” here represent an octoploidization event?

Response: Thank you for your feedback and suggestions. We addressed each below and hope we have clarified issues and provided additional support when needed. Particularly the papers you mentioned were helpful: Shao et al., 2024 (<https://doi.org/10.1016/j.xplc.2024.100832>) and Sun et al., 2024 (<https://doi.org/10.1073/pnas.2313921121>), which were published just before and after the submission of this manuscript respectively. Shao et al., provides a higher quality genome for bombax that corroborates the dating for the whole genome duplication (WGD) shared between cotton and baobab, and they refer to this event as a WGD in the text and a WGP (whole genome polyploidy) in the figures. Sun et al. 2024, further show that the WGD shared between baobab and cotton is a decaployploid event resulting from successive allotetraploid and allohexaploid events that occurred close together. The Paterson group postulated that cotton experienced a decaployploid event (Paterson et al., 2012; Wang et al., 2016), although since we were just seeing a 4:1 ratio with cacao, we chose to call this event more generically a WGD event similar to Shao et al., 2024. The higher quality bombax (Shao et al., 2024) and balsa (Sun et al., 2024) show a clear 5:1 syntenic depth with cacao, compared to the 4:1 syntenic depth we observe between baobab and cacao; although 6% have a syntenic depth of 5 consistent with a loss of the 5th copy in baobab compared to balsa and bombax (Supplementary Fig. 8g). We have updated the results to include references to these publications and we have changed “WGD” to “WGM” (whole-genome multiplication), which is the language used by Sun et al 2024.; we have also explained the syntenic depth a bit more including that baobab has a 5:1 syntenic depth if you include the 6% in five copies. In addition, we have updated the discussion with a brief explanation of the WGM history. **Ln. 609 to Ln. 624:** “Baobab belongs to the Malvaceae family, which comprises 4,225 species across 243 genera and nine subfamilies, including economically and ecologically important plants such as cotton (*Gossypium spp.*), the cotton tree (*Bombax ceiba*), cacao (*Theobroma cacao*), and baobab (*Adansonia digitata*) (Sun et al.,2024). WGM have played a crucial role in the evolution of these plants. For example, following the ancient core eudicot whole-genome triplication (γ) event, cotton plants have undergone repeated polyploidization and hybridization, resulting in polyploid offspring with combined genomes that diverge significantly from their diploid parents, thereby initiating new evolutionary trajectories. The decaploid ancestor of cotton (a 5x multiplication) significantly contributed to the development of spinnable fibers (Sun et al.,2024, Wan et al.,2024, Paterson et al.,2012). Similarly, polyploidy in *Bombax ceiba* has facilitated its robust growth and vibrant blooms in tropical regions (Shao et al.,2024), while ancient WGM in cacao have led to the emergence of diverse and flavorful varieties. Here, we demonstrate that baobab has a unique proportion of DNA transposons, experienced WGM and present a high-quality chromosome-level haploid assembly of *Adansonia digitata* (Ad77271a) alongside draft genomes from a sibling tree (Ad77271b), two geographically diverse *A. digitata* (AdOHT and AdKB) and distinct species *A. za* (Aza135) from Madagascar. We also show a distinct baobab population in Namibia.”

References:

Shao L, Jin S, Chen J, Yang G, Fan R, et al. High-quality genomes of *Bombax ceiba* and *Ceiba pentandra* provide insights into the evolution of Malvaceae species and differences in their natural fiber development. *Plant Commun.* 2024 May 13;5(5):100832. doi: 10.1016/j.xplc.2024.100832. Epub 2024 Feb 5. PMID: 38321741; PMCID: PMC11121743.

Sun, P. et al. Subgenome-aware analyses reveal the genomic consequences of ancient allopolyploid hybridizations throughout the cotton family. *Proc. Natl. Acad. Sci. U. S. A.* 121, e2313921121 (2024).

Wang X, Guo H, Wang J, Lei T, Liu T, Wang Z, Li Y, Lee TH, Li J, Tang H, Jin D, Paterson AH. Comparative genomic de-convolution of the cotton genome revealed a decaploid ancestor and widespread chromosomal fractionation. *New Phytol.* 2016 Feb;209(3):1252-63. doi: 10.1111/nph.13689. Epub 2015 Oct 7. PMID: 26756535.

Paterson AH, Wendel JF, Gundlach H, Guo H, Jenkins J, Jin D, Llewellyn D, Showmaker KC, Shu S, Udall J, Yoo MJ, Byers R, Chen W, Doron-Faigenboim A, Duke MV, Gong L, Grimwood J, Grover C, Grupp K, Hu G, Lee TH, Li J, Lin L, Liu T, Marler BS, Page JT, Roberts AW, Romanel E, Sanders WS, Szadkowski E, Tan X, Tang H, Xu C, Wang J, Wang Z, Zhang D, Zhang L, Ashrafi H, Bedon F, Bowers JE, Brubaker CL, Chee PW, Das S, Gingle AR, Haigler CH, Harker D, Hoffmann LV, Hovav R, Jones DC, Lemke C, Mansoor S, ur Rahman M, Rainville LN, Rambani A, Reddy UK, Rong JK, Saranga Y, Scheffler BE, Scheffler JA, Stelly DM, Triplett BA, Van Deynze A, Vaslin MF, Waghmare VN, Walford SA, Wright RJ, Zaki EA, Zhang T, Dennis ES, Mayer KF, Peterson DG, Rokhsar DS, Wang X, Schmutz J. Repeated polyploidization of *Gossypium* genomes and the evolution of spinnable cotton fibres. *Nature.* 2012 Dec 20;492(7429):423-7. doi: 10.1038/nature11798. PMID: 23257886.

2. I also concern about the issue of gene loss as Referee 1# raised. The authors have not fixed it very well. Why is the original gene copy number four in baobabs? If there is only one ancient tetraploidization event, the original gene copy number in a haploid genome should be two but not four. The authors did not provide additional evidence for the claim about “most genomes fractionate gene copies back to a diploid state after WGD” in the main text or “some genes are lost” in the rebuttal. If the authors want to highlight the functional importance of “clock/flowering/temperature genes are retained in four copies”, I would suggest that they should provide comparisons of gene copy with more species (e.g. cottons and *B. ceiba*). Recent studies suggested a decaploidization event in Malvaceae. If so, the syntenic depth and gene loss here need more discussions.

Response: We have changed WGD to WGM in the text to reflect the language used in Sun et al., and added text that this WGM is a past decaploidization. We agree that it would strengthen our argument to show a comparison of gene copies for the circadian/flowering/temperature genes so we have added a table to the supplement (Response Table 1) and added sentences to the text. **Ln 627 to Ln 631:** “Baobab has undergone several round WGM starting with the WGT-γ followed by what is thought to be a decaploidization (Sun et al., 2024, wang et. al. 2016, Paterson et al., 2012 & Shao et al., 2024,), the latter of which has remained largely unfractionated with between four and five full chromosomes retained as compared to cacao (Fig. 5a; Supplementary Fig. 8g; Supplementary Fig. 12).” Also **Ln 461 to Ln 476**

	OG	Ad77271a	B. ceiba	G. raimondii	A. thaliana	V. vinifera	Q. rubra	gene	type
1	OG0002676	3	3	3	2	1	1	CCA1/LHY	clock
2	OG0002990	4	2	2	4	1	2	RVE1/RVE2/RVE7	clock
3	OG0000823	5	6	4	3	2	2	RVE3/RVE5/RVE6	clock
4	OG0008057	3	3	1	2	1	1	RVE4/RVE8	clock
5	OG0006813	4	4	1	2	1	1	PRR9	clock
6	OG0000842	7	6	4	2	3	3	PRR3/7	clock
7	OG0007965	1	1	3	1	1	1	PRR1	clock
8	OG0005091	3	2	3	1	1	1	PRR5	clock
9	OG0004177	4	4	2	2	2	1	ZTL/LKP2	clock
10	OG0008873	2	2	2	1	1	1	FKF1	clock
11	OG0001524	4	4	5	2	2	2	ELF3	clock
12	OG0002940	2	3	5	2	2	3	ELF4	clock
13	OG0002858	3	4	3	2	1	1	LUX	clock
14	OG0001908	4	6	3	1	1	1	LNK1	clock
15	OG0003471	3	4	1	1	1	1	LNK2	clock
16	OG0003692	2	3	2	1	1	1	GI	clock
17	OG0005135	3	3	1	1	1	1	FCA	flowering
18	OG0002762	8	3	2	3	1	1	VIN3/VEL1/VEL	flowering
19	OG0003516	2	4	3	2	2	2	FVE	flowering
20	OG0008018	2	3	1	1	1	1	TFL2	flowering
21	OG0005608	3	7	1	3	1	2	CO	flowering
22	OG0009853	2	1	1	1	1	1	LD	flowering
23	OG0007488	2	1	2	1	1	2	FRI	flowering
24	OG0000318	6	8	6	5	3	4	FT	flowering
25	OG0009131	2	1	1	1	1	1	DET1	light
26	OG0009581	1	1	2	1	1	1	PHYA	light
27	OG0003293	4	3	2	3	2	2	PHYB/PHYD/PHYE	light
28	OG0010520	1	1	1	1	1	1	PHYC	light
29	OG0003899	2	4	2	1	2	3	COP1	light
30	OG0008582	1	2	2	1	1	1	CRY1	light
31	OG0004498	1	2	2	1	1	1	CRY2	light
32	OG0002580	4	4	2	2	2	2	SPA1	light
33	OG0002519	3	3	1	2	1	1	HY5/HYH	light
34	OG0003669	4	3	1	1	2	1	PIF3	light
35	OG0007706	1	2	1	2	1	1	PIF4/PIL6	light
	TOTAL	106	113	78	62	48	52		

Response Table 1. Circadian, flowering, and light ortholog copy numbers.

References:

Sun, P. et al. Subgenome-aware analyses reveal the genomic consequences of ancient allopolyploid hybridizations throughout the cotton family. Proc. Natl. Acad. Sci. U. S. A. 121, e2313921121 (2024).

Wang X, Guo H, Wang J, Lei T, Liu T, Wang Z, Li Y, Lee TH, Li J, Tang H, Jin D, Paterson AH. Comparative genomic de-convolution of the cotton genome revealed a decaploid ancestor and widespread chromosomal fractionation. New Phytol. 2016 Feb;209(3):1252-63. doi: 10.1111/nph.13689. Epub 2015 Oct 7. PMID: 26756535.

Paterson AH, Wendel JF, Gundlach H, Guo H, Jenkins J, Jin D, Llewellyn D, Showmaker KC, Shu S, Udall J, Yoo MJ, Byers R, Chen W, Doron-Faigenboim A, Duke MV, Gong L, Grimwood J, Grover C, Grupp K, Hu G, Lee TH, Li J, Lin L, Liu T, Marler BS, Page JT, Roberts AW, Romanel E, Sanders WS, Szadkowski E, Tan X, Tang H, Xu C, Wang J, Wang Z, Zhang D, Zhang L, Ashrafi H, Bedon F, Bowers JE, Brubaker CL, Chee PW, Das S, Gingle AR, Haigler CH, Harker D, Hoffmann LV, Hovav R, Jones DC, Lemke C, Mansoor S, ur Rahman M, Rainville LN, Rambani A, Reddy UK, Rong JK, Saranga Y, Scheffler BE, Scheffler JA, Stelly DM, Triplett BA, Van Deynze A, Vaslin MF, Waghmare VN, Walford SA, Wright RJ, Zaki EA, Zhang T, Dennis ES, Mayer KF, Peterson DG, Rokhsar DS, Wang X, Schmutz J. Repeated polyploidization of Gossypium genomes and the evolution of spinnable cotton fibres. Nature. 2012 Dec 20;492(7429):423-7. doi: 10.1038/nature11798. PMID: 23257886.

Shao L, Jin S, Chen J, Yang G, Fan R, et al. High-quality genomes of *Bombax ceiba* and *Ceiba pentandra* provide insights into the evolution of Malvaceae species and differences in their natural fiber development. *Plant Commun.* 2024 May 13;5(5):100832. doi: 10.1016/j.xplc.2024.100832. Epub 2024 Feb 5. PMID: 38321741; PMCID: PMC1112174

3. The authors claimed that “the haploid representation most likely contains a combination of non-phased variation from the four subgenomes” (I think they are indeed be “two subgenomes”) in the rebuttal. If the genome is non-phased, it is extremely hard to assign the Ks rates to the speciation peaks. The authors’ response is ok for a normal diploid genome. But it could be a different situation for a non-phased haploid genome of an autotetraploid species.

My essential argument is that the genome of the two siblings can’t be simply treated as two subgenomes. The siblings only represent combinations of random chromosomes or contigs, and the recombination between homeologs can’t be overlooked. The potential artefacts caused by autopolyploid can lead to complex result of assembly. Currently, the assembly of the autopolyploid baobab genome is still technologically treated as a normal one of a diploid organism.

Response: A haploid representation of an autotetraploid is one copy of the four underlying genomes. Even if we had resolved the genome into four haplotypes, they still would not have been phased since we do not have parental information; the best case is that we would have them phased by chromosome, but each chromosome set would represent unphased genomes. To date, almost all diploid genomes published are non-phased haploid representations in plants; only in the past several years are we seeing haplotyped genomes and even fewer fully phased genomes. The unphased haploid genomes have been used to assign Ks rates for speciation peaks. Using the un-phased haploid Ad77271a assembly we accurately recapitulated the baobab-amborella and baobab-grape peaks, suggesting that this method does work on unphased haploid genomes as has been seen previously. It is true we cannot treat the sibling as another haplotype, but the sibling is a powerful tool to estimate the differences in alleles. In a later comment we show that only 14% of orthologs between the siblings are the “same,” (or have zero variation) suggesting that we do have decent representation of allele diversity between these two siblings. We further address the potential for artifacts in another response for which we still have been unable to identify any glaring artifacts. It should be noted that the single K-mer peak for *A. digitata* suggests that the underlying genomes of baobab are not that complex with a haploid genome heterozygosity below 1% (0.65%). We additionally address the use of the haploid representation in another comment below on Fst.

Several conclusions from this manuscript remained highly controversial to me,

1. Line329. Are there any impacts of assembly artefacts on the number and distribution of TEs? Similar concerns are remained on the number of genes, gene expansion, and patterns of centromere and telomere.

Response: Please see our response to reviewer #2 in our previous response: “All the measures that we have tried (coverage, BUSCO, k-mer, synteny, etc) suggest that the haploid representation is a high quality, non-collapsed version of the genome.” To summarize, coverage analysis suggests the assembly doesn’t have issues across different features such as repeats and genes. BUSCO scores suggest the gene space is complete so we are not missing genes and the duplication rate is consistent with a haploid genome that doesn’t have unresolved haplotypes integrated. The centromeres are on the megabase scale suggesting they are complete or near complete; but we should point out very few genomes to date are checked for centromere completeness. The telomeres assembled well and are long. However, the best

piece of evidence supporting a high quality assembly is the collinearity with all the genomes we present in the manuscript. The key is the synteny and orthologs with cacao; 93% of Ad77271a genes are collinear/syntenic with cacao (Figure 5a; Supplementary Fig. 8). We did not highlight it in our previous response but the HiC contact map (Fig. 1c) also provides high quality support for the genome structure. We cannot find any evidence of assembly artifacts including misassemblies, gene expansions/contractions, TE expansions/contractions, nor issues with the centromeres/telomeres. Of course no genome assembly is without some type of artifacts (until we can generate full chromosomes with single molecule reads), but based on standard measure the Ad77271a haploid genome looks good.

2. Line335. Is it possible that the two peaks of TEs represent homologs (later) and homeologs (earlier) respectively?

Response: The earlier striking peak around 11 million years ago (MYA) could represent homeologs resulting from a WGM event and corresponds to the interval of autotetraploidization in *A. digitata*. The later peak, on the other hand, might represent remnants of TEs, with only specific elements like CACTA and Copia experiencing bursts of activity. This peak could also be related to more recent duplications, which suggests that homologous genes have been created through segmental or tandem duplications. These more recent WGM might not be as widespread as those from the ancient WGM event, but they still contribute to the genomic landscape observed today.

3. Line436. It may not be a good idea to infer the ancient WGD history in Malvaceae by using of the autopolyploid genomes.

Response: We did not use autopolyploid genomes for WGD history inference. Our comparative genomics study and inference of WGD were conducted on baobab genomes alongside Malvaceae species such as cotton (*Gossypium raimondii*), bombax (*Bombax ceiba*), and cacao (*Theobroma cacao*). Additionally, we included grape (*Vitis vinifera*), which has undergone only core eudicot whole genome triplication (WGT), and Amborella (*Amborella trichopoda*), which is sister to the eudicot lineage and lacks a whole genome duplication (WGD) event.

4. Line461. The timing of autotetraploidy event seems considerably underestimated given only parts of the sibling genomes are homeologs.

Response: Looking at the Ka/Ks results, only 14% of comparisons between Ad77271a and Ad77271b have zero values, i.e. these are probably the same copy of the homeolog/homolog. Therefore, these results are consistent with most comparisons representing homeologs/homologs. In contrast, the Ka/Ks results between AdKB and AdOHT resulted in 2 and 6% zero values, which we would expect all or most of these are homeologs/homologs. The divergence dates for AdKB and AdOHT are 6 MYA and 11 MYA respectively. We have revised our text to include this nuance of the data that it could be between 3-11 MYA to account for possible underestimation.

5. Line533. It is worth noting that Fst is a population genetic estimation based on the model for diploids. Polyploid Fst should be cautiously interpreted.

I do agree with that assembly of the “true subgenome” is of great challenges. However, the relevant conclusions in the manuscript are doubtful before new analyses specific to autopolyploid (not affected by polyploidy or homeologs) are introduced.

Response: Fst is a population-level metric that operates on the number of alleles in a given state. Yes, the given expectation depends on ploidy, but it's constant; thus like the difference in all the Fst estimators we've compared, the question isn't that they are liable to give erroneous results vs. one another, it's just that they each scale a bit differently. We have carried out these experiments before with results that have held up under reanalysis (Monnahan et al., 2019). However, we agree Fst should be used with caution, and is particularly problematic when contrasting values generated in comparisons of different ploidies (Meirmans et al., 2018); however, this estimator can be safely used on within-cyotype variation. We have added the following to the discussion. **Ln 642 to Ln 644:** “We note that Fst is a population genetic estimation based on a diploid model. Thus we interpret our polyploid Fst results cautiously.”

References:

Monnahan P, Kolář F, Baduel P, Sailer C, Koch J, Horvath R, Laenen B, Schmickl R, Paajanen P, Šrámková G, Bohutínská M, Arnold B, Weisman CM, Marhold K, Slotte T, Bomblies K, Yant L. Pervasive population genomic consequences of genome duplication in *Arabidopsis arenosa*. *Nat Ecol Evol.* 2019 Mar;3(3):457-468. doi: 10.1038/s41559-019-0807-4. Epub 2019 Feb 25. PMID: 30804518.

Meirmans PG, Liu S, van Tienderen PH. The Analysis of Polyploid Genetic Data. *J Hered.* 2018 Mar 16;109(3):283-296. doi: 10.1093/jhered/esy006. PMID: 29385510.

Besides, I am struggling about some conflicting interpretation of results here, Basing on resequencing data, the author claimed that some genomic signals, such as gene dosage of some pathway and centromere regions are “highly dynamic and location specific” among populations within the baobab species. If so, the previous conclusions on these genomic signals by comparing the baobab genome with other species seem unreliable. For example, if the centromere regions are highly dynamic, is it possible that the conclusion on centromeres (Line274-288) is just a description on one population or individual, instead of the species?

Response: All of the centromere regions were similar (same base repeat and size) across the four baobab genomes that were sequenced so we have no expectation that Ad77271a is different from all other baobabs. We also analyzed the genomes in the Wan et al. (2024) and found the same centromere base repeat and they assembled a similar amount of centromere sequence. It is known from *Arabidopsis* work (as well as other species with highly resolved centromeres) that centromere sequences can be divergent in terms of copy number and organization across different accessions/landraces (Wlodzimierz et al., 2023).

Reference:

Wlodzimierz P, Rabanal FA, Burns R, Naish M, Primetis E, Scott A, Mandáková T, Gorringer N, Tock AJ, Holland D, Fritschi K, Habring A, Lanz C, Patel C, Schlegel T, Collenberg M, Mielke M, Nordborg M, Roux F, Shirsekar G, Alonso-Blanco C, Lysak MA, Novikova PY, Bousios A, Weigel D, Henderson IR. Cycles of satellite and transposon evolution in *Arabidopsis* centromeres. *Nature.* 2023 Jun;618(7965):557-565. doi: 10.1038/s41586-023-06062-z. Epub 2023 May 17. PMID: 37198485.

Wan, JN., Wang, SW., Leitch, A.R. et al. The rise of baobab trees in Madagascar. *Nature* 629, 1091–1099 (2024). <https://doi.org/10.1038/s41586-024-07447-4>

Does the retention of gene copies in several pathways contribute to local adaptation of populations or the common adaptation of the species?

Response: We are arguing here that the extra gene copies provide local adaptation, but the *Fst* analysis is at the population level, so it seems these factors are playing a role across the at least *A. digitata* in specific environments.

Similarly, I found that the Gypsy number in Table 2 is highly various among samples, even between the two siblings. There are two possible reasons: one is due to the highly dynamics, and the other is assembly artefacts.

Response: We agree that the observed differences could stem from two primary sources: assembly artifacts or the dynamic nature of Gypsy elements. We undertook additional analyses, as we explained to reviewer #2 in the previous response, to address possible assembly artifacts (also see above responses in regards to assembly artifacts). We checked coverage across the genome, as well as the LTR-RTs, and found that the coverage was consistent across genomes and LTR-RTs, providing additional support that these regions are not expanded/contracted or misassembled. We also tested TE completeness using the LTR Assembly Index (LAI). The LAI score categorizes the assemblies within the range of a "reference" assembly (Mokhtar et al., 2023). In terms of the dynamic nature of the Gypsy elements, it was found that they can cause the genome size of *C. pentandra* to be two-fold larger compared to *B. ceiba*, although both species share the same WGM event along with baobab (Shao et al., 2024). In addition, there is a great deal of variation (almost 2 fold) across all of the baobab genomes in terms of Gypsy element content (Wan et al., 2024), suggesting that Gypsy may be variable in the baobab lineage. Also, while the copy number is different, the overall percent "Masked LTR-RT TE" (Table 2) is similar across the baobab genomes we sequenced (10-14% for *A. digitata*, and 9% for *A. za*).

Reference:

Mokhtar MM, Abd-Elhalim HM, El Allali A. A large-scale assessment of the quality of plant genome assemblies using the LTR assembly index. *AoB Plants*. 2023 Apr 4;15(3):plad015. doi: 10.1093/aobpla/plad015. PMID: 37197714; PMCID: PMC10184434.

Shao L, Jin S, Chen J, Yang G, Fan R, et al. High-quality genomes of *Bombax ceiba* and *Ceiba pentandra* provide insights into the evolution of Malvaceae species and differences in their natural fiber development. *Plant Commun*. 2024 May 13;5(5):100832. doi: 10.1016/j.xplc.2024.100832. Epub 2024 Feb 5. PMID: 38321741; PMCID: PMC11121743.

Wan, JN., Wang, SW., Leitch, A.R. et al. The rise of baobab trees in Madagascar. *Nature* 629, 1091–1099 (2024). <https://doi.org/10.1038/s41586-024-07447-4>.

Other comments:

Line168. Why is the genome size of *A. grandidieri* and *A. rubrostipa* were much larger than the others given they are all diploid?

Response: The K-mer frequency genome size estimates for both *A. grandidieri* and *A. rubrostipa* are almost exactly double the haploid genome size for the baobabs and they only have one K-mer frequency peak consistent with a homozygous diploid genome. Based on the K-mer frequency profile for *A. digitata* (Fig. 1a), and our experience with other genomes with

similar architectures (as noted to reviewer #2), this profile usually means these are hybrids or truly homozygous diploid genomes. Since the genomes are double the size and only have one peak, we hypothesize they are hybrid genomes. It has been shown that there are hybridization events between baobab species such as *A. rubrostipa* and *A. za* (Wan et al., 2024; Leong Pock et al., 2013). We have added a sentence to this effect in the results: **Ln 159 to Ln 162**: “The genome size estimates for both *A. grandidieri* and *A. rubrostipa* were double the predicted haploid genome size, and both had a single K-mer frequency peak consistent with these samples representing hybrids between two species as has been seen between *A. rubrostipa* and *A. za*”

Reference:

Wan, JN., Wang, SW., Leitch, A.R. et al. The rise of baobab trees in Madagascar. *Nature* 629, 1091–1099 (2024). <https://doi.org/10.1038/s41586-024-07447-4>.

Leong Pock Tsy JM, Lumaret R, Flaven-Noguier E, Sauve M, Dubois MP, Danthu P. Nuclear microsatellite variation in Malagasy baobabs (*Adansonia*, Bombacoideae, Malvaceae) reveals past hybridization and introgression. *Ann Bot.* 2013 Dec;112(9):1759-73. doi: 10.1093/aob/mct230. Epub 2013 Nov 1. PMID: 24187031; PMCID: PMC3838555.

Line174. Why is the heterozygosity of *A. digitata* accurate to two decimal places, but others are accurate to one decimal place? Is the conclusion “one of the highest heterozygosity” still correct if the data of *A. perrieri* and *A. za* were accurate to two decimal places as *A. digitata*?

Response: Corrected

Line220-228. I’m not so confident about the comparison initiated by the lower-quality assembly of Aza135. Is there any phenotypic difference between *A. digitata* and Aza135, which can be explained by this translocation? The authors highlighted genes associated with DNA repair and longevity, isn’t *A. za* a long-lived organism?

Response: *A. za* is considered long-living, with radiocarbon dating showing the largest *za* baobab to be 900 years old (Patrut et. al. 2016). We have rephrased the sentence to avoid implying that the translocation is directly linked to longevity. Ecological valence analysis of *A. za* revealed it to be highly competitive for habitat space, leading to greater population expansion in Madagascar, similar to *A. digitata* in mainland Africa (Wan et. al. 2024). Moreover, the study showed ongoing hybridization between *A. perrieri* and *A. za*. Further analysis suggested ancient introgression followed by significant recombination, which broke up introgressed segments as the polyploid genome diverged and developed different chromosomal structures.

Reference

Patrut A, Patrut RT, Danthu P, Leong Pock-Tsy J-M, Rakosy L, Lowy DA, et al. (2016) AMS Radiocarbon Dating of Large *Za* Baobabs (*Adansonia za*) of Madagascar. *PLoS ONE* 11(1): e0146977. <https://doi.org/10.1371/journal.pone.0146977>

Wan, JN., Wang, SW., Leitch, A.R. et al. The rise of baobab trees in Madagascar. *Nature* 629, 1091–1099 (2024). <https://doi.org/10.1038/s41586-024-07447-4>

Line297. What is the meaning of the number 54.74%, 54.94% and 62.52% shown here? Are they significantly larger than the number in other organisms? I realized that the detection method is based on deep learning, so the comparison will be reasonable if the average levels of DNA methylation in other organisms can be detected in the same way.

Response: The numbers 54.74%, 54.94%, and 62.52% represent the mean percentage of total genomic modified bases (5mC) detected using the Dorado basecaller (v.0.7.2) with "--pore r9.4.1". These methylation levels are high, as observed in Malvaceae genomes that experienced recent whole-genome duplication (Shao et al., 2024). In *G. raimondii*, 71% of CpG, 57% of CHG, and 13% of CHH are methylated, while in *T. cacao*, 44% of CpG, 15% of CHG, and 1% of CHH are methylated (genome-wide weighted detection using bisulfite methods). This methylation variation reflects the evolutionary and life histories of plant species in general (Niederhuth et al., 2016). The Oxford Nanopore Technologies (ONT) basecalling algorithms and several bioinformatic post-processing features have demonstrated higher accuracy compared to traditional bisulfite conversion methods (Ferguson et al., 2022). Note that in plants, 5mC occurs at CpG dinucleotides as well as CHG and CHH elements. Further characterization of methylation in baobab indicated hypermethylation in transposable elements contrasted with hypomethylation of genes. Moreover, we observed enhanced methylation in gene bodies and specific coding regions compared to intergenic regions (Supplementary Fig. 15).

Reference

Niederhuth CE, Bewick AJ, Ji L, Alabady MS, Kim KD, Li Q, Rohr NA, Rambani A, Burke JM, Udall JA, Egesi C, Schmutz J, Grimwood J, Jackson SA, Springer NM, Schmitz RJ. Widespread natural variation of DNA methylation within angiosperms. *Genome Biol.* 2016 Sep 27;17(1):194. doi: 10.1186/s13059-016-1059-0. PMID: 27671052; PMCID: PMC5037628.

Shao L, Jin S, Chen J, Yang G, Fan R, et al. High-quality genomes of *Bombax ceiba* and *Ceiba pentandra* provide insights into the evolution of Malvaceae species and differences in their natural fiber development. *Plant Commun.* 2024 May 13;5(5):100832. doi: 10.1016/j.xplc.2024.100832. Epub 2024 Feb 5. PMID: 38321741; PMCID: PMC11121743.

Ferguson, S., McLay, T., Andrew, R.L. et al. Species-specific basecallers improve actual accuracy of nanopore sequencing in plants. *Plant Methods* 18, 137 (2022). <https://doi.org/10.1186/s13007-022-00971-2>

Line386. "baobab has some unique gene content" sounds meaningless since each species have its own unique gene content.

Response: Revised

Line435. Two literatures addressing the genomic evolution in Malvaceae have provided some new insights which deserved consideration since WGD is one of the most important issue dissected in this study.

<https://doi.org/10.1016/j.xplc.2024.100832>

<https://doi.org/10.1073/pnas.2313921121>

Response: Thank you for the papers. The Sun et al., 2024, which was published around when we submitted this work, does provide a nice explanation for the WGD/WGM event we identify in

baobab, and as stated above we have modified our language (WGD vs WGM) in the manuscript to reflect their findings. Also, the Sun et al., and Shao et al., papers provide three new chromosome scale genomes that also show the retention of almost complete chromosomes post the WGM, supporting our finding as well.

Line444. I realized that the divergence time here is not so consistent with the time estimated using the calibrated tree (Supplementary Fig. 10). Which one is more reliable?

Response: We have revised the analysis to include additional genomes and removed *Gossypium hirsutum* as suggested. These dates match up with literature findings and analyses of synonymous substitutions per synonymous site (Ks). The baobab ancient WGM at 30 MYA, and the split between *A. digitata* and *A. za* happened around 20 MYA. Shao et al., 2024 also find a similar timing for the ancient WGM at about 30 MYA.

References

Wan, JN., Wang, SW., Leitch, A.R. et al. The rise of baobab trees in Madagascar. *Nature* 629, 1091–1099 (2024). <https://doi.org/10.1038/s41586-024-07447-4>.

Shao L, Jin S, Chen J, Yang G, Fan R, et al. High-quality genomes of *Bombax ceiba* and *Ceiba pentandra* provide insights into the evolution of Malvaceae species and differences in their natural fiber development. *Plant Commun.* 2024 May 13;5(5):100832. doi: 10.1016/j.xplc.2024.100832. Epub 2024 Feb 5. PMID: 38321741; PMCID: PMC11121743.

Line450. *G. hirsutum* is an allotetraploid cotton with AADD genome. It is not a good idea to use it for rebuilding of the gene-based species tree.

Response: We have reconstructed the gene-based species tree using *G. arboreum*, *G. raimondii*, and seven additional species instead.

Response Fig. 14: Time-calibrated phylogenetic tree of *Adansonia digitata* (Ad77271a), *Adansonia za* (Aza135), *Bombax ceiba*, *Gossypium raimondii*, *Gossypium arboreum*, *Theobroma cacao*, *Carica papaya*, *Arabidopsis thaliana*, and *Oryza sativa*. The phylogenetic analysis is based on the protein sequences of the 9 plants, with *O. sativa* as an outgroup. The timescale in million years ago (MYA) is provided at the bottom. *A. thaliana* to *O. sativa* (142.1–163.5 MYA) (<http://www.timetree.org/>) was used for calibration.

Line488. Is it possible that the high number of gene copies is just due to assembly artefacts?

Response: See previous response to genome artifacts; BUSCO, LAI, N50, and synteny support a high quality genome. Also, the recently released genomes from Shao et al., Sun et al., and Wan et al., support our genome structure and gene copy number.

References:

Wan, JN., Wang, SW., Leitch, A.R. et al. The rise of baobab trees in Madagascar. *Nature* 629, 1091–1099 (2024). <https://doi.org/10.1038/s41586-024-07447-4>

Shao L, Jin S, Chen J, Yang G, Fan R, et al. High-quality genomes of *Bombax ceiba* and *Ceiba pentandra* provide insights into the evolution of Malvaceae species and differences in their natural fiber development. *Plant Commun.* 2024 May 13;5(5):100832. doi: 10.1016/j.xplc.2024.100832. Epub 2024 Feb 5. PMID: 38321741; PMCID: PMC11121743.

Sun, P. et al. Subgenome-aware analyses reveal the genomic consequences of ancient allopolyploid hybridizations throughout the cotton family. *Proc. Natl. Acad. Sci. U. S. A.* 121, e2313921121 (2024).

Line915. Supplementary Fig. 4b is same to Fig. 1d.

Response: Fig. 1d pertains to Ad77271a, while Supplementary Fig. 4b pertains to Ad77271b. Both figures show the minor allele frequency coverage and, by extension, show the expected similarity between sibling genomes.

#####

First response to reviewers

Reviewer #1 (Remarks to the Author):

This paper describes a haploid chromosome-level reference genome for baobab, *Adansonia digitata*, as well as draft assemblies for a sibling tree, two trees from distinct locations in Africa, and a related species from Madagascar. The paper describes a paleopolyploid origin for baobab, shared with the Malvoideae ~30 million, and a more recent autotetraploidy event 3-4 million mya that coincides with a recent burst of TE insertions. Resequencing of 25 *A. digitata* trees from Africa revealed three subpopulations that suggest gene flow through most of West Africa but separated from East Africa.

I have read this paper with great interest, amongst other things because the paper describes new, high-quality genomic data of a truly iconic species. However, I have to say that I was a little disappointed after reading the paper. Please allow me to explain why.

Response: Thank you for your valuable feedback on our manuscript. Your thoughtful evaluation and suggestions are greatly appreciated, and we are grateful for your support throughout this process. We are fully dedicated to addressing all the concerns you raised and have made revisions to enhance the quality of the manuscript.

For instance, I found the entire section on whole genome duplication (WGD) quite superficial and to contain some statements that are difficult to understand. For instance, the authors state that “The Ks revealed a consistent timing of the separation of baobab-amborella and baobab-grape at 128 and 96 MYA respectively (Fig. 5b). Both cacao and cotton diverged from the ancestor of *Adansonia* around 30 MYA, which was the first piece of evidence that a baobab-specific WGD occurred around this time.” First, the approach to determine the timing of WGDs and speciation events seems only based on estimated Ks values, which is concerning since different species can have very different substitution rates. Also, how are these Ks values calculated? Which software is used to get those Ks values? Which substitution rates are used for the inference of divergence time? There is no information on this unless I have missed it. Furthermore, one must be cautious to compare Ks peak values to determine which event occurred first, a speciation or a WGD event. In my opinion, if there is some concern, dot plot analysis and in particular gene tree – species tree reconciliation methods should be applied to unequivocally (and still then ...) place WGDs in time.

Response: Generally, we start WGD analyses with the amborella and grape comparisons to determine if the general WGD/WGT history is consistent using a specific substitution rate; in this case we found that the a substitution rate of 6.56×10^{-9} (Gaut et al., 1996) recapitulated the timing of baobab-amborella and baobab-grape as has been described by many papers as noted in the text. Of course, these are estimates on the order of millions of years and should be taken

as directional just like dotplots and gene trees. All the genome comparisons were first visualized as dotplots, and syntenic depth was calculated. The syntenic depth plots as well as dotplots were added to the supplemental figures (**Supplementary Fig. 8 and Supplementary Fig. 9**). In addition, we have added time-calibrated phylogeny based on 1,296 single-copy orthologue genes to the revised manuscript (**Supplementary Fig. 10; Response Fig. 1**). The resulting timing slightly shifts the split of Ad77271a to 21 MYA from the previously estimated 17 MYA. However, the overall pattern remains consistent with the Ks analysis.

Response Fig. 1: Time-calibrated phylogenetic tree of *Theobroma cacao*, *Bombax ceiba*, *Gossypium hirsutum*, *Gossypium raimondii*, *Adansonia za* (Aza135), and *Adansonia digitata* (Ad77271a) based on 1,296 single-copy orthologous genes. Sky-blue horizontal bars indicate the 95% Highest Posterior Density (HPD) intervals of estimated divergence dates. Median node ages are shown beside the HPD bars, and timescale in million years ago (MYA) is provided at the bottom.

We have provided a more detailed explanation of the methodology used for calculating Ks values and inferences of divergence times in the revised manuscript. The supplemented “methods” section is as follows:

Page 21 lines 32~46:

Molecular dating and whole genome duplication (WGD): A total of 216 single-copy orthologues from *T. cacao*, *B. ceiba*, *G. hirsutum*, *G. raimondii*, *A.za* (Aza135), and *A. digitata* (Ad77271a), encompassing 1,296 single-copy orthologous genes, were analyzed. Protein sequences were aligned using MUSCLE (v.3.8.31) and converted to corresponding coding sequences (CDS) by substituting each amino acid with its triplet bases. The CDS for each single-copy gene family were concatenated to form a super gene for each species. The second nucleotide of each codon from the aligned sequences was extracted to construct a phylogenetic tree using BEAST (v2.7.6) under the HKY model with a Gamma category count of 4. The tree prior was set to the calibrated Yule model, with birth-rate parameters set to an Alpha (shape) parameter of 0.001 and a Beta (scale) parameter of 1000. Secondary calibration of molecular clocks was performed using priors from the <http://www.timetree.org/> database for the cotton split from bombacoid/malvoid (50 MYA)^{107,108}. For WGD, synonymous substitution rates (Ks values) were calculated using the MScan (Python version) with default parameters. A substitution rate of 6.56×10^{-9} was then utilized to estimate the time of duplication events¹⁰⁹

Reference:

Gaut, B. S., Morton, B. R., McCaig, B. C. & Clegg, M. T. Substitution rate comparisons between grasses and palms: synonymous rate differences at the nuclear gene *Adh* parallel rate differences at the plastid gene *rbcL*. *Proceedings of the National Academy of Sciences* 93, 10274–10279 (1996).

The authors state that “The self-alignments of genomes informed the timing of the most recent WGD in a species, which clarified that all of the baobab, bombax and cotton genomes shared the same WGD event at about 30 MYA, while cacao experienced its last WGD around 118 MYA that was consistent with it having only the WGT- γ (Fig. 5c).” What does the self-alignment of genomes mean? From the figure, I assume it refers to age distributions made from collections of paralogous genes (all genes, or only homeologs)? Finding similar Ks peak values for different organisms can indeed suggest a shared WGD, but it could also suggest independent WGDs in different branches at similar times in evolution.

Response: The “self-alignment of genomes” (*Ad77271a* vs. *Ad77271a*, etc.) refers to estimating the Ks based on paralog comparisons where paralogs result from past WGD events. We edited it to say “self-self alignments” with paralogs mentioned. We agree that they don’t have to share the same WGD, so we changed this wording. Based on the timing of the time calibrated tree, it implies independent WGDs. The text has been updated to make this clearer.

Second, the entire section could do with some more detailed explaining. “Both cacao and cotton diverged from the ancestor of *Adansonia* around 30 MYA, which was the first piece of evidence that a baobab-specific WGD occurred around this time”. How is this first evidence for a baobab-specific WGD? It seems problematic to infer a specific WGD event solely based on divergence time without further elucidation or reference to accompanying figures. Additionally, there’s a notable absence of a summarizing sentence indicating the presence of a WGD event shared with the Malvoideae but not with cacao. Instead of ambiguously stating “around ~30 Mya,” it would be more precise to assert that this WGD event occurred after the divergence with cacao but prior to the divergence within the ancestor of the Malvoideae.

Response: We have revised the text and asserted that the WGD event happened after the split from cacao, as shown in **Response Fig. 1**.

The authors state that “When we zoom into just the baobab complex, we saw that all genomes have a minor Ks peak around 30 MYA in addition to peaks at 4, 6, 11 and 17 MYA for *Ad77271b*, 1 *AdKB*, *AdOHT* and *Aza135* respectively (Supplementary Fig. 7). We hypothesize that the autotetraploidy event might be seen as the distance between the *Ad77271a* and *Ad77271b* siblings since they most likely represent distinct and random haplotypes, which would place the autotetraploidy event in the 4 MYA range with a similar timing of the last detectable TEs burst (Fig. 2b).” I think this is not good enough and the authors should try harder to explain what these different peaks (obtained by Ks rates) refer to? For instance, it is not clear to me what e.g., the 17 my old peak could refer to then?

Response: We have expanded the explanation of Ks peak values to offer a clearer interpretation, particularly regarding peaks like the one at 17 MYA as a speciation event, being

in the range of the Ad77271a-Aza135 split depicted by the time-calibrated tree approach (21 MYA). While we have noted the observed elevated Ks values, confirming the precise events (speciation/duplication) and their timing requires additional datasets, such as fossil records, which are beyond the scope of this paper. Nonetheless, we believe readers will appreciate the complexity of the baobab's evolutionary journey and recognize the need for more targeted experiments to validate the hypotheses raised in this study.

The authors state that “We hypothesize that the genes retained as four copies in the baobab genome may represent specific biology that was important to baobab.” There is a huge literature (none of it cited) about gene balance and gene dosage issues that explain the biased retention of genes of certain functional categories (transcription factors, developmental genes, genes encoding proteins acting in multiprotein complexes) that has not necessarily to do with ‘specific biology of importance to baobab’, especially not when WGDs are relatively recent, or very conserved (see further). All the classes of genes the authors mention are the genes typically found in excess in genomes following WGDs. This has been widely documented, not only for plants but appears to be a general phenomenon.

Related to that, the authors state that “In addition to the WGD 4 MYA, baobab also experienced a WGD 30 MYA that is shared across the Malvaceae; yet unlike other Malvaceae species studied to date, baobab retained almost all four copies resulting from the WGD. While most genomes fractionate gene copies back to a diploid state after WGD, it is thought that some genes may not be fractionated and retained to modify gene dosage in specific pathways important to the organism.” The authors do not seem to be aware that gene loss takes time, and is related to substitution rates, which are a function of generation times, metabolic rates, and population sizes. Therefore, simply assuming that the difference in gene loss or retention following WGDs between two species with significantly different life styles, is due to selection for genes of importance and can be linked to adaptation, is short-sighted, I think and could lead to wrong conclusions.

Response: The haploid baobab genome is unique amongst plant genomes sequenced to date in that it has retained almost four full chromosomes after the WGD 30 MYA; if the base chromosome number is 10 based on cacao, then the 42 chromosomes represent mostly a retention and some splitting/fusions (**Response Fig. 2**). Soybean is similar and has retained almost all four copies after the last WGD 15 MYA. While these processes take time, based on the current high-quality genomes most fractionate back to two copies in the time frames analyzed. This assessment is based on an analysis of 123 high quality plant genomes (**Zhao et al., 2021**) that were analyzed for WGD, and we further analyzed to understand how often this happens in reference to the circadian, light and flowering time genes (**Michael 2021**). We do cite what we think is the most relevant background on fractionation and dosage (**Cheng et al., 2018**) and now we have added one other (**Ruju et al., 2020**). However, *A. digitata* does not retain all four copies suggesting that some genes are lost while the overall organization and structure of the chromosomes are retained. We agree that the language that these are “baobab specific” may be too strong so we have removed this wording. These concepts are consistent with current thinking in the genomics field regarding polyploidy and WGD (**Van de peer et al., 2017**). We also point out cases in the text where genes have been retained and play functions specific to a species such as in soybean and CAM plants; we highlight this rare event in Fig. 5d where syntenic copies of clock/flowering/temperature genes are retained in four copies. We are aware that transcription factors, developmental genes, and genes encoding proteins acting in multiprotein complexes are often retained after WGD and this is why we have highlighted gene

classes that are distinct: chromatin, exocytosis, and flower timing/development; the latter category we further develop later.

Response Fig. 2: The *A. digitata* genome retained many almost full chromosomes (Chr) after whole genome duplication (WGD). The Ad77271a (blue, bottom) genome was aligned to the cacao (*Theobroma cacao*; red, top) genome at the protein level with MCscan (Python version with default parameters). The cacao genome was used since it clearly displays the retention of full Chr structure (length) with the Ad77271a genome (Fig. 5a main text), which also highlights the continuity and accuracy of the Ad77271 assembly. The syntenic regions were visualized using the `jcvi.graphics.karyotype` tool with the settings `--minspan=30`. The resulting anchor file was analyzed for the number of syntenic connections between cacao and Ad77271a Chrs, and the syntenic blocks greater than 10 or 300 genes were retained for plotting. Each cacao Chr was plotted separately to highlight the complete Ad77271a Chr, and colors were randomly assigned to syntenic blocks by Ad77271a Chr. Cacao Chr 1, 2, 3, 4, 6, and 9 had all four copies retained in the Ad77271a genome with different levels and areas of fractionation. Chr 1 and 4 were fused before the Ad77271a WGD resulting in the Chr 1 arms being split into 8 copies with connections to Chr4. A similar yet more complicated fusion/breaking occurred

between Chr 5, 7, 8, and 10, which is why the four Chr retention pattern is less obvious in these plots compared to the dotplot (Fig. 5a main text).

References:

Cheng, F. *et al.* Gene retention, fractionation and subgenome differences in polyploid plants. *Nat Plants* **4**, 258–268 (2018).

Raju, S. K. K. Gene Dosage Balance Immediately following Whole-Genome Duplication in Arabidopsis. *The Plant cell* vol. 32 1344–1345 (2020).

Zhao T, Zwaenepoel A, Xue JY, Kao SM, Li Z, Schranz ME, Van de Peer Y. Whole- genome microsynteny-based phylogeny of angiosperms. *Nat Commun.* 2021 Jun 9;12(1):3498. doi: 10.1038/s41467-021-23665-0. PMID: 34108452; PMCID: PMC8190143.

Michael TP. Core circadian clock and light signaling genes brought into genetic linkage across the green lineage. *Plant Physiol.* 2022 Sep 28;190(2):1037-1056. doi: 10.1093/plphys/kiac276. PMID: 35674369; PMCID: PMC9516744.

Van de Peer Y, Mizrahi E, Marchal K. The evolutionary significance of polyploidy. *Nat Rev Genet.* 2017 Jul;18(7):411-424. doi: 10.1038/nrg.2017.26. Epub 2017 May 15. PMID: 28502977.

The authors do discuss some gene families in more detail, but most of it is very speculative. As a matter of fact, the paper is riddled with statements such as ‘we hypothesise’, ‘we speculate’, ‘these results could point’, ‘may provide insight’, ‘provides clues’, etc., which, I have to say, become a bit frustrating after a while.

Response: We understand your concerns about the speculative nature of some of our statements; however, this is a genome manuscript where many things cannot be tested readily. We have revised the manuscript providing more concrete statements where possible. Additionally, we have clearly distinguished between well-supported findings and hypotheses to enhance the clarity and robustness of our discussion. We would like to note that for many of these “speculative” findings, we provide several lines of evidence supporting them. Our main biological finding concerning circadian, chromatin and flowering genes were supported in both our WGD retention analysis, as well as the variation analyses across the 25 resequenced trees (population Fst and local PCA). We try not to overstate our findings, but we hope readers will be able to make their own conclusions based on the several lines of evidence we present. We have created a FigShare with all the raw data so researchers can check the veracity of our data and analyses.

The authors also conclude that the whole genome duplication unveiled in *A. digitata*, about 4 mya, coincided with an unprecedented amplification of DNA transposable elements (TEs) compared to Long Terminal Repeat Retrotransposons (LTR-RT). Again, this is very speculative given the way the authors have computed divergence times and estimates of the timing of WGDs. Also, why is the WGD inferred at 4 MYA consistent with an autotetraploidy event. Could this not have been an allopolyploidy event?

Response: We have conducted additional analyses using a time-calibrated phylogenetic tree as you suggested, which confirmed that the timing of speciation and WGD events are congruent

(Response Fig. 1). Therefore, the event 4 MYA that separates the haplotypes (Ad77271a vs. Ad77271b) represents the polyploidy event. It is true that distinguishing between auto- and allopolyploidy can be difficult because the hybridization of closely related species can appear as an autotetraploid event. However, we do not detect any evidence for an allopolyploidy event, such as distinct centromere sequences (Response Fig. 3) like we have seen in homeologs of *Eragrostis tef* (VanBuren et al., 2020), or a high level of variation in the subgenomes (once again the comparison between Ad77271a vs. Ad77271b). Therefore, based on our genome assemblies and annotations, we can assert that the WGD in *A. digitata* is consistent with an autotetraploid rather than an allotetraploid event. We now provide a clearer rationale for our inferences, supported by additional data and analyses.

Response Fig. 3: The *A. digitata* (Ad77271a) centromere arrays are identical across all 42 chromosomes. The centromere base repeats in arrays greater than 10 kb were extracted and

aligned using mafft (--reorder --adjustdirection --maxiterate 1000 --retree 1 --globalpair) and visualized with FigTree. Only arrays from chromosomes 6, 7, 9, and 20 (full chromosomes with cacao) are highlighted with different colors to show that there is not a specific pattern by chromosome.

Reference:

VanBuren R, Man Wai C, Wang X, Pardo J, Yocca AE, Wang H, Chaluvadi SR, Han G, Bryant D, Edger PP, Messing J, Sorrells ME, Mockler TC, Bennetzen JL, Michael TP. Exceptional subgenome stability and functional divergence in the allotetraploid Ethiopian cereal teff. *Nat Commun.* 2020 Feb 14;11(1):884. doi: 10.1038/s41467-020-14724-z. PMID: 32060277; PMCID: PMC7021729.

Other comments:

Page 4 line 29:

“We long read Oxford Nanopore Technologies (ONT) sequenced...” sounds awkward, I think.

Response: Revised.

Page 4, line 36:

The flow cytometry result shows that the genome size is around 920Mb, while the Kmer analysis result suggests a genome size of 749 Mb. However, the final haploid genome size is estimated at around 675Mb, so how to explain the 70 Mb size difference between the genome survey result and the final assembly? A 1.9% heterozygosity level (for *A. digitata*) is high, could this be part of the explanation (if the assembly is not completely phased)?

Response: Genome sizes from either flow cytometry or k-mer frequency are both estimates and can vary based on several factors (**Pflug et al., 2020**). Flow cytometry can result in estimates that differ from the actual genome size due to the selection of cultivars, staining techniques or the controls used, while k-mer genome size estimates vary due to sequencing reads used (long vs short reads), quality of sequencing data, coverage of sequencing data and the k-mer size chosen. A 70 Mb difference between the assembled genome size and k-mer frequency estimate is in the margin of error; however, we updated Fig. 1 as suggested with the genome scope result and now the k-mer frequency estimate is 659 Mb (different estimation methods even provide slightly different results), which is directly in line with the assembled genome size. Also, the genome scope figure more clearly highlights the broad single peak that we see in some autopolyploids (discussed in the text); the broad peak also causes issues with accurately estimating genome size using k-mer frequency.

Some of the hardest regions of a genome to assemble are the centromere, rDNA and TEs (**Michael and Jackson, 2013**). In the first wave of plant genomes based on either Sanger or short read sequencing, the centromere, rDNA and TEs were often collapsed leading to an assembled genome size smaller than the actual. Now with long read sequencing we are now rarely finding that regions collapsed and in addition we are assembling full centromeres and TEs while rDNA arrays still are problematic (**Naish et al., 2021**). While the centromere arrays are

large in our assemblies, there is always a chance that we have collapsed the rDNA array, which we quantify in the text.

We state in the text that we have only assembled the haploid genome of *A. digitata*. Therefore, the haploid representation most likely contains a combination of non-phased variation from the four subgenomes. While we have tried to separate the sub genomes, we were unsuccessful (we suspected due to the low level of SVs and SNPs between subgenomes), so we sequenced a sibling to (Ad77271b) to ascertain the level of heterozygosity. While the overall heterozygosity is estimated at 1.9%, which reflects the underlying four subgenomes (haplotypes) of the tetraploid, the comparison of the sibling assemblies Ad77271a and Ad77271b was 0.65%.

References:

Pflug JM, Holmes VR, Burrus C, Johnston JS, Maddison DR. Measuring Genome Sizes Using Read-Depth, k-mers, and Flow Cytometry: Methodological Comparisons in Beetles (Coleoptera). *G3 (Bethesda)*. 2020 Sep 2;10(9):3047-3060. doi: 10.1534/g3.120.401028. PMID: 32601059; PMCID: PMC7466995.

Michael and Jackson. The First 50 Plant Genomes. *The Plant Genome*. 2013 July 01; <https://doi.org/10.3835/plantgenome2013.03.0001in>.

Naish M, Alonge M, Wlodzimierz P, Tock AJ, Abramson BW, Schmücker A, Mandáková T, Jamge B, Lambing C, Kuo P, Yelina N, Hartwick N, Colt K, Smith LM, Ton J, Kakutani T, Martienssen RA, Schneeberger K, Lysak MA, Berger F, Bousios A, Michael TP, Schatz MC, Henderson IR. The genetic and epigenetic landscape of the Arabidopsis centromeres. *Science*. 2021 Nov 12;374(6569):eabi7489. doi: 10.1126/science.abi7489. Epub 2021 Nov 12. PMID: 34762468.

Page 5, lines 23~32:

This large translocation could hold significance for the baobab tree. Nevertheless, the absence of *RPA1B* does not correlate with the translocation of these genes. We cannot attribute the function of losing this gene to the function of gene translocation. They do not have a direct relationship.

Response: We don't say that *RPA1B* is lost. We say that *RPA1B* is part of the 1 Mb translocation along with other genes that have potentially relevant functions and gene ontology (GO) enrichment. We highlight *RPA1B* because it also overlaps with UV-B signaling that is important in baobab due to its unique copies of these genes in the genome (highlighted later in the text).

Page5 line 34:

The Kmer analysis result shown in Fig 1a looks strange. With a heterozygosity level of 1.9%, the result should show a peak indicating Kmers from heterozygous loci and a main peak. Why not just provide the result from GenomeScope?

Response: In the revised version, we have substituted the image with genome scope output as suggested (**Response Fig. 4**). As noted above, genome scope estimates the genome size to be similar in size to the assembled genome (659 vs. 674 Mb), and the heterozygosity is 1.63%, which is similar to our estimate based on read mapping (1.9%).

Response Fig. 4: Characteristics of baobab (*Adansonia digitata*) genome. a GenomScope estimation of *A. digitata* genome size using 19-mer sequence counts and ploidy set to 4, the K-mer frequency depicts a unimodal pattern suggesting a diploid homozygous genome with a size of 659 Mb.

Page5 line 35:

Is Fig. s 1a,b the same as Fig. 1a;b? This causes confusion (throughout the manuscript).

Response: Revised.

Page 6 Table 2:

Some information in Table 2 is not clear. Does the N50 shown in the table refers to the Contig N50 or the Scaffold N50? Does the BUSCO assessment refers to the genome assembly or to (the completeness of) the annotation? The table would be better divided into three subsections, I think, namely genome assembly, repeat content, and annotation.

Response: We have addressed your concerns in the revised version. The table is now divided into three subsections, as suggested.

Page 7 line 25-28:

This information is already shown in Table 2, there is no need to reiterate it here.

Response: We have removed the redundant information.

Page 8 Fig 2:

Is it possible to use the same color palette for the same TE element?

Response: We have implemented as suggested (**Response Fig. 5**).

Response Fig. 5: Insights into transposable elements (TEs) in baobab. **a** Barplot comparison of TEs classification and sizes in five baobab genomes: Ad77271a, Ad77271b, Aza135, AdKB, and AdOHT. Aza135 is highlighted in the center to emphasize TE divergence. Except for Aza135 (*A. za*), the other four genomes belong to *A. digitata*. Distinct colors in the legend denote TE classes. The TE proportion to genome size (Mb) and percentage are displayed on the left and right y-axes, respectively. **b** Density plots of intact TEs, displaying the distribution of different types of TEs. Mutator transposons showed a burst around 11 million years ago. **c** Accumulation of DNA mutators in the centromeric regions of the Ad77271a chromosome 3. Four mutator Terminal Inverted Repeat (TIR) transposons (TE_00000631: Mutator631, TE_00000927: Mutator927, TE_00000845: Mutator845, and TE_00000967: Mutator967) with a potential role in centromere creation or repositioning are displayed. **d** High solo to intact LTR ratio in Ad77271a chromosomes shows an aggressive purging mechanism following WGD, contributing to a smaller genome size (659 Mb).

Page 8 line 19:
Fig.s 3 -> Fig. 3?

Response: We have corrected the figure reference to "Fig. 3" in the revised version.

Page 9 line 6:
Fig.s 3 -> Fig. 3?

Response: Revised.

Page 9 line 7:
The methylation part is very descriptive. It would be interesting to investigate whether the methylation links to any trait or adaptation of baobab.

Response: Investigating the potential links between methylation and specific traits or adaptations of baobab is indeed an intriguing avenue for future research. We have expanded the analysis to include global methylation patterns in relation to genes and transposable elements. Our observations indicate the presence of methylation across the entire genome, with subtle elevations on DNA mutators or centromeric regions, such as on chromosome 11 (**Response Fig. 6**).

Response Fig. 6: Methylation patterns across *A. digitata* chromosomes. Global distribution of genome features in *A. digitata*. Centromere arrays are plotted as darkgreen bars at $y=0.8$. Genes are represented in blue, Mutator transposons in green, CACTA transposons in red, Gypsy retrotransposons in purple, Copia retrotransposons in orange, and CpG methylation in black. Transposable elements, such as Mutator transposons, tend to be located opposite to genes and in the centromeric regions. Methylation is present throughout the chromosome. The window positions are divided by the chromosome total length to create a relative scale between 0 and 1.

Page 9 line 26:

It would be good to add another *Adansonia* (*Aza135*) in the gene family analysis to check whether it is also having these baobab specific orthogroups.

Response: We have added *Aza135*, *AdOHT* and *AdKB* to the gene family analysis (**Response Fig. 7**) and discussed the gene ontology (GO) terms of baobab-specific orthogroups (**Response Fig. 8**).

Response Fig. 7: An UpSet plot showing orthogroups across 20 plant species: *A. digitata* (Ad77271a, Ad77271b, AdKB, AdOHT), *A. za* (Aza135), *A. calamus*, *A. gramineus*, *A. thaliana*, *A. trichopoda*, *B. ceiba*, *D. zibethinus*, *G. biloba*, *G. hirsutum*, *G. raimondii*, *K. fedtschenkoi*, *P. trichocarpa*, *Q. rubra*, *S. grande*, *T. cacao* and *V. vinifera*. The number of species-specific orthogroups and shared orthogroups is shown on the main bar y-axis; red bars correspond to the baobabs. The x-axis bars correspond to the number of orthogroups containing species.

Response Fig. 8: Subset of the 125 significant GO terms in 817 specific orthogroups unique to baobab (*Adansonia digitata*, including accessions Ad77271a, Ad77271b, AdKB, and AdOHT, and *Adansonia za* Aza135) compared to a diverse set of fifteen plant species: *Acorus calamus*, *Acorus gramineus*, *Arabidopsis thaliana*, *Amborella trichopoda*, *Bombax ceiba*, *Durio zibethinus*, *Ginkgo biloba*, *Gossypium hirsutum*, *Gossypium raimondii*, *Kalanchoe fedtschenkoi*, *Populus trichocarpa*, *Quercus rubra*, *Symphonia grande*, *Theobroma cacao*, and *Vitis vinifera*. The x-axis represents the gene ratio (gene count / 1178 genes). The y-axis lists the GO term names. Dot size indicates the study count, while dot color reflects the p-values, adjusted using the FDR BH method implemented in GOATOOLS v1.3.11. The color gradient corresponds to the range of p-values.

Cacao is not the sister group of baobab based on Fig. 4d. Is there something special in the gene families shared by baobab and cacao as the authors only list the result between these two species?

Response: We selected cacao as the comparison species due to its status as an outgroup of Malvaceae species with extensive evolutionary information available.

Page 12 line 9:

Fig.s 5 -> Fig. 5; Fig 5a is a syntenic dot plot, not syntenic depth plot.

Response: Revised.

In Supplementary Fig. 9, please highlight some syntenic regions.

Response: Implemented as suggested, thank you (**Response Fig. 9**).

Response Fig. 9: Comparative analysis of autotetraploid *A. digitata* and diploid *Theobroma cacao* genomes. Inner to outer tracks depict: a Syntenic genes, b GC content, c Gene density, and d Chromosome information. Prefixes 'Ad' and 'Tc' denote baobab and cacao respectively. The circos plot illustrates 42 pseudomolecules for baobab and 10 for cacao, with a window size of 100 kb. The red asterisk highlights the metacentric and acrocentric centromeres in baobab.

Page 12 line 11:

Fig.s 5 -> Fig. 5

Response: Revised.

Page 12 line 13-14:

Should it be “amborella and grape...”? The ratio also does not seem correct, could the author please also provide the syntenic depth plot for those comparisons.

Response: We have paraphrased the sentence for clarity and provided syntenic depth plots for the comparisons (**Response Fig. 10**).

Response Fig. 10: Syntenic depth ratios between Ad77271a and **a** Ad77271b **b** AdKB **c** AdOHT **d** Aza135 **e** cotton **f** bombax **g** cacao **h** grape and **i** amborella.

Page 12 line 16:
Fig.s 4 -> Fig.4

Response: Revised.

Page 14 line 1:
“as well 1 as 1 and 3 were less related”?

Response: We have revised the sentence to clarify that population 1 is as distantly related to population 2 as population 3 is to population 2.

Page 19 line 10:

Could the authors kindly provide the parameters which they provided to Funannotate? What evidence are used for gene prediction?

Response: The revised manuscript is as follows:

Repeats and gene prediction: EDTA (v1.9.6)⁹⁸ was used to construct a repeat library and softmask complex repeats. Tandem Repeats Finder (v4.09)⁹⁹ was employed to identify centromere and telomere sequences, as well as softmask simple repeats. The ONT cDNA library was mapped against the reference using Minimap2 with splice presets and then assembled using Stringtie (v2.2.1) with the long reads processing flag. Genes in baobab were predicted via the Funannotate (v1.8.2) pipeline with the Stringtie transcript models included as

evidence in addition to the uniprot protein set. Predicted proteins were characterized using EggNOG-mapper v2.0.1¹⁰⁰.

References:

Ou, S. et al. Author Correction: Benchmarking transposable element annotation methods for creation of a streamlined, comprehensive pipeline. *Genome Biol.* 23, 76 (2022).

Benson, G. Tandem repeats finder: a program to analyze DNA sequences. *Nucleic Acids Res.* 27, 573–580 (1999).

Cantalapiedra, C. P., Hernández-Plaza, A., Letunic, I., Bork, P. & Huerta-Cepas, J. eggNOG-mapper v2: Functional Annotation, Orthology Assignments, and Domain Prediction at the Metagenomic Scale. *Mol. Biol. Evol.* 38, 5825–5829 (2021).

Page 19 line 14:

Substitution rate varied a lot between species, Malvaceae and Salicaceae are not that closely related, using the substitution rate from *Populus* is probably not accurate enough for the calculation of the LTR insertion time.

Response: While we recognize that baobabs have distinct evolutionary histories, which may limit the direct applicability of substitution rates from other taxa. As noted above, we leveraged this substitution rate because it recapitulated the published timing of known events.

In conclusion, although the paper discusses the genomes of an iconic species, I found the paper to be very descriptive and to be honest, seems to reveal very little about the evolution and/or adaptation of baobab, as nevertheless suggested by the title. The authors state that ‘this research not only unravels baobab genomic evolution mysteries but also provides a crucial sequence for expediting gene discovery, enhancing breeding efforts, and aiding baobab species conservation.’ I’m not sure what ‘genomic evolution mysteries’ the authors refer to or solved? The paper lists TEs, describes a WGD, there is some information on methylation, and some discussion on the genetic diversity in baobab, none of which provides, in my opinion, exciting novel discoveries that shed light on the evolution, diversification, or adaptation and specific lifestyle of baobab. Again, what is the significance of statements such as ‘these results indicate there are geographical or environmental factors that have limited gene flow between these populations.’?!

Response: We have reevaluated our conclusions to ensure that they effectively highlight the novel insights gained from our research. Additionally, we have refined our language to better articulate the evolutionary implications of our findings, emphasizing their relevance to baobab diversification and adaptation. We appreciate your input and have taken steps to strengthen the manuscript accordingly.

Reviewer #2 (Remarks to the Author):

The manuscript presents a comprehensive genomic study of the African baobab tree, *Adansonia digitata*, a long-lived species with significant ecological and economic importance. The authors constructed a chromosome-level reference genome for *A. digitata* using long-read Oxford Nanopore sequencing and HiC data and draft assemblies for related species,

uncovering unique genomic features. Unlike most plant genomes dominated by LTR retrotransposons, the baobab genome has a high proportion of DNA transposons. The authors report that a whole-genome duplication event occurred 30 million years ago, shared with other members of the Malvaceae family. In addition, an autotetraploid event 4 million years ago that correlates with a burst of DNA transposon insertions was identified. Moreover, the authors report that baobab retains multiple copies of circadian, light, and flowering-related genes, which may contribute to its longevity and unique pollination strategies. They resequenced 25 *A. digitata* trees across Africa, finding geographic variation that divided the species into three main subpopulations.

Response: We appreciate your professional summary of our work.

The baobab genome is undoubtedly fascinating. The genomic resources and insights offered by this study could form the basis for conservation efforts focused on preserving the diversity of the baobab. This research significantly enhances our understanding of the baobab genome and its evolutionary history. Perhaps clearer conclusions on the practical implications for breeding and conservation strategies could be provided.

Response: We have revised the manuscript to include clearer significance of the study

Page 3 line 42 to page 4 line 17:

Here, we report genome size estimates for all eight recognized baobab species using a K-mer-based method from short-read genomic sequences. This method can provide independent estimates to those obtained previously using Feulgen staining and flow cytometry ¹. Our primary objectives are to create a reference genome, evaluate genetic diversity, investigate key genomic features, and study evolutionary history. We generated a haploid chromosome-scale assembly of *A. digitata* (Ad77271a; originally from Tanzania) as well as long read draft assemblies of an Ad77271a sibling (Ad77271b), and offspring of the following trees; the Kord Bao Sudan (AdKB), the Okahao Heritage Tree from Namibia (AdOHT), and an additional species, *A. za* (Aza135) from Madagascar. Additionally, we resequenced 25 additional *A. digitata* trees representing different regions of Africa with short reads to assess genetic diversity. The findings from this work revealed: (a) a high proportion of DNA transposons compared to LTR retrotransposons, suggesting an unusual genomic structure; (b) specific set of DNA mutator transposons potentially contributing to new centromere formation; (c) retention of nearly complete chromosomes post-whole genome duplication (WGD), indicating a unique evolutionary path in genome maintenance; (d) a recent WGD event approximately 30 million years ago leading to gene expansion in flower development, chromatin regulation, and exocytosis; (e) expansion of UV RESISTANCE LOCUS 8 (UVR8) genes, which suggests a unique mechanism for genome protection in long-living trees; and (f) discovery of genetically distinct Namibian baobab populations, enhancing our understanding of baobab diversity and suggesting the need for targeted conservation initiatives. These insights will facilitate breeding, conservation and domestication efforts.

In addition, here are my detailed comments:

1. Baobab is autotetraploid. It went through a whole genome duplication and has $2n=4x=168$

chromosomes. The reported reference genome has 42. So not only the homologs are collapsed as typical for reference genome assemblies but also the homeologs. While I recognize the challenges in assembling sequences to individual chromosomes, I am concerned about the potential impact on the study's conclusions. 4 million years is a lot of time for mutations to happen, hence the sequences wouldn't necessarily collapse but lead to assembly artefacts and problems especially if the subgenomes don't recombine. Do they? If they don't, they must have evolved and changed in 4 million years. Collapsing them can lead to a lot of wrong conclusions. Was this considered when running the expansion/contraction analyses of gene clusters and copy number variation analyses? The expansion in copy number of specific genes mentioned in the manuscript can also be due to this. Also the translocations. These might be present in one subgenome and not in the other. Have you tried to separate the subgenomes with your long reads?

Response: We agree; we were hoping to resolve all four haplotypes of the baobab genome. As we note in the text, additional methods (that may or may not be possible in baobab) will be required to assemble the fully phased, haplotype resolved genome. For autotetraploid genomes with an even higher level of heterozygosity (both SV and SNPs/INDEL) such as the potato genome, a pollen approach was required. We did try to resolve the subgenomes with the long reads but were unsuccessful.

The k-mer frequency plot suggests that the sub genomes are very similar since we observe a single peak. As noted in the text, and discussed below, we also have observed this k-mer frequency profile in *Lemna* that has a low level of SV and SNPs/INDEL variation. In figure 1a we include the k-mer frequency plot that has only one peak, which is consistent with a homozygous diploid genome. This is a similar plot you would see for an inbreeding species like *Arabidopsis* which has very low heterozygosity and is diploid. Since our allele frequency analysis confirmed *A. digitata* is tetraploid, as was predicted by other methods, the k-mer frequency suggests that the subgenomes are very similar at the nucleotide level. As noted to reviewer #1, the heterozygosity between the subgenomes is on the order of 0.65%.

We conducted a coverage analysis (in response to DNA TE vs LTR query) and we observed consistent coverage across the genome and repeats suggesting there is not large-scale collapsing. In addition, across the four *A. digitata* genomes we find a high level of synteny, especially with the diploid *A. za* and cacao genomes, which suggests we do not have a high level of collapsing. All four *A. digitata* genomes displayed the same translocation event with *A. za* consistent with this being real and not an artifact of collapsing (it is unlikely all four genomes would collapse in the same way). The four *A. digitata* genomes share similar gene expansions/contractions consistent with these gene families being resolved correctly. All the measures that we have tried (coverage, BUSCO, k-mer, synteny, etc) suggest that the haploid representation is a high quality, non-collapsed version of the genome.

2. Are you sure about "buhibab"? It happens that I speak Arabic and I don't recognize the word at all. In standard Arabic, "fruit with many seeds" would typically be expressed as "فاكهة ذات بذور كثيرة" - *fākihah dhāt bidhūr kathīrah* or "فاكهة ذات حبوب كثيرة" - *fākihah dhāt ḥubūb kathīrah*, where "بذور" - *bidhūr* and "حبوب" - *ḥubūb* both mean seeds.

You are citing a citation, that is citing

this: <https://www.sciencedirect.com/science/article/pii/S0254629911001189#bb0160> and

this: https://agritrop.cirad.fr/531802/1/document_531802.pdf and the first is citing the second.

So it all comes down to Diop et al. 2005 which mentions that the word "baobab" originates from the Arabic term "bu hibab," which is claimed to mean "fruit with many seeds." However, the text

notes that this origin is controversial, and various explorers and researchers have used different terms to describe the baobab tree and its fruit over time. The concept that "bu hibab" might mean "fruit with many seeds" could either be an adaptation or interpretation of the term in historical texts.

Hibab sounds close to ḥubūb/ huboob but anyway, obviously this is not a solid fact. So please, if you don't mind and unless you can be sure about the origin, please rephrase your sentence "The word "baobab" comes from the Arabic name 'buhibab,' which means a fruit with many seeds1."

Response: We appreciate your perspective on this matter. Given the uncertainty surrounding the origin of the word "baobab," we have removed the specific Arabic term "buhibab" to avoid potential inaccuracies.

3. There is a translocation between the *A. digitata* species and Aza135 on Chr23 that moved ~120 genes including genes related to longevity closer to the telomere. The hypothesis is that telomere-associated proteins may be more active or better maintained near telomeric regions, I suppose? E.g. moving DNA repair genes closer to telomeres could facilitate more efficient repair of telomeric DNA and the maintenance of chromosome stability, right? Maybe you can make the hypothesis more clear. But then again this only affects one chromosome. And also is there is a difference in terms of longevity between Aza135 and the other species?

Response: In the section on the translocation, we are just highlighting that the 120 genes in this region share interesting biology (and GO enrichment) related to our finding of unique UV-B genes; see the comment to reviewer#1. It is an interesting hypothesis that moving genes closer to the telomere may play a role in maintenance or chromosome stability. Another hypothesis is that this translocation creates a species barrier between *A. digitata* and *A. za*, but this is probably too speculative to include as well.

4. How can the variation in repeat content be explained? Does it correlate with the phylogeny?

Response: The repeat variation is consistent across the *A. digitata* genomes (44-45%) even though we have seen that the repeat variation can vary substantially across accessions of the same species. The fact that *A. za* has an overall lower repeat content (35%) could reflect a different evolutionary trajectory of repeats (it only takes one LTR to copy and paste to cause genome expansion) or it could be the specific *A. za* tree we sequenced. We have too few genomes and species to draw general conclusions about repeat content and phylogeny. However, the consistency across the *A. digitata* genomes suggests repeat content could be conserved across different trees.

5. By inter-homeolog recombination you mean that homeologs chromosomes recombine, is that correct? I can't find a description of that in the Lemna minor publication. Maybe I'm overlooking.

Response: We have edited this section for clarity. We are trying to make the case that in polyploid genomes with low levels of structural variation (SV) and SNPs/INDELS, a single k-mer peak is sometimes observed.

6. It is very interesting that there are more DNA than LTR transposons. However, don't you think

it is possible that a lot of LTR elements collapsed in the ONT assembly? You can see that looking at depth of coverage and mapping rates when mapping the reads back to the genome.

Response: All five assemblies have more DNA TEs, and we don't see any bias in coverage suggesting the LTRs are not collapsed. Generally, with long read based assemblies we don't see the LTRs collapsing; generally the read length is greater than TEs allowing their proper resolution. However, we conducted a coverage analysis across the genome, as well as the three classes of LTR-RTs, and found that the coverage was consistent across genomes and LTRs, providing an additional data point that these regions are fully assembled. We tested this another way using the LTR Assembly Index (LAI) for the whole genome (calculated using https://github.com/oushujun/LTR_retriever) and found mean LAI was 9 with a standard deviation of 4.87 (sliding window size of 3 Mb with 300 Kb steps). This LAI score categorizes our assembly within the range of a "reference" assembly (Mokhtar et al. 2023).

Reference:

Mokhtar MM, Abd-Elhalim HM, El Allali A. A large-scale assessment of the quality of plant genome assemblies using the LTR assembly index. *AoB Plants*. 2023 Apr 4;15(3):plad015. doi: 10.1093/aobpla/plad015. PMID: 37197714; PMCID: PMC10184434.

7. Why do AdOHT and Aza135 have such a low BUSCO score?

Response: BUSCO scores of 93 and 94% are still quite respectable and appropriate for what we use them for in this manuscript. These BUSCO scores are lower than Ad77271a, Ad77271b and AdKB, which are almost complete (C: 98%), but the AdOHT and Aza135 BUSCO scores are still quite good.

8. Fig. 4c: did you find this enriched in the gene family expansions of baobab? Can you please provide the tables of enriched terms?

Response: Yes, we found genes shown in Fig. 4c enriched and we have included a supplementary table (13 and 14) in revised version of the manuscript

9. Did you only use two of the baobab genomes for Orthofinder? Why?

Response: In the revised manuscript, we show all five genomes for which we have long reads (**Response Fig. 5**). However, for the identification of baobab single-copy genes and the additional time-calibrated phylogenetic tree analysis (**Response Fig. 1**), we focused specifically on the reference genome (Ad77271a and the sibling genome). This decision was made to ensure consistency and accuracy in our comparative genomic analyses, as the reference genome provides the most comprehensive and well-annotated dataset for these specific analyses.

Response Fig. 7: An UpSet plot showing orthogroups across 20 plant species: *A. digitata* (Ad77271a, Ad77271b, AdKB, AdOHT), *A. za* (Aza135), *A. calamus*, *A. gramineus*, *A. thaliana*, *A. trichocarpa*, *B. ceiba*, *D. zibethinus*, *G. biloba*, *G. hirsutum*, *G. raimondii*, *K. fedtschenkoii*, *P. trichocarpa*, *Q. rubra*, *S. grande*, *T. cacao*, *V. vinifera*. The number of species-specific orthogroups and shared orthogroups is shown on the main bar y-axis; red bars correspond to the baobabs. The x-axis bars correspond to the number of orthogroups containing species.

10. The cells in your suppl. table 7, column E are being formatted to dates from row 96 onwards.

Response: Revised, thanks.

11. Please add the URL of the Baobab database <https://resources.michael.salk.edu/baobab/index.html> to the manuscript. The link in the text is not working.

Response: We have updated it and ensured it works properly (**Response Fig. 11**).

Baobab Genome Database

Response Fig. 11: Baobab genome portal: Access the baobab genome portal at <https://resources.michael.salk.edu/baobab/index.html> for comprehensive information and resources on baobab genomes.

12. Page 13, line 28: (population 3; top left; green circles). You need to refer to the figure. Fig 6a, I assume.

Response: Yes, you are correct. The sentence now reads: "population 3; top left; green circles (Fig. 6a)." We have made the necessary corrections.

13. "Genes in baobab were predicted via the Funannotate (v1.8.2) pipeline with modifications" What modifications? And were all the baobab genomes annotated like this?

Response: We have added the specific annotation parameters used in the Funannotate pipeline in the revised version of the manuscript; see the comment to reviewer#1. Yes, all the baobab genomes were annotated using the same pipeline to ensure consistency across our analyses.

14. Would it be possible to name the chromosomes chr1 instead of just "1"?

Response: We have updated the manuscript to name the chromosomes as "chr1" instead of just "1."

Reviewer #3 (Remarks to the Author):

I was happy to read this manuscript presenting very interesting research and very much needed if the domestication of African baobab should be started. The introduction summarizes the state of our knowledge of African baobab but in the end, I miss much more clearly explained problem statement and also the general and specific objectives should be expressed in better way. E.g. why we need to know the ploidy of *A. digitata*, and how it can help us in further domestication of this species.

Response: Thank you for your suggestions and encouraging feedback. We have revised the introduction to more clearly articulate the problem statement and explicitly state the general and specific objectives of the study, including the importance of understanding the ploidy of *A. digitata*, and its implications for domestication efforts.

The supplemented words in the "Introduction" section are as follows:

Page 3 lines 1~17

Baobabs are some of the oldest and largest non-clonal living organisms on Earth with trees that can live over 2,400 years with canopy sizes of greater than 500 m³ and trunks reaching diameters of up to 10.8 meters (35 feet)⁸. However, baobabs are unlike most large and long-lived trees; they are succulents characterized by parenchyma-rich tissues that efficiently store

water and therefore, do not form “growth rings” or true wood ⁹. Achieving maturity in the wild presents a considerable challenge for baobab trees since seedlings face predation from caterpillars, goats, and cattle ¹⁰. Despite having bisexual flowers, baobabs are mostly self-incompatible, depending on external pollinators for successful fertilization ^{11,12}. In natural populations, *A. digitata* is primarily pollinated by bats ^{13,14}, with occasional visits by bushbabies ¹⁵, and hawkmoths in southern Africa ¹². Since *A. digitata* is an obligate outcrosser, the populations harbor a high level of diversity, which is observed as heterozygosity in the genome (Fig. 1a; Table 1) ^{12,16}. Understanding the ploidy of *A. digitata* is essential for breeding and genetic improvement programs, as polyploidy affects traits like growth rate and environmental adaptability. This understanding can help overcome the species' slow maturation period of 8 to 23 years and aid in its domestication. Additionally, measuring genome size across different baobab species provides valuable evolutionary insights and aids in designing genomic sequencing projects ^{17,18,19}.

Page 3 line 42 to Page 4 line 17: (see the comment to reviewer#2)

The results from this research should be of the interest of wider scientific community, however, sometimes I felt lost such a high number presented results in Table and six figures. Would it be possible to consolidate result explain them specially in much understandable manner for wider audience even without the deep experience in this specific topic. I just feel overwhelmed with such a high number of highly specific data results. However the implication of the results are very well enlighten in the discussion part

We have revised the results to make them clearer and more accessible to a wider audience, while still including detailed information for experts. Some of the novel discoveries are illustrated in graphical format (**Response Fig. 5**).

Response Fig. 5: Insights into transposable elements (TEs) in baobab. **a** Barplot comparison of TEs classification and sizes in five baobab genomes: Ad77271a, Ad77271b, Aza135, AdKB, and AdOHT. Aza135 is highlighted in the center to emphasize TE divergence. Except for Aza135 (*A. za*), the other four genomes belong to *A. digitata*. Distinct colors in the legend denote TE classes. The TE proportion to genome size (Mb) and percentage are displayed on the left and right y-axes, respectively. **b** Density plots of intact TEs, displaying the distribution of different types of TEs. Mutator transposons showed a burst around 11 million years ago. **c** Accumulation of DNA mutators in the centromeric regions of the Ad77271a chromosome 3. Four mutator Terminal Inverted Repeat (TIR) transposons (TE_00000631: Mutator631, TE_00000927: Mutator927, TE_00000845: Mutator845, and TE_00000967: Mutator967) with a potential role in centromere creation or repositioning are displayed. **d** High solo to intact LTR ratio in Ad77271a chromosomes shows an aggressive purging mechanism following WGD, contributing to a smaller genome size (659 Mb).

Reference:

Klein, S. J. & O'Neill, R. J. Transposable elements: genome innovation, chromosome diversity, and centromere conflict. *Chromosome Res.* **26**, 5–23 (2018).

#####

#####

Reviewer #4 (Remarks to the Author):

I think most of my concerns have been well addressed by the authors and I'm particularly happy to see that additional citations and discussions are added appropriately. This manuscript deserves to be published yet some remaining concerns need further illustrations.

Thank you for your positive feedback; we are glad that the revisions have addressed most of your concerns. We hope that the responses below clarify the remaining issues. We appreciate the time you have taken to critically evaluate our manuscript.

1. Regarding the WGM history, it becomes much easier to understand the 4:1 ratio between baobab and cacao in Fig. 5 and Supplementary Fig. 8. But I am confused by the 4:2 ratio with grape and the 6:1 ratio with Amborella. I noticed that the authors have removed the related sentences from the main text. Please can the authors provide possible explanations on the ratios?

Syntenic depth ratios are influenced by the stringency of cutoff thresholds applied during analysis (number of genes in a block, block size and ortholog cutoff), making these ratios approximate rather than fixed. Aiming to filter out orthologs resulting from shared WGM, we included only reciprocal best hits; using --cscore of 0.70, for example, shifts 4:2 to 5:2 for grape. Similarly, for amborella, what appears as a 6:1 ratio could range from 5:1 to 8:2, depending on the threshold used. In addition, the syntenic depth reported in the figures is an arbitrary cutoff in McScan. For instance, if you look at Ad77271a vs. amborella (using the parameters we used in the manuscript), there is a syntenic depth of 8 (three additional depths in black) but McScan calls it 5 (in red) and we see a similar situation in Aza135, ceiba and bombax (**Response Figure 1E, F, G, H**). The situation is more complex in comparison with grape where we expect a 5:1 ratio because they share the whole genome triplication (WGT) and then baobab has the 5x WGM event(s). However, in most genomes we analyze that share the WGT with grape, they have a grape syntenic depth of 2 instead of 1, including Ad77271, ceiba and bombax (**Response Figure 1A, B, C, D**); this is also true of recent haplotype resolved versions of grape. In general, we use these syntenic depth plots as directional since both biological as well as experimental cutoffs influence the ratio. The decision to remove related sentences was made to minimize potential confusion.

Response Fig. 1. Syntenic depth between Ad77271a, Aza135, ceiba and bombax versus grape and amborella. Syntenic depth between A) Ad77271a vs. grape; B) Aza135 vs. grape; C) ceiba (Cpen) vs. grape; D) bombax (Bcei) vs. grape; A) Ad77271a vs. amborella; B) Aza135 vs. amborella; C) ceiba (Cpen) vs. amborella; D) bombax (Bcei) vs. amborella.

2. As to Response Table 1, baobab and B. ceiba share a quite similar pattern of copy number of the circadian, flowering, and light genes. I think this could be appropriately interpreted as the feature of subfamily Bombacoideae rather than specific to baobab lineage (Line 491).

We agree that these patterns likely reflect a characteristic feature of the Bombacoideae subfamily rather than being exclusive to the baobab lineage. Gene copy number commonalities suggest that the retention and expansion of these gene families may have played a crucial role in the adaptation of Bombacoideae species to their environments. We have added a sentence stating that these features are indicative of broader subfamily-level evolutionary processes.

Ln 494 to Ln 499: “Across most plant genomes analyzed, circadian, flowering and light-related genes are reduced back to one or two copies in the genome during fractionation, presumably to ensure the correct gene dosage (Cheng et al. 2018, Raju, S.K.K. 2020, and Michael, T.P. 2022). However, in Bombacoideae subfamily, with baobab as an example, we found that among circadian, flowering and light orthologs, only six (out of 35) orthogroups had one gene copy, while more than half (18/35) contained three or four copies (Supplementary Table 11; Supplementary Table 15; Supplementary Table 7)”.

References:

Cheng, F. *et al.* Gene retention, fractionation and subgenome differences in polyploid plants. *Nat Plants* **4**, 258–268 (2018).

Raju, S. K. K. Gene Dosage Balance Immediately following Whole-Genome Duplication in *Arabidopsis*. *The Plant cell* vol. 32 1344–1345 (2020).

Michael, T. P. Core circadian clock and light signaling genes brought into genetic linkage across the green lineage. *Plant Physiol.* **190**, 1037–1056 (2022).

3. The authors have clarified the relationship of the two sibling haploids. However, I am still not so confident about the conclusions basing on comparison between these two siblings, especially the estimated time of the autotetraploid event. I would prefer a clear definition about the Ks peak at 4 MYA between the two siblings, which may be neither a speciation peak nor an autotetraploidization peak. Despite the claims on dating of the autotetraploid event have been currently toned down, I do hope some words could be spend on highlighting the probable limitation or potential bias in the main text.

We have softened our claims, presenting this as a hypothesis rather than a definitive conclusion. The Ks peak at 4 MYA, as noted, may not correspond directly to a speciation or autotetraploidization event. Instead, we propose that it could also represent an indirect signal reflecting complex evolutionary dynamics influenced by gene flow or incomplete lineage sorting. By framing our findings as part of a broader hypothesis, we aim to encourage further research

and provide a balanced perspective on the polyploidy event's timing and its implications for the evolutionary history of baobabs.

Ln 451 to Ln 466: “The self-self alignments (paralogs) of genomes provided insights into the timing of WGM events. This analysis clarified that all of the baobab, bombax and cotton genomes experienced WGM events around 30 MYA (Conover et al. 2019), while cacao underwent its last WGM approximately 118 MYA, consistent with it only experiencing the WGT- γ event, as previously reported (Fig. 5d) (Argout et al. 2010). Focusing on the baobab complex, we observed a minor Ks peak around 30 MYA across all genomes, along with additional peaks at 4, 6, 11 and 17 MYA for Ad77271b, AdKB, AdOHT and Aza135 respectively (Supplementary Fig. 6d). We hypothesize that these peaks represent speciation and polyploidization events. The recent WGM in *A. digitata* resulted in an autotetraploid genome, as further evidenced by minor allele frequency coverage showing two peaks, indicating a chromosomal cluster from the two pairs of homologous chromosomes (four subgenomes) (Fig. 1d). This WGM in *A. digitata* is estimated to have occurred between 4 to 11 MYA, with additional evidence provided by a burst of TEs and time-calibrated gene tree (Fig. 3b; Supplementary Fig. 10).”

References:

Conover, J. L. et al. A Malvaceae mystery: A mallow maelstrom of genome multiplications and maybe misleading methods? *J. Integr. Plant Biol.* **61**, 12–31 (2019).

Argout, X. et al. The genome of *Theobroma cacao*. *Nat. Genet.* **43**, 101–108 (2010).

4. The authors' explanation on the large estimated genome size of *A. grandidieri* seems probable but not so reliable given the relatively low heterozygosity of *A. grandidieri*. Moreover, it is surprise to see that historical hybridization can lead to such a various genome size. Is it possibly attributed to a special sample, some unexpected methodological problems or some unique genomic features? I suggest that the present results of genome size of diploid baobabs here should be interpreted very cautiously.

We have seen this in other hybrids. For instance in the duckweed hybrid *Lemna japonica* (*L. minor* x *L. turionifera*) the genome size is double that expected for the haploid genome size we estimated using flow cytometry, and the heterozygosity is very low. At first (before we knew it was a hybrid), we assumed it was a polyploid due to the doubled size and low heterozygosity. However, after we sequenced the genome we determined it was a hybrid between *L. minor* x *L. turionifera*, which we also validated using GISH (**Response Figure 2**). We have now seen this across several hybrids and can also recapitulate this by mixing reads from haploids to make a synthetic hybrid sequence. However, this does not mean that this is what happened to the *A. grandidieri* or *A. rubrostipa* samples we tested. We acknowledge potential influences from sample-specific factors, methodological artifacts, or unique genomic features and have carefully noted these complexities and the need for further validation in the manuscript.

Ln 152 to Ln 165: “Feulgen staining and flow cytometry have been used historically to estimate genome size (Henniges et al. 2013). Using cytological methods, it was suggested that *A. digitata* has 42 chromosomes with a haploid genome size of approximately 920 megabases (Mb) and a 2C-DNA value of 3.8 pg (Islam-Faridi et al. 2020). Here, we skim sequenced all eight recognized baobab species and employed K-mer frequency analysis to estimate the

genome sizes of *A. digitata*, *A. madagascariensis*, *A. perrieri*, *A. za*, *A. gregorii*, *A. grandidieri*, *A. rubrostipa*, and *A. suarezensis*. Notably, the genome sizes ranged from 646 megabytes (Mb) in *A. perrieri* to 1.5 gigabytes (Gb) in *A. grandidieri*. The genome size estimates for both *A. grandidieri* and *A. rubrostipa* were double the predicted haploid genome size, and both had a single K-mer frequency peak consistent with these samples representing hybrids between two species, as has been reported between *A. rubrostipa* and *A. za* (Wan et al. 2024). However, these genome sizes could be influenced by sampling, methodological artifacts, or unique genomic features and thus require alternative approaches for validation. The K-mer based genome size estimates were overall consistent with the previous estimates (Woods et al., 2023, Wan et al., 2024; Islam-Faridi et al., 2020; and Leong et al., 2013).”

References:

- Woods, S., O’Neill, K. & Pirro, S. The Complete Genome Sequence of (Malvaceae, Malvales), the African Baobab. *Biodivers Genomes* **2023**, (2023).
- Wan, J.-N. *et al.* The rise of baobab trees in Madagascar. *Nature* **629**, 1091–1099 (2024).
- Islam-Faridi, N., Sakhanokho, H. F. & Dana Nelson, C. New chromosome number and cyto-molecular characterization of the African Baobab (*Adansonia digitata* L.) - ‘The Tree of Life’. *Sci. Rep.* **10**, 13174 (2020).
- Henniges, M. C. *et al.* The Plant DNA C-Values Database: A One-Stop Shop for Plant Genome Size Data. *Methods Mol. Biol.* **2703**, 111–122 (2023).
- Leong Pock Tsy, J.-M. *et al.* Nuclear microsatellite variation in Malagasy baobabs (*Adansonia*, Bombacoideae, Malvaceae) reveals past hybridization and introgression. *Ann. Bot.* **112**, 1759–1773 (2013).

Response Fig. 2. *Lemna japonica* (*L. minor* x *L. turionifera*) hybrid K-mer profile is double the haploid genome size and has low heterozygosity. A) The *Lemna japonica* line 8434 genome size is estimated at 734 Mb with a heterozygosity of 0.0691% by K-mer frequency with Illumina reads and visualized with GenomeScope. The actual assembled genome size is 399 Mb for *L. turionifera* haplotype and 340 Mb for the *L. minor* haplotype. B) Genomic in situ hybridization (GISH) using probes against both the *L. minor* and *L. turionifera* genomes showing that *Lemna japonica* line 8434 is in fact a hybrid.

5. In Supplementary Fig. 11, it is a big surprise to see such a “complex” collinearity between Ad77271a and Aza135.

We were actually surprised there wasn't more variation between the *Adansonia* species since these species are also separated geographically by an Ocean (*A. digitata* in Africa and *A. za madagascar*). We have added a sentence to clarify the observed collinearity.

Ln 444 to Ln 446: “Following the split, Aza135 experienced chromosome rearrangements, resulting in smaller chromosomes (Supplementary Fig. 11).”

Ln 691 to Ln 708: “Finally, the resequencing of 25 *A. digitata* accessions from across Africa revealed that *Adansonia* trees situated north of the equator exhibited some divergence from those located southward, with approximately 6% of the variation attributed to this distinction. However, the most partitioning, about 30% of the variation, occurred along an east-west axis as exemplified by the distinctiveness between populations 1 and 2 (Fig. 6; Supplementary Fig. 16). Intriguingly, Namibian baobab trees clustered into two populations (2 and 3), which correspond to different watersheds, suggesting limited gene flow due to possible geographic barriers and adaptation to distinct climate regimes (Leong et al., 2013). Another study conducted in Niger and Mali supported this notion, indicating variations in baobab species across the continent; specifically, it found that West African germplasms exhibited faster growth and better adaptation to arid environments compared to their East African counterparts (Korbo et al., 2011). A recent study on the evolutionary history of baobab trees suggests that a diploid progenitor dispersed from Madagascar to Africa, where autoploidy occurred in West Africa, creating the current tetraploid *A. digitata*. This neo-polyploidy, with its ecological advantages, likely contributed to *A. digitata*'s spread eastwards and southwards (Wan et al., 2024). Additionally, the observed complex collinearity between Ad77271a from mainland Africa and Aza135 from Madagascar

underscores the extent of chromosomal restructuring and divergence within the species (Supplementary Fig. 11).”

References:

- Leong Pock Tsy, J.-M. *et al.* Nuclear microsatellite variation in Malagasy baobabs (*Adansonia*, Bombacoideae, Malvaceae) reveals past hybridization and introgression. *Ann. Bot.* **112**, 1759–1773 (2013).
- Korbo, A. *et al.* Comparison of East and West African populations of baobab (*Adansonia digitata* L.). *Agrofor. Syst.* **85**, 505–518 (2011).
- Wan, J.-N. *et al.* The rise of baobab trees in Madagascar. *Nature* **629**, 1091–1099 (2024).

#####

Second response to reviewers

Reviewer #4 (Remarks to the Author):

The authors have made relatively good responses to the concerns raised by Referee #1, point by point. However, some results still look highly confusing which need more illustrations.

1. As shown in Responding Fig. 1, two WGD events were inferred in Ad77271a, one is ancient and the other is autotetraploidy which belongs to baobab lineage. If Ad77271a genome is a haploid (i.e. subgenome), it would be surprise to observe a syntenic depth ratio between Ad77271a and cacao as 4:1, comparing to a ratio between Ad77271a and grape as 4:2, as shown in Responding Fig. 2 & 10. Does the ancient “WGD” here represent an octoploidization event?

Response: Thank you for your feedback and suggestions. We addressed each below and hope we have clarified issues and provided additional support when needed. Particularly the papers you mentioned were helpful: Shao et al., 2024 (<https://doi.org/10.1016/j.xplc.2024.100832>) and Sun et al., 2024 (<https://doi.org/10.1073/pnas.2313921121>), which were published just before and after the submission of this manuscript respectively. Shao et al., provides a higher quality genome for bombax that corroborates the dating for the whole genome duplication (WGD) shared between cotton and baobab, and they refer to this event as a WGD in the text and a WGP (whole genome polyploidy) in the figures. Sun et al. 2024, further show that the WGD shared between baobab and cotton is a decaployploid event resulting from successive allotetraploid and allohexaploid events that occurred close together. The Paterson group postulated that cotton experienced a decaployploid event (Paterson et al., 2012; Wang et al., 2016), although since we were just seeing a 4:1 ratio with cacao, we chose to call this event more generically a WGD event similar to Shao et al., 2024. The higher quality bombax (Shao et al., 2024) and balsa (Sun et al., 2024) show a clear 5:1 syntenic depth with cacao, compared to the 4:1 syntenic depth we observe between baobab and cacao; although 6% have a syntenic depth of 5 consistent with a loss of the 5th copy in baobab compared to balsa and bombax (Supplementary Fig. 8g). We have updated the results to include references to these publications and we have changed “WGD” to “WGM” (whole-genome multiplication), which is the language used by Sun et al 2024.; we have also explained the syntenic depth a bit more including that baobab has a 5:1 syntenic depth if you include the 6% in five copies. In addition, we have updated the discussion with a brief explanation of the WGM history. **Ln. 609 to Ln.**

624: “Baobab belongs to the Malvaceae family, which comprises 4,225 species across 243 genera and nine subfamilies, including economically and ecologically important plants such as cotton (*Gossypium spp.*), the cotton tree (*Bombax ceiba*), cacao (*Theobroma cacao*), and baobab (*Adansonia digitata*) (Sun et al.,2024). WGM have played a crucial role in the evolution of these plants. For example, following the ancient core eudicot whole-genome triplication (γ) event, cotton plants have undergone repeated polyploidization and hybridization, resulting in polyploid offspring with combined genomes that diverge significantly from their diploid parents, thereby initiating new evolutionary trajectories. The decaploid ancestor of cotton (a 5 \times multiplication) significantly contributed to the development of spinnable fibers (Sun et al.,2024, Wan et al.,2024, Paterson et al.,2012). Similarly, polyploidy in *Bombax ceiba* has facilitated its robust growth and vibrant blooms in tropical regions (Shao et al.,2024), while ancient WGM in cacao have led to the emergence of diverse and flavorful varieties. Here, we demonstrate that baobab has a unique proportion of DNA transposons, experienced WGM and present a high-quality chromosome-level haploid assembly of *Adansonia digitata* (Ad77271a) alongside draft genomes from a sibling tree (Ad77271b), two geographically diverse *A. digitata* (AdOHT and AdKB) and distinct species *A. za* (Aza135) from Madagascar. We also show a distinct baobab population in Namibia.”

References:

Shao L, Jin S, Chen J, Yang G, Fan R, et al. High-quality genomes of *Bombax ceiba* and *Ceiba pentandra* provide insights into the evolution of Malvaceae species and differences in their natural fiber development. *Plant Commun.* 2024 May 13;5(5):100832. doi: 10.1016/j.xplc.2024.100832. Epub 2024 Feb 5. PMID: 38321741; PMCID: PMC11121743.

Sun, P. et al. Subgenome-aware analyses reveal the genomic consequences of ancient allopolyploid hybridizations throughout the cotton family. *Proc. Natl. Acad. Sci. U. S. A.* 121, e2313921121 (2024).

Wang X, Guo H, Wang J, Lei T, Liu T, Wang Z, Li Y, Lee TH, Li J, Tang H, Jin D, Paterson AH. Comparative genomic de-convolution of the cotton genome revealed a decaploid ancestor and widespread chromosomal fractionation. *New Phytol.* 2016 Feb;209(3):1252-63. doi: 10.1111/nph.13689. Epub 2015 Oct 7. PMID: 26756535.

Paterson AH, Wendel JF, Gundlach H, Guo H, Jenkins J, Jin D, Llewellyn D, Showmaker KC, Shu S, Udall J, Yoo MJ, Byers R, Chen W, Doron-Faigenboim A, Duke MV, Gong L, Grimwood J, Grover C, Grupp K, Hu G, Lee TH, Li J, Lin L, Liu T, Marler BS, Page JT, Roberts AW, Romanel E, Sanders WS, Szadkowski E, Tan X, Tang H, Xu C, Wang J, Wang Z, Zhang D, Zhang L, Ashrafi H, Bedon F, Bowers JE, Brubaker CL, Chee PW, Das S, Gingle AR, Haigler CH, Harker D, Hoffmann LV, Hovav R, Jones DC, Lemke C, Mansoor S, ur Rahman M, Rainville LN, Rambani A, Reddy UK, Rong JK, Saranga Y, Scheffler BE, Scheffler JA, Stelly DM, Triplett BA, Van Deynze A, Vaslin MF, Waghmare VN, Walford SA, Wright RJ, Zaki EA, Zhang T, Dennis ES, Mayer KF, Peterson DG, Rokhsar DS, Wang X, Schmutz J. Repeated polyploidization of *Gossypium* genomes and the evolution of spinnable cotton fibres. *Nature.* 2012 Dec 20;492(7429):423-7. doi: 10.1038/nature11798. PMID: 23257886.

2. I also concern about the issue of gene loss as Referee 1# raised. The authors have not fixed it very well. Why is the original gene copy number four in baobabs? If there is only one ancient

tetraploidization event, the original gene copy number in a haploid genome should be two but not four. The authors did not provide additional evidence for the claim about “most genomes fractionate gene copies back to a diploid state after WGD” in the main text or “some genes are lost” in the rebuttal. If the authors want to highlight the functional importance of “clock/flowering/temperature genes are retained in four copies”, I would suggest that they should provide comparisons of gene copy with more species (e.g. cottons and *B. ceiba*). Recent studies suggested a decaploidization event in Malvaceae. If so, the syntenic depth and gene loss here need more discussions.

Response: We have changed WGD to WGM in the text to reflect the language used in Sun et al., and added text that this WGM is a past decaploidization. We agree that it would strengthen our argument to show a comparison of gene copies for the circadian/flowering/temperature genes so we have added a table to the supplement (Response Table 1) and added sentences to the text. **Ln 627 to Ln 631:** “Baobab has undergone several round WGM starting with the WGT-γ followed by what is thought to be a decaploidization (Sun et al., 2024, wang et. al. 2016, Paterson et al., 2012 & Shao et al., 2024,) the latter of which has remained largely unfractionated with between four and five full chromosomes retained as compared to cacao (Fig. 5a; Supplementary Fig. 8g; Supplementary Fig. 12).” Also **Ln 461 to Ln 476**

	OG	Ad77271a	B. ceiba	G. raimondii	A. thaliana	V. vinifera	Q. rubra	gene	type
1	OG0002676	3	3	3	2	1	1	CCA1/LHY	clock
2	OG0002990	4	2	2	4	1	2	RVE1/RVE2/RVE7	clock
3	OG0000823	5	6	4	3	2	2	RVE3/RVE5/RVE6	clock
4	OG0008057	3	3	1	2	1	1	RVE4/RVE8	clock
5	OG0006813	4	4	1	2	1	1	PRR9	clock
6	OG0000842	7	6	4	2	3	3	PRR3/7	clock
7	OG0007965	1	1	3	1	1	1	PRR1	clock
8	OG0005091	3	2	3	1	1	1	PRR5	clock
9	OG0004177	4	4	2	2	2	1	ZTL/LKP2	clock
10	OG0008873	2	2	2	1	1	1	FKF1	clock
11	OG0001524	4	4	5	2	2	2	ELF3	clock
12	OG0002940	2	3	5	2	2	3	ELF4	clock
13	OG0002858	3	4	3	2	1	1	LUX	clock
14	OG0001908	4	6	3	1	1	1	LNK1	clock
15	OG0003471	3	4	1	1	1	1	LNK2	clock
16	OG0003692	2	3	2	1	1	1	GI	clock
17	OG0005135	3	3	1	1	1	1	FCA	flowering
18	OG0002762	8	3	2	3	1	1	VIN3/VEL1/VEL	flowering
19	OG0003516	2	4	3	2	2	2	FVE	flowering
20	OG0008018	2	3	1	1	1	1	TFL2	flowering
21	OG0005608	3	7	1	3	1	2	CO	flowering
22	OG0009853	2	1	1	1	1	1	LD	flowering
23	OG0007488	2	1	2	1	1	2	FRI	flowering
24	OG0000318	6	8	6	5	3	4	FT	flowering
25	OG0009131	2	1	1	1	1	1	DET1	light
26	OG0009581	1	1	2	1	1	1	PHYA	light
27	OG0003293	4	3	2	3	2	2	PHYB/PHYD/PHYE	light
28	OG0010520	1	1	1	1	1	1	PHYC	light
29	OG0003899	2	4	2	1	2	3	COP1	light
30	OG0008582	1	2	2	1	1	1	CRY1	light
31	OG0004498	1	2	2	1	1	1	CRY2	light
32	OG0002580	4	4	2	2	2	2	SPA1	light
33	OG0002519	3	3	1	2	1	1	HYS/HYH	light
34	OG0003669	4	3	1	1	2	1	PIF3	light
35	OG0007706	1	2	1	2	1	1	PIF4/PIL6	light
	TOTAL	106	113	78	62	48	52		

Response Table 1. Circadian, flowering, and light ortholog copy numbers.

References:

Sun, P. et al. Subgenome-aware analyses reveal the genomic consequences of ancient allopolyploid hybridizations throughout the cotton family. *Proc. Natl. Acad. Sci. U. S. A.* 121, e2313921121 (2024).

Wang X, Guo H, Wang J, Lei T, Liu T, Wang Z, Li Y, Lee TH, Li J, Tang H, Jin D, Paterson AH. Comparative genomic de-convolution of the cotton genome revealed a decaploid ancestor and widespread chromosomal fractionation. *New Phytol.* 2016 Feb;209(3):1252-63. doi: 10.1111/nph.13689. Epub 2015 Oct 7. PMID: 26756535.

Paterson AH, Wendel JF, Gundlach H, Guo H, Jenkins J, Jin D, Llewellyn D, Showmaker KC, Shu S, Udall J, Yoo MJ, Byers R, Chen W, Doron-Faigenboim A, Duke MV, Gong L, Grimwood J, Grover C, Grupp K, Hu G, Lee TH, Li J, Lin L, Liu T, Marler BS, Page JT, Roberts AW, Romanel E, Sanders WS, Szadkowski E, Tan X, Tang H, Xu C, Wang J, Wang Z, Zhang D, Zhang L, Ashrafi H, Bedon F, Bowers JE, Brubaker CL, Chee PW, Das S, Gingle AR, Haigler CH, Harker D, Hoffmann LV, Hovav R, Jones DC, Lemke C, Mansoor S, ur Rahman M, Rainville LN, Rambani A, Reddy UK, Rong JK, Saranga Y, Scheffler BE, Scheffler JA, Stelly DM, Triplett BA, Van Deynze A, Vaslin MF, Waghmare VN, Walford SA, Wright RJ, Zaki EA, Zhang T, Dennis ES, Mayer KF, Peterson DG, Rokhsar DS, Wang X, Schmutz J. Repeated polyploidization of *Gossypium* genomes and the evolution of spinnable cotton fibres. *Nature.* 2012 Dec 20;492(7429):423-7. doi: 10.1038/nature11798. PMID: 23257886.

Shao L, Jin S, Chen J, Yang G, Fan R, et al. High-quality genomes of *Bombax ceiba* and *Ceiba pentandra* provide insights into the evolution of Malvaceae species and differences in their natural fiber development. *Plant Commun.* 2024 May 13;5(5):100832. doi: 10.1016/j.xplc.2024.100832. Epub 2024 Feb 5. PMID: 38321741; PMCID: PMC1112174

3. The authors claimed that “the haploid representation most likely contains a combination of non-phased variation from the four subgenomes” (I think they are indeed be “two subgenomes”) in the rebuttal. If the genome is non-phased, it is extremely hard to assign the Ks rates to the speciation peaks. The authors’ response is ok for a normal diploid genome. But it could be a different situation for a non-phased haploid genome of an autotetraploid species. My essential argument is that the genome of the two siblings can’t be simply treated as two subgenomes. The siblings only represent combinations of random chromosomes or contigs, and the recombination between homeologs can’t be overlooked. The potential artefacts caused by autopolyploid can lead to complex result of assembly. Currently, the assembly of the autopolyploid baobab genome is still technologically treated as a normal one of a diploid organism.

Response: A haploid representation of an autotetraploid is one copy of the four underlying genomes. Even if we had resolved the genome into four haplotypes, they still would not have been phased since we do not have parental information; the best case is that we would have them phased by chromosome, but each chromosome set would represent unphased genomes. To date, almost all diploid genomes published are non-phased haploid representations in plants; only in the past several years are we seeing haplotyped genomes and even fewer fully phased genomes. The unphased haploid genomes have been used to assign Ks rates for speciation peaks. Using the un-phased haploid Ad77271a assembly we accurately recapitulated the baobab-amborella and baobab-grape peaks, suggesting that this method does work on unphased haploid genomes as has been seen previously. It is true we cannot treat the sibling as another haplotype, but the sibling is a powerful tool to estimate the differences in alleles. In a later comment we show that only 14% of orthologs between the siblings are the “same,” (or

have zero variation) suggesting that we do have decent representation of allele diversity between these two siblings. We further address the potential for artifacts in another response for which we still have been unable to identify any glaring artifacts. It should be noted that the single K-mer peak for *A. digitata* suggests that the underlying genomes of baobab are not that complex with a haploid genome heterozygosity below 1% (0.65%). We additionally address the use of the haploid representation in another comment below on Fst.

Several conclusions from this manuscript remained highly controversial to me,

1. Line329. Are there any impacts of assembly artefacts on the number and distribution of TEs? Similar concerns are remained on the number of genes, gene expansion, and patterns of centromere and telomere.

Response: Please see our response to reviewer #2 in our previous response: “All the measures that we have tried (coverage, BUSCO, k-mer, synteny, etc) suggest that the haploid representation is a high quality, non-collapsed version of the genome.” To summarize, coverage analysis suggests the assembly doesn’t have issues across different features such as repeats and genes. BUSCO scores suggest the gene space is complete so we are not missing genes and the duplication rate is consistent with a haploid genome that doesn’t have unresolved haplotypes integrated. The centromeres are on the megabase scale suggesting they are complete or near complete; but we should point out very few genomes to date are checked for centromere completeness. The telomeres assembled well and are long. However, the best piece of evidence supporting a high quality assembly is the collinearity with all the genomes we present in the manuscript. The key is the synteny and orthologs with cacao; 93% of Ad77271a genes are collinear/syntenic with cacao (Figure 5a; Supplementary Fig. 8). We did not highlight it in our previous response but the HiC contact map (Fig. 1c) also provides high quality support for the genome structure. We cannot find any evidence of assembly artifacts including misassemblies, gene expansions/contractions, TE expansions/contractions, nor issues with the centromeres/telomeres. Of course no genome assembly is without some type of artifacts (until we can generate full chromosomes with single molecule reads), but based on standard measure the Ad77271a haploid genome looks good.

2. Line335. Is it possible that the two peaks of TEs represent homologs (later) and homeologs (earlier) respectively?

Response: The earlier striking peak around 11 million years ago (MYA) could represent homeologs resulting from a WGM event and corresponds to the interval of autotetraploidization in *A. digitata*. The later peak, on the other hand, might represent remnants of TEs, with only specific elements like CACTA and Copia experiencing bursts of activity. This peak could also be related to more recent duplications, which suggests that homologous genes have been created through segmental or tandem duplications. These more recent WGM might not be as widespread as those from the ancient WGM event, but they still contribute to the genomic landscape observed today.

3. Line436. It may not be a good idea to infer the ancient WGD history in Malvaceae by using of the autopolyploid genomes.

Response: We did not use autopolyploid genomes for WGD history inference. Our comparative genomics study and inference of WGD were conducted on baobab genomes alongside Malvaceae species such as cotton (*Gossypium raimondii*), bombax (*Bombax ceiba*), and cacao

(*Theobroma cacao*). Additionally, we included grape (*Vitis vinifera*), which has undergone only core eudicot whole genome triplication (WGT), and Amborella (*Amborella trichopoda*), which is sister to the eudicot lineage and lacks a whole genome duplication (WGD) event.

4. Line461. The timing of autotetraploidy event seems considerably underestimated given only parts of the sibling genomes are homeologs.

Response: Looking at the Ka/Ks results, only 14% of comparisons between Ad77271a and Ad77271b have zero values, i.e. these are probably the same copy of the homeolog/homolog. Therefore, these results are consistent with most comparisons representing homeologs/homologs. In contrast, the Ka/Ks results between AdKB and AdOHT resulted in 2 and 6% zero values, which we would expect all or most of these are homeologs/homologs. The divergence dates for AdKB and AdOHT are 6 MYA and 11 MYA respectively. We have revised our text to include this nuance of the data that it could be between 3-11 MYA to account for possible underestimation.

5. Line533. It is worth noting that Fst is a population genetic estimation based on the model for diploids. Polyploid Fst should be cautiously interpreted.

I do agree with that assembly of the “true subgenome” is of great challenges. However, the relevant conclusions in the manuscript are doubtful before new analyses specific to autopolyploid (not affected by polyploidy or homeologs) are introduced.

Response: Fst is a population-level metric that operates on the number of alleles in a given state. Yes, the given expectation depends on ploidy, but it's constant; thus like the difference in all the Fst estimators we've compared, the question isn't that they are liable to give erroneous results vs. one another, it's just that they each scale a bit differently. We have carried out these experiments before with results that have held up under reanalysis (Monnahan et al., 2019). However, we agree Fst should be used with caution, and is particularly problematic when contrasting values generated in comparisons of different ploidies (Meirmans et al., 2018); however, this estimator can be safely used on within-cytotype variation. We have added the following to the discussion. **Ln 642 to Ln 644:** “We note that Fst is a population genetic estimation based on a diploid model. Thus we interpret our polyploid Fst results cautiously.”

References:

Monnahan P, Kolář F, Baduel P, Sailer C, Koch J, Horvath R, Laenen B, Schmickl R, Paajanen P, Šrámková G, Bohutínská M, Arnold B, Weisman CM, Marhold K, Slotte T, Bomblies K, Yant L. Pervasive population genomic consequences of genome duplication in *Arabidopsis arenosa*. *Nat Ecol Evol.* 2019 Mar;3(3):457-468. doi: 10.1038/s41559-019-0807-4. Epub 2019 Feb 25. PMID: 30804518.

Meirmans PG, Liu S, van Tienderen PH. The Analysis of Polyploid Genetic Data. *J Hered.* 2018 Mar 16;109(3):283-296. doi: 10.1093/jhered/esy006. PMID: 29385510.

Besides, I am struggling about some conflicting interpretation of results here, Basing on resequencing data, the author claimed that some genomic signals, such as gene dosage of some pathway and centromere regions are “highly dynamic and location specific” among populations within the baobab species. If so, the previous conclusions on these genomic signals by comparing the baobab genome with other species seem unreliable. For example, if

the centromere regions are highly dynamic, is it possible that the conclusion on centromeres (Line274-288) is just a description on one population or individual, instead of the species?

Response: All of the centromere regions were similar (same base repeat and size) across the four baobab genomes that were sequenced so we have no expectation that Ad77271a is different from all other baobabs. We also analyzed the genomes in the Wan et al. (2024) and found the same centromere base repeat and they assembled a similar amount of centromere sequence. It is known from Arabidopsis work (as well as other species with highly resolved centromeres) that centromere sequences can be divergent in terms of copy number and organization across different accessions/landraces (Wlodzimierz et al., 2023).

Reference:

Wlodzimierz P, Rabanal FA, Burns R, Naish M, Primetis E, Scott A, Mandáková T, Gorringer N, Tock AJ, Holland D, Fritschi K, Habring A, Lanz C, Patel C, Schlegel T, Collenberg M, Mielke M, Nordborg M, Roux F, Shirsekar G, Alonso-Blanco C, Lysak MA, Novikova PY, Bousios A, Weigel D, Henderson IR. Cycles of satellite and transposon evolution in Arabidopsis centromeres. *Nature*. 2023 Jun;618(7965):557-565. doi: 10.1038/s41586-023-06062-z. Epub 2023 May 17. PMID: 37198485.

Wan, JN., Wang, SW., Leitch, A.R. et al. The rise of baobab trees in Madagascar. *Nature* 629, 1091–1099 (2024). <https://doi.org/10.1038/s41586-024-07447-4>

Does the retention of gene copies in several pathways contribute to local adaptation of populations or the common adaptation of the species?

Response: We are arguing here that the extra gene copies provide local adaptation, but the Fst analysis is at the population level, so it seems these factors are playing a role across the at least *A. digitata* in specific environments.

Similarly, I found that the Gypsy number in Table 2 is highly various among samples, even between the two siblings. There are two possible reasons: one is due to the highly dynamics, and the other is assembly artefacts.

Response: We agree that the observed differences could stem from two primary sources: assembly artifacts or the dynamic nature of Gypsy elements. We undertook additional analyses, as we explained to reviewer #2 in the previous response, to address possible assembly artifacts (also see above responses in regards to assembly artifacts). We checked coverage across the genome, as well as the LTR-RTs, and found that the coverage was consistent across genomes and LTR-RTs, providing additional support that these regions are not expanded/contracted or misassembled. We also tested TE completeness using the LTR Assembly Index (LAI). The LAI score categorizes the assemblies within the range of a "reference" assembly (Mokhtar et al., 2023). In terms of the dynamic nature of the Gypsy elements, it was found that they can cause the genome size of *C. pentandra* to be two-fold larger compared to *B. ceiba*, although both species share the same WGM event along with baobab (Shao et al., 2024). In addition, there is a great deal of variation (almost 2 fold) across all of the baobab genomes in terms of Gypsy element content (Wan et al., 2024), suggesting that Gypsy may be variable in the baobab lineage. Also, while the copy number is different, the overall percent "Masked LTR-RT TE" (Table 2) is similar across the baobab genomes we sequenced (10-14% for *A. digitata*, and 9% for *A. za*).

Reference:

Mokhtar MM, Abd-Elhalim HM, El Allali A. A large-scale assessment of the quality of plant genome assemblies using the LTR assembly index. *AoB Plants*. 2023 Apr 4;15(3):plad015. doi: 10.1093/aobpla/plad015. PMID: 37197714; PMCID: PMC10184434.

Shao L, Jin S, Chen J, Yang G, Fan R, et al. High-quality genomes of *Bombax ceiba* and *Ceiba pentandra* provide insights into the evolution of Malvaceae species and differences in their natural fiber development. *Plant Commun*. 2024 May 13;5(5):100832. doi: 10.1016/j.xplc.2024.100832. Epub 2024 Feb 5. PMID: 38321741; PMCID: PMC11121743.

Wan, JN., Wang, SW., Leitch, A.R. et al. The rise of baobab trees in Madagascar. *Nature* 629, 1091–1099 (2024). <https://doi.org/10.1038/s41586-024-07447-4>.

Other comments:

Line168. Why is the genome size of *A. grandidieri* and *A. rubrostipa* were much larger than the others given they are all diploid?

Response: The K-mer frequency genome size estimates for both *A. grandidieri* and *A. rubrostipa* are almost exactly double the haploid genome size for the baobabs and they only have one K-mer frequency peak consistent with a homozygous diploid genome. Based on the K-mer frequency profile for *A. digitata* (Fig. 1a), and our experience with other genomes with similar architectures (as noted to reviewer #2), this profile usually means these are hybrids or truly homozygous diploid genomes. Since the genomes are double the size and only have one peak, we hypothesize they are hybrid genomes. It has been shown that there are hybridization events between baobab species such as *A. rubrostipa* and *A. za* (Wan et al., 2024; Leong Pock et al., 2013). We have added a sentence to this effect in the results: **Ln 159 to Ln 162:** “The genome size estimates for both *A. grandidieri* and *A. rubrostipa* were double the predicted haploid genome size, and both had a single K-mer frequency peak consistent with these samples representing hybrids between two species as has been seen between *A. rubrostipa* and *A. za*”

Reference:

Wan, JN., Wang, SW., Leitch, A.R. et al. The rise of baobab trees in Madagascar. *Nature* 629, 1091–1099 (2024). <https://doi.org/10.1038/s41586-024-07447-4>.

Leong Pock Tsy JM, Lumaret R, Flaven-Noguier E, Sauve M, Dubois MP, Danthu P. Nuclear microsatellite variation in Malagasy baobabs (*Adansonia*, Bombacoideae, Malvaceae) reveals past hybridization and introgression. *Ann Bot*. 2013 Dec;112(9):1759-73. doi: 10.1093/aob/mct230. Epub 2013 Nov 1. PMID: 24187031; PMCID: PMC3838555.

Line174. Why is the heterozygosity of *A. digitata* accurate to two decimal places, but others are accurate to one decimal place? Is the conclusion “one of the highest heterozygosity” still correct if the data of *A. perrieri* and *A. za* were accurate to two decimal places as *A. digitata*?

Response: Corrected

Line220-228. I'm not so confident about the comparison initiated by the lower-quality assembly of Aza135. Is there any phenotypic difference between *A. digitata* and Aza135, which can be

explained by this translocation? The authors highlighted genes associated with DNA repair and longevity, isn't *A. za* a long-lived organism?

Response: *A. za* is considered long-living, with radiocarbon dating showing the largest *za* baobab to be 900 years old (Patrut et al. 2016). We have rephrased the sentence to avoid implying that the translocation is directly linked to longevity. Ecological valence analysis of *A. za* revealed it to be highly competitive for habitat space, leading to greater population expansion in Madagascar, similar to *A. digitata* in mainland Africa (Wan et al. 2024). Moreover, the study showed ongoing hybridization between *A. perrieri* and *A. za*. Further analysis suggested ancient introgression followed by significant recombination, which broke up introgressed segments as the polyploid genome diverged and developed different chromosomal structures.

Reference

Patrut A, Patrut RT, Danthu P, Leong Pock-Tsy J-M, Rakosy L, Lowy DA, et al. (2016) AMS Radiocarbon Dating of Large *Za* Baobabs (*Adansonia za*) of Madagascar. PLoS ONE 11(1): e0146977. <https://doi.org/10.1371/journal.pone.0146977>

Wan, JN., Wang, SW., Leitch, A.R. et al. The rise of baobab trees in Madagascar. Nature 629, 1091–1099 (2024). <https://doi.org/10.1038/s41586-024-07447-4>

Line297. What is the meaning of the number 54.74%, 54.94% and 62.52% shown here? Are they significantly larger than the number in other organisms? I realized that the detection method is based on deep learning, so the comparison will be reasonable if the average levels of DNA methylation in other organisms can be detected in the same way.

Response: The numbers 54.74%, 54.94%, and 62.52% represent the mean percentage of total genomic modified bases (5mC) detected using the Dorado basecaller (v.0.7.2) with "--pore r9.4.1". These methylation levels are high, as observed in Malvaceae genomes that experienced recent whole-genome duplication (Shao et al., 2024). In *G. raimondii*, 71% of CpG, 57% of CHG, and 13% of CHH are methylated, while in *T. cacao*, 44% of CpG, 15% of CHG, and 1% of CHH are methylated (genome-wide weighted detection using bisulfite methods). This methylation variation reflects the evolutionary and life histories of plant species in general (Niederhuth et al., 2016). The Oxford Nanopore Technologies (ONT) basecalling algorithms and several bioinformatic post-processing features have demonstrated higher accuracy compared to traditional bisulfite conversion methods (Ferguson et al., 2022). Note that in plants, 5mC occurs at CpG dinucleotides as well as CHG and CHH elements. Further characterization of methylation in baobab indicated hypermethylation in transposable elements contrasted with hypomethylation of genes. Moreover, we observed enhanced methylation in gene bodies and specific coding regions compared to intergenic regions (Supplementary Fig. 15).

Reference

Niederhuth CE, Bewick AJ, Ji L, Alabady MS, Kim KD, Li Q, Rohr NA, Rambani A, Burke JM, Udall JA, Egesi C, Schmutz J, Grimwood J, Jackson SA, Springer NM, Schmitz RJ. Widespread natural variation of DNA methylation within angiosperms. Genome Biol. 2016 Sep 27;17(1):194. doi: 10.1186/s13059-016-1059-0. PMID: 27671052; PMCID: PMC5037628.

Shao L, Jin S, Chen J, Yang G, Fan R, et al. High-quality genomes of *Bombax ceiba* and *Ceiba pentandra* provide insights into the evolution of Malvaceae species and differences in their

natural fiber development. *Plant Commun.* 2024 May 13;5(5):100832. doi: 10.1016/j.xplc.2024.100832. Epub 2024 Feb 5. PMID: 38321741; PMCID: PMC11121743.

Ferguson, S., McLay, T., Andrew, R.L. et al. Species-specific basecallers improve actual accuracy of nanopore sequencing in plants. *Plant Methods* 18, 137 (2022). <https://doi.org/10.1186/s13007-022-00971-2>

Line386. “baobab has some unique gene content” sounds meaningless since each species have its own unique gene content.

Response: Revised

Line435. Two literatures addressing the genomic evolution in Malvaceae have provided some new insights which deserved consideration since WGD is one of the most important issue dissected in this study.

<https://doi.org/10.1016/j.xplc.2024.100832>

<https://doi.org/10.1073/pnas.2313921121>

Response: Thank you for the papers. The Sun et al., 2024, which was published around when we submitted this work, does provide a nice explanation for the WGD/WGM event we identify in baobab, and as stated above we have modified our language (WGD vs WGM) in the manuscript to reflect their findings. Also, the Sun et al., and Shao et al., papers provide three new chromosome scale genomes that also show the retention of almost complete chromosomes post the WGM, supporting our finding as well.

Line444. I realized that the divergence time here is not so consistent with the time estimated using the calibrated tree (Supplementary Fig. 10). Which one is more reliable?

Response: We have revised the analysis to include additional genomes and removed *Gossypium hirsutum* as suggested. These dates match up with literature findings and analyses of synonymous substitutions per synonymous site (Ks). The baobab ancient WGM at 30 MYA, and the split between *A. digitata* and *A. za* happened around 20 MYA. Shao et al., 2024 also find a similar timing for the ancient WGM at about 30 MYA.

References

Wan, JN., Wang, SW., Leitch, A.R. et al. The rise of baobab trees in Madagascar. *Nature* 629, 1091–1099 (2024). <https://doi.org/10.1038/s41586-024-07447-4>.

Shao L, Jin S, Chen J, Yang G, Fan R, et al. High-quality genomes of *Bombax ceiba* and *Ceiba pentandra* provide insights into the evolution of Malvaceae species and differences in their natural fiber development. *Plant Commun.* 2024 May 13;5(5):100832. doi: 10.1016/j.xplc.2024.100832. Epub 2024 Feb 5. PMID: 38321741; PMCID: PMC11121743.

Line450. *G. hirsutum* is an allotetraploid cotton with AADD genome. It is not a good idea to use it for rebuilding of the gene-based species tree.

Response: We have reconstructed the gene-based species tree using *G. arboreum*, *G. raimondii*, and seven additional species instead.

Response Fig. 14: Time-calibrated phylogenetic tree of *Adansonia digitata* (Ad77271a), *Adansonia za* (Aza135), *Bombax ceiba*, *Gossypium raimondii*, *Gossypium arboreum*, *Theobroma cacao*, *Carica papaya*, *Arabidopsis thaliana*, and *Oryza sativa*. The phylogenetic analysis is based on the protein sequences of the 9 plants, with *O. sativa* as an outgroup. The timescale in million years ago (MYA) is provided at the bottom. *A. thaliana* to *O. sativa* (142.1–163.5 MYA) (<http://www.timetree.org/>) was used for calibration.

Line488. Is it possible that the high number of gene copies is just due to assembly artefacts?

Response: See previous response to genome artifacts; BUSCO, LAI, N50, and synteny support a high quality genome. Also, the recently released genomes from Shao et al., Sun et al., and Wan et al., support our genome structure and gene copy number.

References:

Wan, JN., Wang, SW., Leitch, A.R. et al. The rise of baobab trees in Madagascar. *Nature* 629, 1091–1099 (2024). <https://doi.org/10.1038/s41586-024-07447-4>

Shao L, Jin S, Chen J, Yang G, Fan R, et al. High-quality genomes of *Bombax ceiba* and *Ceiba pentandra* provide insights into the evolution of Malvaceae species and differences in their natural fiber development. *Plant Commun.* 2024 May 13;5(5):100832. doi: 10.1016/j.xplc.2024.100832. Epub 2024 Feb 5. PMID: 38321741; PMCID: PMC11121743.

Sun, P. et al. Subgenome-aware analyses reveal the genomic consequences of ancient allopolyploid hybridizations throughout the cotton family. *Proc. Natl. Acad. Sci. U. S. A.* 121, e2313921121 (2024).

Line915. Supplementary Fig. 4b is same to Fig. 1d.

Response: Fig. 1d pertains to Ad77271a, while Supplementary Fig. 4b pertains to Ad77271b. Both figures show the minor allele frequency coverage and, by extension, show the expected similarity between sibling genomes.

#####

First response to reviewers

Reviewer #1 (Remarks to the Author):

This paper describes a haploid chromosome-level reference genome for baobab, *Adansonia digitata*, as well as draft assemblies for a sibling tree, two trees from distinct locations in Africa, and a related species from Madagascar. The paper describes a paleopolyploid origin for baobab, shared with the Malvoideae ~30 million, and a more recent autotetraploidy event 3-4 million mya that coincides with a recent burst of TE insertions. Resequencing of 25 *A. digitata* trees from Africa revealed three subpopulations that suggest gene flow through most of West Africa but separated from East Africa.

I have read this paper with great interest, amongst other things because the paper describes new, high-quality genomic data of a truly iconic species. However, I have to say that I was a little disappointed after reading the paper. Please allow me to explain why.

Response: Thank you for your valuable feedback on our manuscript. Your thoughtful evaluation and suggestions are greatly appreciated, and we are grateful for your support throughout this process. We are fully dedicated to addressing all the concerns you raised and have made revisions to enhance the quality of the manuscript.

For instance, I found the entire section on whole genome duplication (WGD) quite superficial and to contain some statements that are difficult to understand. For instance, the authors state that “The Ks revealed a consistent timing of the separation of baobab-amborella and baobab-grape at 128 and 96 MYA respectively (Fig. 5b). Both cacao and cotton diverged from the ancestor of *Adansonia* around 30 MYA, which was the first piece of evidence that a baobab-specific WGD occurred around this time.” First, the approach to determine the timing of WGDs and speciation events seems only based on estimated Ks values, which is concerning since different species can have very different substitution rates. Also, how are these Ks values calculated? Which software is used to get those Ks values? Which substitution rates are used for the inference of divergence time? There is no information on this unless I have missed it. Furthermore, one must be cautious to compare Ks peak values to determine which event occurred first, a speciation or a WGD event. In my opinion, if there is some concern, dot plot analysis and in particular gene tree – species tree reconciliation methods should be applied to unequivocally (and still then ...) place WGDs in time.

Response: Generally, we start WGD analyses with the amborella and grape comparisons to determine if the general WGD/WGT history is consistent using a specific substitution rate; in this case we found that the a substitution rate of 6.56×10^{-9} (Gaut et al., 1996) recapitulated the timing of baobab-amborella and baobab-grape as has been described by many papers as noted in the text. Of course, these are estimates on the order of millions of years and should be taken

as directional just like dotplots and gene trees. All the genome comparisons were first visualized as dotplots, and syntenic depth was calculated. The syntenic depth plots as well as dotplots were added to the supplemental figures (**Supplementary Fig. 8 and Supplementary Fig. 9**). In addition, we have added time-calibrated phylogeny based on 1,296 single-copy orthologue genes to the revised manuscript (**Supplementary Fig. 10; Response Fig. 1**). The resulting timing slightly shifts the split of Ad77271a to 21 MYA from the previously estimated 17 MYA. However, the overall pattern remains consistent with the Ks analysis.

Response Fig. 1: Time-calibrated phylogenetic tree of *Theobroma cacao*, *Bombax ceiba*, *Gossypium hirsutum*, *Gossypium raimondii*, *Adansonia za* (Aza135), and *Adansonia digitata* (Ad77271a) based on 1,296 single-copy orthologous genes. Sky-blue horizontal bars indicate the 95% Highest Posterior Density (HPD) intervals of estimated divergence dates. Median node ages are shown beside the HPD bars, and timescale in million years ago (MYA) is provided at the bottom.

We have provided a more detailed explanation of the methodology used for calculating Ks values and inferences of divergence times in the revised manuscript. The supplemented “methods” section is as follows:

Page 21 lines 32~46:

Molecular dating and whole genome duplication (WGD): A total of 216 single-copy orthologues from *T. cacao*, *B. ceiba*, *G. hirsutum*, *G. raimondii*, *A. za* (Aza135), and *A. digitata* (Ad77271a), encompassing 1,296 single-copy orthologous genes, were analyzed. Protein sequences were aligned using MUSCLE (v.3.8.31) and converted to corresponding coding sequences (CDS) by substituting each amino acid with its triplet bases. The CDS for each single-copy gene family were concatenated to form a super gene for each species. The second nucleotide of each codon from the aligned sequences was extracted to construct a phylogenetic tree using BEAST (v2.7.6) under the HKY model with a Gamma category count of 4. The tree prior was set to the calibrated Yule model, with birth-rate parameters set to an Alpha (shape) parameter of 0.001 and a Beta (scale) parameter of 1000. Secondary calibration of molecular clocks was performed using priors from the <http://www.timetree.org/> database for the cotton split from bombacoid/malvoid (50 MYA)^{107,108}. For WGD, synonymous substitution rates (Ks values) were calculated using the MScan (Python version) with default parameters. A substitution rate of 6.56×10^{-9} was then utilized to estimate the time of duplication events¹⁰⁹

Reference:

Gaut, B. S., Morton, B. R., McCaig, B. C. & Clegg, M. T. Substitution rate comparisons between grasses and palms: synonymous rate differences at the nuclear gene *Adh* parallel rate differences at the plastid gene *rbcl*. *Proceedings of the National Academy of Sciences* 93, 10274–10279 (1996).

The authors state that “The self-alignments of genomes informed the timing of the most recent WGD in a species, which clarified that all of the baobab, bombax and cotton genomes shared the same WGD event at about 30 MYA, while cacao experienced its last WGD around 118 MYA that was consistent with it having only the WGT- γ (Fig. 5c).” What does the self-alignment of genomes mean? From the figure, I assume it refers to age distributions made from collections of paralogous genes (all genes, or only homeologs)? Finding similar Ks peak values for different organisms can indeed suggest a shared WGD, but it could also suggest independent WGDs in different branches at similar times in evolution.

Response: The “self-alignment of genomes” (Ad77271a vs. Ad77271a, etc.) refers to estimating the Ks based on paralog comparisons where paralogs result from past WGD events. We edited it to say “self-self alignments” with paralogs mentioned. We agree that they don’t have to share the same WGD, so we changed this wording. Based on the timing of the time calibrated tree, it implies independent WGDs. The text has been updated to make this clearer.

Second, the entire section could do with some more detailed explaining. “Both cacao and cotton diverged from the ancestor of *Adansonia* around 30 MYA, which was the first piece of evidence that a baobab-specific WGD occurred around this time”. How is this first evidence for a baobab-specific WGD? It seems problematic to infer a specific WGD event solely based on divergence time without further elucidation or reference to accompanying figures. Additionally, there’s a notable absence of a summarizing sentence indicating the presence of a WGD event shared with the Malvoideae but not with cacao. Instead of ambiguously stating “around ~30 Mya,” it would be more precise to assert that this WGD event occurred after the divergence with cacao but prior to the divergence within the ancestor of the Malvoideae.

Response: We have revised the text and asserted that the WGD event happened after the split from cacao, as shown in **Response Fig. 1**.

The authors state that “When we zoom into just the baobab complex, we saw that all genomes have a minor Ks peak around 30 MYA in addition to peaks at 4, 6, 11 and 17 MYA for Ad77271b, 1 AdKB, AdOHT and Aza135 respectively (Supplementary Fig. 7). We hypothesize that the autotetraploidy event might be seen as the distance between the Ad77271a and Ad77271b siblings since they most likely represent distinct and random haplotypes, which would place the autotetraploidy event in the 4 MYA range with a similar timing of the last detectable TEs burst (Fig. 2b).” I think this is not good enough and the authors should try harder to explain what these different peaks (obtained by Ks rates) refer to? For instance, it is not clear to me what e.g., the 17 my old peak could refer to then?

Response: We have expanded the explanation of Ks peak values to offer a clearer interpretation, particularly regarding peaks like the one at 17 MYA as a speciation event, being in the range of the Ad77271a-Aza135 split depicted by the time-calibrated tree approach (21

MYA). While we have noted the observed elevated K_s values, confirming the precise events (speciation/duplication) and their timing requires additional datasets, such as fossil records, which are beyond the scope of this paper. Nonetheless, we believe readers will appreciate the complexity of the baobab's evolutionary journey and recognize the need for more targeted experiments to validate the hypotheses raised in this study.

The authors state that “We hypothesize that the genes retained as four copies in the baobab genome may represent specific biology that was important to baobab.” There is a huge literature (none of it cited) about gene balance and gene dosage issues that explain the biased retention of genes of certain functional categories (transcription factors, developmental genes, genes encoding proteins acting in multiprotein complexes) that has not necessarily to do with ‘specific biology of importance to baobab’, especially not when WGDs are relatively recent, or very conserved (see further). All the classes of genes the authors mention are the genes typically found in excess in genomes following WGDs. This has been widely documented, not only for plants but appears to be a general phenomenon.

Related to that, the authors state that “In addition to the WGD 4 MYA, baobab also experienced a WGD 30 MYA that is shared across the Malvaceae; yet unlike other Malvaceae species studied to date, baobab retained almost all four copies resulting from the WGD. While most genomes fractionate gene copies back to a diploid state after WGD, it is thought that some genes may not be fractionated and retained to modify gene dosage in specific pathways important to the organism.” The authors do not seem to be aware that gene loss takes time, and is related to substitution rates, which are a function of generation times, metabolic rates, and population sizes. Therefore, simply assuming that the difference in gene loss or retention following WGDs between two species with significantly different life styles, is due to selection for genes of importance and can be linked to adaptation, is short-sighted, I think and could lead to wrong conclusions.

Response: The haploid baobab genome is unique amongst plant genomes sequenced to date in that it has retained almost four full chromosomes after the WGD 30 MYA; if the base chromosome number is 10 based on cacao, then the 42 chromosomes represent mostly a retention and some splitting/fusions (**Response Fig. 2**). Soybean is similar and has retained almost all four copies after the last WGD 15 MYA. While these processes take time, based on the current high-quality genomes most fractionate back to two copies in the time frames analyzed. This assessment is based on an analysis of 123 high quality plant genomes (**Zhao et al., 2021**) that were analyzed for WGD, and we further analyzed to understand how often this happens in reference to the circadian, light and flowering time genes (**Michael 2021**). We do cite what we think is the most relevant background on fractionation and dosage (**Cheng et al., 2018**) and now we have added one other (**Ruju et al., 2020**). However, *A. digitata* does not retain all four copies suggesting that some genes are lost while the overall organization and structure of the chromosomes are retained. We agree that the language that these are “baobab specific” may be too strong so we have removed this wording. These concepts are consistent with current thinking in the genomics field regarding polyploidy and WGD (**Van de peer et al., 2017**). We also point out cases in the text where genes have been retained and play functions specific to a species such as in soybean and CAM plants; we highlight this rare event in Fig. 5d where syntenic copies of clock/flowering/temperature genes are retained in four copies. We are aware that transcription factors, developmental genes, and genes encoding proteins acting in multiprotein complexes are often retained after WGD and this is why we have highlighted gene

classes that are distinct: chromatin, exocytosis, and flower timing/development; the latter category we further develop later.

Response Fig. 2: The *A. digitata* genome retained many almost full chromosomes (Chr) after whole genome duplication (WGD). The Ad77271a (blue, bottom) genome was aligned to the cacao (*Theobroma cacao*; red, top) genome at the protein level with MCscan (Python version with default parameters). The cacao genome was used since it clearly displays the retention of full Chr structure (length) with the Ad77271a genome (Fig. 5a main text), which also highlights the continuity and accuracy of the Ad77271a assembly. The syntenic regions were visualized using the `jcvi.graphics.karyotype` tool with the settings `--minspan=30`. The resulting anchor file was analyzed for the number of syntenic connections between cacao and Ad77271a Chrs, and the syntenic blocks greater than 10 or 300 genes were retained for plotting. Each cacao Chr was plotted separately to highlight the complete Ad77271a Chr, and colors were randomly assigned to syntenic blocks by Ad77271a Chr. Cacao Chr 1, 2, 3, 4, 6, and 9 had all four copies retained in the Ad77271a genome with different levels and areas of fractionation. Chr 1 and 4 were fused before the Ad77271a WGD resulting in the Chr 1 arms being split into 8 copies with connections to Chr4. A similar yet more complicated fusion/breaking occurred

between Chr 5, 7, 8, and 10, which is why the four Chr retention pattern is less obvious in these plots compared to the dotplot (Fig. 5a main text).

References:

Cheng, F. *et al.* Gene retention, fractionation and subgenome differences in polyploid plants. *Nat Plants* **4**, 258–268 (2018).

Raju, S. K. K. Gene Dosage Balance Immediately following Whole-Genome Duplication in Arabidopsis. *The Plant cell* vol. 32 1344–1345 (2020).

Zhao T, Zwaenepoel A, Xue JY, Kao SM, Li Z, Schranz ME, Van de Peer Y. Whole-genome microsynteny-based phylogeny of angiosperms. *Nat Commun.* 2021 Jun 9;12(1):3498. doi: 10.1038/s41467-021-23665-0. PMID: 34108452; PMCID: PMC8190143.

Michael TP. Core circadian clock and light signaling genes brought into genetic linkage across the green lineage. *Plant Physiol.* 2022 Sep 28;190(2):1037-1056. doi: 10.1093/plphys/kiac276. PMID: 35674369; PMCID: PMC9516744.

Van de Peer Y, Mizrahi E, Marchal K. The evolutionary significance of polyploidy. *Nat Rev Genet.* 2017 Jul;18(7):411-424. doi: 10.1038/nrg.2017.26. Epub 2017 May 15. PMID: 28502977.

The authors do discuss some gene families in more detail, but most of it is very speculative. As a matter of fact, the paper is riddled with statements such as ‘we hypothesise’, ‘we speculate’, ‘these results could point’, ‘may provide insight’, ‘provides clues’, etc., which, I have to say, become a bit frustrating after a while.

Response: We understand your concerns about the speculative nature of some of our statements; however, this is a genome manuscript where many things cannot be tested readily. We have revised the manuscript providing more concrete statements where possible. Additionally, we have clearly distinguished between well-supported findings and hypotheses to enhance the clarity and robustness of our discussion. We would like to note that for many of these “speculative” findings, we provide several lines of evidence supporting them. Our main biological finding concerning circadian, chromatin and flowering genes were supported in both our WGD retention analysis, as well as the variation analyses across the 25 resequenced trees (population Fst and local PCA). We try not to overstate our findings, but we hope readers will be able to make their own conclusions based on the several lines of evidence we present. We have created a FigShare with all the raw data so researchers can check the veracity of our data and analyses.

The authors also conclude that the whole genome duplication unveiled in *A. digitata*, about 4 mya, coincided with an unprecedented amplification of DNA transposable elements (TEs) compared to Long Terminal Repeat Retrotransposons (LTR-RT). Again, this is very speculative given the way the authors have computed divergence times and estimates of the timing of WGDs. Also, why is the WGD inferred at 4 MYA consistent with an autotetraploidy event. Could this not have been an allopolyploidy event?

Response: We have conducted additional analyses using a time-calibrated phylogenetic tree as you suggested, which confirmed that the timing of speciation and WGD events are congruent

(Response Fig. 1). Therefore, the event 4 MYA that separates the haplotypes (Ad77271a vs. Ad77271b) represents the polyploidy event. It is true that distinguishing between auto- and allopolyploidy can be difficult because the hybridization of closely related species can appear as an autotetraploid event. However, we do not detect any evidence for an allopolyploidy event, such as distinct centromere sequences (Response Fig. 3) like we have seen in homeologs of *Eragrostis tef* (VanBuren et al., 2020), or a high level of variation in the subgenomes (once again the comparison between Ad77271a vs. Ad77271b). Therefore, based on our genome assemblies and annotations, we can assert that the WGD in *A. digitata* is consistent with an autotetraploid rather than an allotetraploid event. We now provide a clearer rationale for our inferences, supported by additional data and analyses.

Response Fig. 3: The *A. digitata* (Ad77271a) centromere arrays are identical across all 42 chromosomes. The centromere base repeats in arrays greater than 10 kb were extracted and

aligned using mafft (--reorder --adjustdirection --maxiterate 1000 --retree 1 --globalpair) and visualized with FigTree. Only arrays from chromosomes 6, 7, 9, and 20 (full chromosomes with cacao) are highlighted with different colors to show that there is not a specific pattern by chromosome.

Reference:

VanBuren R, Man Wai C, Wang X, Pardo J, Yocca AE, Wang H, Chaluvadi SR, Han G, Bryant D, Edger PP, Messing J, Sorrells ME, Mockler TC, Bennetzen JL, Michael TP. Exceptional subgenome stability and functional divergence in the allotetraploid Ethiopian cereal teff. *Nat Commun.* 2020 Feb 14;11(1):884. doi: 10.1038/s41467-020-14724-z. PMID: 32060277; PMCID: PMC7021729.

Other comments:

Page 4 line 29:

“We long read Oxford Nanopore Technologies (ONT) sequenced...” sounds awkward, I think.

Response: Revised.

Page 4, line 36:

The flow cytometry result shows that the genome size is around 920Mb, while the Kmer analysis result suggests a genome size of 749 Mb. However, the final haploid genome size is estimated at around 675Mb, so how to explain the 70 Mb size difference between the genome survey result and the final assembly? A 1.9% heterozygosity level (for *A. digitata*) is high, could this be part of the explanation (if the assembly is not completely phased)?

Response: Genome sizes from either flow cytometry or k-mer frequency are both estimates and can vary based on several factors (Pflug et al., 2020). Flow cytometry can result in estimates that differ from the actual genome size due to the selection of cultivars, staining techniques or the controls used, while k-mer genome size estimates vary due to sequencing reads used (long vs short reads), quality of sequencing data, coverage of sequencing data and the k-mer size chosen. A 70 Mb difference between the assembled genome size and k-mer frequency estimate is in the margin of error; however, we updated Fig. 1 as suggested with the genome scope result and now the k-mer frequency estimate is 659 Mb (different estimation methods even provide slightly different results), which is directly in line with the assembled genome size. Also, the genome scope figure more clearly highlights the broad single peak that we see in some autopolyploids (discussed in the text); the broad peak also causes issues with accurately estimating genome size using k-mer frequency.

Some of the hardest regions of a genome to assemble are the centromere, rDNA and TEs (Michael and Jackson, 2013). In the first wave of plant genomes based on either Sanger or short read sequencing, the centromere, rDNA and TEs were often collapsed leading to an assembled genome size smaller than the actual. Now with long read sequencing we are now rarely finding that regions collapsed and in addition we are assembling full centromeres and TEs while rDNA arrays still are problematic (Naish et al., 2021). While the centromere arrays are

large in our assemblies, there is always a chance that we have collapsed the rDNA array, which we quantify in the text.

We state in the text that we have only assembled the haploid genome of *A. digitata*. Therefore, the haploid representation most likely contains a combination of non-phased variation from the four subgenomes. While we have tried to separate the sub genomes, we were unsuccessful (we suspected due to the low level of SVs and SNPs between subgenomes), so we sequenced a sibling to (Ad77271b) to ascertain the level of heterozygosity. While the overall heterozygosity is estimated at 1.9%, which reflects the underlying four subgenomes (haplotypes) of the tetraploid, the comparison of the sibling assemblies Ad77271a and Ad77271b was 0.65%.

References:

Pflug JM, Holmes VR, Burrus C, Johnston JS, Maddison DR. Measuring Genome Sizes Using Read-Depth, k-mers, and Flow Cytometry: Methodological Comparisons in Beetles (Coleoptera). *G3 (Bethesda)*. 2020 Sep 2;10(9):3047-3060. doi: 10.1534/g3.120.401028. PMID: 32601059; PMCID: PMC7466995.

Michael and Jackson. The First 50 Plant Genomes. *The Plant Genome*. 2013 July 01; <https://doi.org/10.3835/plantgenome2013.03.0001in>.

Naish M, Alonge M, Wlodzimierz P, Tock AJ, Abramson BW, Schmücker A, Mandáková T, Jamge B, Lambing C, Kuo P, Yelina N, Hartwick N, Colt K, Smith LM, Ton J, Kakutani T, Martienssen RA, Schneeberger K, Lysak MA, Berger F, Bousios A, Michael TP, Schatz MC, Henderson IR. The genetic and epigenetic landscape of the Arabidopsis centromeres. *Science*. 2021 Nov 12;374(6569):eabi7489. doi: 10.1126/science.abi7489. Epub 2021 Nov 12. PMID: 34762468.

Page 5, lines 23~32:

This large translocation could hold significance for the baobab tree. Nevertheless, the absence of *RPA1B* does not correlate with the translocation of these genes. We cannot attribute the function of losing this gene to the function of gene translocation. They do not have a direct relationship.

Response: We don't say that *RPA1B* is lost. We say that *RPA1B* is part of the 1 Mb translocation along with other genes that have potentially relevant functions and gene ontology (GO) enrichment. We highlight *RPA1B* because it also overlaps with UV-B signaling that is important in baobab due to its unique copies of these genes in the genome (highlighted later in the text).

Page5 line 34:

The Kmer analysis result shown in Fig 1a looks strange. With a heterozygosity level of 1.9%, the result should show a peak indicating Kmers from heterozygous loci and a main peak. Why not just provide the result from GenomeScope?

Response: In the revised version, we have substituted the image with genome scope output as suggested (**Response Fig. 4**). As noted above, genome scope estimates the genome size to be similar in size to the assembled genome (659 vs. 674 Mb), and the heterozygosity is 1.63%, which is similar to our estimate based on read mapping (1.9%).

Response Fig. 4: Characteristics of baobab (*Adansonia digitata*) genome. a GenomScope estimation of *A. digitata* genome size using 19-mer sequence counts and ploidy set to 4, the K-mer frequency depicts a unimodal pattern suggesting a diploid homozygous genome with a size of 659 Mb.

Page5 line 35:

Is Fig. s 1a,b the same as Fig. 1a;b? This causes confusion (throughout the manuscript).

Response: Revised.

Page 6 Table 2:

Some information in Table 2 is not clear. Does the N50 shown in the table refers to the Contig N50 or the Scaffold N50? Does the BUSCO assessment refers to the genome assembly or to (the completeness of) the annotation? The table would be better divided into three subsections, I think, namely genome assembly, repeat content, and annotation.

Response: We have addressed your concerns in the revised version. The table is now divided into three subsections, as suggested.

Page 7 line 25-28:

This information is already shown in Table 2, there is no need to reiterate it here.

Response: We have removed the redundant information.

Page 8 Fig 2:

Is it possible to use the same color palette for the same TE element?

Response: We have implemented as suggested (**Response Fig. 5**).

Response Fig. 5: Insights into transposable elements (TEs) in baobab. **a** Barplot comparison of TEs classification and sizes in five baobab genomes: Ad77271a, Ad77271b, Aza135, AdKB, and AdOHT. Aza135 is highlighted in the center to emphasize TE divergence. Except for Aza135 (*A. za*), the other four genomes belong to *A. digitata*. Distinct colors in the legend denote TE classes. The TE proportion to genome size (Mb) and percentage are displayed on the left and right y-axes, respectively. **b** Density plots of intact TEs, displaying the distribution of different types of TEs. Mutator transposons showed a burst around 11 million years ago. **c** Accumulation of DNA mutators in the centromeric regions of the Ad77271a chromosome 3. Four mutator Terminal Inverted Repeat (TIR) transposons (TE_00000631: Mutator631, TE_00000927: Mutator927, TE_00000845: Mutator845, and TE_00000967: Mutator967) with a potential role in centromere creation or repositioning are displayed. **d** High solo to intact LTR ratio in Ad77271a chromosomes shows an aggressive purging mechanism following WGD, contributing to a smaller genome size (659 Mb).

Page 8 line 19:
Fig.s 3 -> Fig. 3?

Response: We have corrected the figure reference to "Fig. 3" in the revised version.

Page 9 line 6:
Fig.s 3 -> Fig. 3?

Response: Revised.

Page 9 line 7:
The methylation part is very descriptive. It would be interesting to investigate whether the methylation links to any trait or adaptation of baobab.

Response: Investigating the potential links between methylation and specific traits or adaptations of baobab is indeed an intriguing avenue for future research. We have expanded the analysis to include global methylation patterns in relation to genes and transposable elements. Our observations indicate the presence of methylation across the entire genome, with subtle elevations on DNA mutators or centromeric regions, such as on chromosome 11 (**Response Fig. 6**).

Response Fig. 6: Methylation patterns across *A. digitata* chromosomes. Global distribution of genome features in *A. digitata*. Centromere arrays are plotted as darkgreen bars at $y=0.8$. Genes are represented in blue, Mutator transposons in green, CACTA transposons in red, Gypsy retrotransposons in purple, Copia retrotransposons in orange, and CpG methylation in black. Transposable elements, such as Mutator transposons, tend to be located opposite to genes and in the centromeric regions. Methylation is present throughout the chromosome. The window positions are divided by the chromosome total length to create a relative scale between 0 and 1.

Page 9 line 26:

It would be good to add another *Adansonia* (Aza135) in the gene family analysis to check whether it is also having these baobab specific orthogroups.

Response: We have added Aza135, AdOHT and AdKB to the gene family analysis (**Response Fig. 7**) and discussed the gene ontology (GO) terms of baobab-specific orthogroups (**Response Fig. 8**).

Response Fig. 7: An UpSet plot showing orthogroups across 20 plant species: *A. digitata* (Ad77271a, Ad77271b, AdKB, AdOHT), *A. za* (Aza135), *A. calamus*, *A. gramineus*, *A. thaliana*, *A. trichopoda*, *B. ceiba*, *D. zibethinus*, *G. biloba*, *G. hirsutum*, *G. raimondii*, *K. fedtschenkoi*, *P. trichocarpa*, *Q. rubra*, *S. grande*, *T. cacao* and *V. vinifera*. The number of species-specific orthogroups and shared orthogroups is shown on the main bar y-axis; red bars correspond to the baobabs. The x-axis bars correspond to the number of orthogroups containing species.

Response Fig. 8: Subset of the 125 significant GO terms in 817 specific orthogroups unique to baobab (*Adansonia digitata*, including accessions Ad77271a, Ad77271b, AdKB, and AdOHT, and *Adansonia za* Aza135) compared to a diverse set of fifteen plant species: *Acorus calamus*, *Acorus gramineus*, *Arabidopsis thaliana*, *Amborella trichopoda*, *Bombax ceiba*, *Durio zibethinus*, *Ginkgo biloba*, *Gossypium hirsutum*, *Gossypium raimondii*, *Kalanchoe fedtschenkoi*, *Populus trichocarpa*, *Quercus rubra*, *Symphonia grande*, *Theobroma cacao*, and *Vitis vinifera*. The x-axis represents the gene ratio (gene count / 1178 genes). The y-axis lists the GO term names. Dot size indicates the study count, while dot color reflects the p-values, adjusted using the FDR BH method implemented in GOATOOLS v1.3.11. The color gradient corresponds to the range of p-values.

Cacao is not the sister group of baobab based on Fig. 4d. Is there something special in the gene families shared by baobab and cacao as the authors only list the result between these two species?

Response: We selected cacao as the comparison species due to its status as an outgroup of Malvaceae species with extensive evolutionary information available.

Page 12 line 9:

Fig.s 5 -> Fig. 5; Fig 5a is a syntenic dot plot, not syntenic depth plot.

Response: Revised.

In Supplementary Fig. 9, please highlight some syntenic regions.

Response: Implemented as suggested, thank you (**Response Fig. 9**).

Response Fig. 9: Comparative analysis of autotetraploid *A. digitata* and diploid *Theobroma cacao* genomes. Inner to outer tracks depict: a Syntenic genes, b GC content, c Gene density, and d Chromosome information. Prefixes 'Ad' and 'Tc' denote baobab and cacao respectively. The circos plot illustrates 42 pseudomolecules for baobab and 10 for cacao, with a window size of 100 kb. The red asterisk highlights the metacentric and acrocentric centromeres in baobab.

Page 12 line 11:

Fig.s 5 -> Fig. 5

Response: Revised.

Page 12 line 13-14:

Should it be “amborella and grape...”? The ratio also does not seem correct, could the author please also provide the syntenic depth plot for those comparisons.

Response: We have paraphrased the sentence for clarity and provided syntenic depth plots for the comparisons (**Response Fig. 10**).

Response Fig. 10: Syntenic depth ratios between Ad77271a and a Ad77271b b AdKB c AdOHT d Aza135 e cotton f bombax g cacao h grape and i amborella.

Page 12 line 16:
Fig.s 4 -> Fig.4

Response: Revised.

Page 14 line 1:
“as well 1 as 1 and 3 were less related”?

Response: We have revised the sentence to clarify that population 1 is as distantly related to population 2 as population 3 is to population 2.

Page 19 line 10:

Could the authors kindly provide the parameters which they provided to Funannotate? What evidence are used for gene prediction?

Response: The revised manuscript is as follows:

Repeats and gene prediction: EDTA (v1.9.6)⁹⁸ was used to construct a repeat library and softmask complex repeats. Tandem Repeats Finder (v4.09)⁹⁹ was employed to identify centromere and telomere sequences, as well as softmask simple repeats. The ONT cDNA library was mapped against the reference using Minimap2 with splice presets and then assembled using Stringtie (v2.2.1) with the long reads processing flag. Genes in baobab were predicted via the Funannotate (v1.8.2) pipeline with the Stringtie transcript models included as

evidence in addition to the uniprot protein set. Predicted proteins were characterized using EggNOG-mapper v2.0.1¹⁰⁰.

References:

Ou, S. et al. Author Correction: Benchmarking transposable element annotation methods for creation of a streamlined, comprehensive pipeline. *Genome Biol.* 23, 76 (2022).

Benson, G. Tandem repeats finder: a program to analyze DNA sequences. *Nucleic Acids Res.* 27, 573–580 (1999).

Cantalapiedra, C. P., Hernández-Plaza, A., Letunic, I., Bork, P. & Huerta-Cepas, J. eggNOG-mapper v2: Functional Annotation, Orthology Assignments, and Domain Prediction at the Metagenomic Scale. *Mol. Biol. Evol.* 38, 5825–5829 (2021).

Page 19 line 14:

Substitution rate varied a lot between species, Malvaceae and Salicaceae are not that closely related, using the substitution rate from *Populus* is probably not accurate enough for the calculation of the LTR insertion time.

Response: While we recognize that baobabs have distinct evolutionary histories, which may limit the direct applicability of substitution rates from other taxa. As noted above, we leveraged this substitution rate because it recapitulated the published timing of known events.

In conclusion, although the paper discusses the genomes of an iconic species, I found the paper to be very descriptive and to be honest, seems to reveal very little about the evolution and/or adaptation of baobab, as nevertheless suggested by the title. The authors state that ‘this research not only unravels baobab genomic evolution mysteries but also provides a crucial sequence for expediting gene discovery, enhancing breeding efforts, and aiding baobab species conservation.’ I’m not sure what ‘genomic evolution mysteries’ the authors refer to or solved? The paper lists TEs, describes a WGD, there is some information on methylation, and some discussion on the genetic diversity in baobab, none of which provides, in my opinion, exciting novel discoveries that shed light on the evolution, diversification, or adaptation and specific lifestyle of baobab. Again, what is the significance of statements such as ‘these results indicate there are geographical or environmental factors that have limited gene flow between these populations.’?!

Response: We have reevaluated our conclusions to ensure that they effectively highlight the novel insights gained from our research. Additionally, we have refined our language to better articulate the evolutionary implications of our findings, emphasizing their relevance to baobab diversification and adaptation. We appreciate your input and have taken steps to strengthen the manuscript accordingly.

Reviewer #2 (Remarks to the Author):

The manuscript presents a comprehensive genomic study of the African baobab tree, *Adansonia digitata*, a long-lived species with significant ecological and economic importance. The authors constructed a chromosome-level reference genome for *A. digitata* using long-read Oxford Nanopore sequencing and HiC data and draft assemblies for related species,

uncovering unique genomic features. Unlike most plant genomes dominated by LTR retrotransposons, the baobab genome has a high proportion of DNA transposons. The authors report that a whole-genome duplication event occurred 30 million years ago, shared with other members of the Malvaceae family. In addition, an autotetraploid event 4 million years ago that correlates with a burst of DNA transposon insertions was identified. Moreover, the authors report that baobab retains multiple copies of circadian, light, and flowering-related genes, which may contribute to its longevity and unique pollination strategies. They resequenced 25 *A. digitata* trees across Africa, finding geographic variation that divided the species into three main subpopulations.

Response: We appreciate your professional summary of our work.

The baobab genome is undoubtedly fascinating. The genomic resources and insights offered by this study could form the basis for conservation efforts focused on preserving the diversity of the baobab. This research significantly enhances our understanding of the baobab genome and its evolutionary history. Perhaps clearer conclusions on the practical implications for breeding and conservation strategies could be provided.

Response: We have revised the manuscript to include clearer significance of the study

Page 3 line 42 to page 4 line 17:

Here, we report genome size estimates for all eight recognized baobab species using a K-mer-based method from short-read genomic sequences. This method can provide independent estimates to those obtained previously using Feulgen staining and flow cytometry¹. Our primary objectives are to create a reference genome, evaluate genetic diversity, investigate key genomic features, and study evolutionary history. We generated a haploid chromosome-scale assembly of *A. digitata* (Ad77271a; originally from Tanzania) as well as long read draft assemblies of an Ad77271a sibling (Ad77271b), and offspring of the following trees; the Kord Bao Sudan (AdKB), the Okahao Heritage Tree from Namibia (AdOHT), and an additional species, *A. za* (Aza135) from Madagascar. Additionally, we resequenced 25 additional *A. digitata* trees representing different regions of Africa with short reads to assess genetic diversity. The findings from this work revealed: (a) a high proportion of DNA transposons compared to LTR retrotransposons, suggesting an unusual genomic structure; (b) specific set of DNA mutator transposons potentially contributing to new centromere formation; (c) retention of nearly complete chromosomes post-whole genome duplication (WGD), indicating a unique evolutionary path in genome maintenance; (d) a recent WGD event approximately 30 million years ago leading to gene expansion in flower development, chromatin regulation, and exocytosis; (e) expansion of UV RESISTANCE LOCUS 8 (UVR8) genes, which suggests a unique mechanism for genome protection in long-living trees; and (f) discovery of genetically distinct Namibian baobab populations, enhancing our understanding of baobab diversity and suggesting the need for targeted conservation initiatives. These insights will facilitate breeding, conservation and domestication efforts.

In addition, here are my detailed comments:

1. Baobab is autotetraploid. It went through a whole genome duplication and has $2n=4x=168$

chromosomes. The reported reference genome has 42. So not only the homologs are collapsed as typical for reference genome assemblies but also the homeologs. While I recognize the challenges in assembling sequences to individual chromosomes, I am concerned about the potential impact on the study's conclusions. 4 million years is a lot of time for mutations to happen, hence the sequences wouldn't necessarily collapse but lead to assembly artefacts and problems especially if the subgenomes don't recombine. Do they? If they don't, they must have evolved and changed in 4 million years. Collapsing them can lead to a lot of wrong conclusions. Was this considered when running the expansion/contraction analyses of gene clusters and copy number variation analyses? The expansion in copy number of specific genes mentioned in the manuscript can also be due to this. Also the translocations. These might be present in one subgenome and not in the other. Have you tried to separate the subgenomes with your long reads?

Response: We agree; we were hoping to resolve all four haplotypes of the baobab genome. As we note in the text, additional methods (that may or may not be possible in baobab) will be required to assemble the fully phased, haplotype resolved genome. For autotetraploid genomes with an even higher level of heterozygosity (both SV and SNPs/INDEL) such as the potato genome, a pollen approach was required. We did try to resolve the subgenomes with the long reads but were unsuccessful.

The k-mer frequency plot suggests that the sub genomes are very similar since we observe a single peak. As noted in the text, and discussed below, we also have observed this k-mer frequency profile in *Lemna* that has a low level of SV and SNPs/INDEL variation. In figure 1a we include the k-mer frequency plot that has only one peak, which is consistent with a homozygous diploid genome. This is a similar plot you would see for an inbreeding species like *Arabidopsis* which has very low heterozygosity and is diploid. Since our allele frequency analysis confirmed *A. digitata* is tetraploid, as was predicted by other methods, the k-mer frequency suggests that the subgenomes are very similar at the nucleotide level. As noted to reviewer #1, the heterozygosity between the subgenomes is on the order of 0.65%.

We conducted a coverage analysis (in response to DNA TE vs LTR query) and we observed consistent coverage across the genome and repeats suggesting there is not large-scale collapsing. In addition, across the four *A. digitata* genomes we find a high level of synteny, especially with the diploid *A. za* and cacao genomes, which suggests we do not have a high level of collapsing. All four *A. digitata* genomes displayed the same translocation event with *A. za* consistent with this being real and not an artifact of collapsing (it is unlikely all four genomes would collapse in the same way). The four *A. digitata* genomes share similar gene expansions/contractions consistent with these gene families being resolved correctly. All the measures that we have tried (coverage, BUSCO, k-mer, synteny, etc) suggest that the haploid representation is a high quality, non-collapsed version of the genome.

2. Are you sure about "buhibab"? It happens that I speak Arabic and I don't recognize the word at all. In standard Arabic, "fruit with many seeds" would typically be expressed as "فاكهة ذات بذور كثيرة" - fākihah dhāt bidhūr kathīrah or "فاكهة ذات حبوب كثيرة" - fākihah dhāt ḥubūb kathīrah, where "بذور" - bidhūr and "حبوب" - ḥubūb both mean seeds.

You are citing a citation, that is citing

this: <https://www.sciencedirect.com/science/article/pii/S0254629911001189#bb0160> and

this: https://agritrop.cirad.fr/531802/1/document_531802.pdf and the first is citing the second.

So it all comes down to Diop et al. 2005 which mentions that the word "baobab" originates from the Arabic term "bu hibab," which is claimed to mean "fruit with many seeds." However, the text

notes that this origin is controversial, and various explorers and researchers have used different terms to describe the baobab tree and its fruit over time. The concept that "bu hibab" might mean "fruit with many seeds" could either be an adaptation or interpretation of the term in historical texts.

Hibab sounds close to ḥubūb/ huboob but anyway, obviously this is not a solid fact. So please, if you don't mind and unless you can be sure about the origin, please rephrase your sentence "The word "baobab" comes from the Arabic name 'buhibab,' which means a fruit with many seeds1."

Response: We appreciate your perspective on this matter. Given the uncertainty surrounding the origin of the word "baobab," we have removed the specific Arabic term "buhibab" to avoid potential inaccuracies.

3. There is a translocation between the *A. digitata* species and Aza135 on Chr23 that moved ~120 genes including genes related to longevity closer to the telomere. The hypothesis is that telomere-associated proteins may be more active or better maintained near telomeric regions, I suppose? E.g. moving DNA repair genes closer to telomeres could facilitate more efficient repair of telomeric DNA and the maintenance of chromosome stability, right? Maybe you can make the hypothesis more clear. But then again this only affects one chromosome. And also is there is a difference in terms of longevity between Aza135 and the other species?

Response: In the section on the translocation, we are just highlighting that the 120 genes in this region share interesting biology (and GO enrichment) related to our finding of unique UV-B genes; see the comment to reviewer#1. It is an interesting hypothesis that moving genes closer to the telomere may play a role in maintenance or chromosome stability. Another hypothesis is that this translocation creates a species barrier between *A. digitata* and *A. za*, but this is probably too speculative to include as well.

4. How can the variation in repeat content be explained? Does it correlate with the phylogeny?

Response: The repeat variation is consistent across the *A. digitata* genomes (44-45%) even though we have seen that the repeat variation can vary substantially across accessions of the same species. The fact that *A. za* has an overall lower repeat content (35%) could reflect a different evolutionary trajectory of repeats (it only takes one LTR to copy and paste to cause genome expansion) or it could be the specific *A. za* tree we sequenced. We have too few genomes and species to draw general conclusions about repeat content and phylogeny. However, the consistency across the *A. digitata* genomes suggests repeat content could be conserved across different trees.

5. By inter-homeolog recombination you mean that homeologs chromosomes recombine, is that correct? I can't find a description of that in the Lemna minor publication. Maybe I'm overlooking.

Response: We have edited this section for clarity. We are trying to make the case that in polyploid genomes with low levels of structural variation (SV) and SNPs/INDELs, a single k-mer peak is sometimes observed.

6. It is very interesting that there are more DNA than LTR transposons. However, don't you think

it is possible that a lot of LTR elements collapsed in the ONT assembly? You can see that looking at depth of coverage and mapping rates when mapping the reads back to the genome.

Response: All five assemblies have more DNA TEs, and we don't see any bias in coverage suggesting the LTRs are not collapsed. Generally, with long read based assemblies we don't see the LTRs collapsing; generally the read length is greater than TEs allowing their proper resolution. However, we conducted a coverage analysis across the genome, as well as the three classes of LTR-RTs, and found that the coverage was consistent across genomes and LTRs, providing an additional data point that these regions are fully assembled. We tested this another way using the LTR Assembly Index (LAI) for the whole genome (calculated using https://github.com/oushujun/LTR_retriever) and found mean LAI was 9 with a standard deviation of 4.87 (sliding window size of 3 Mb with 300 Kb steps). This LAI score categorizes our assembly within the range of a "reference" assembly (Mokhtar et al. 2023).

Reference:

Mokhtar MM, Abd-Elhalim HM, El Allali A. A large-scale assessment of the quality of plant genome assemblies using the LTR assembly index. *AoB Plants*. 2023 Apr 4;15(3):plad015. doi: 10.1093/aobpla/plad015. PMID: 37197714; PMCID: PMC10184434.

7. Why do AdOHT and Aza135 have such a low BUSCO score?

Response: BUSCO scores of 93 and 94% are still quite respectable and appropriate for what we use them for in this manuscript. These BUSCO scores are lower than Ad77271a, Ad77271b and AdKB, which are almost complete (C: 98%), but the AdOHT and Aza135 BUSCO scores are still quite good.

8. Fig. 4c: did you find this enriched in the gene family expansions of baobab? Can you please provide the tables of enriched terms?

Response: Yes, we found genes shown in Fig. 4c enriched and we have included a supplementary table (13 and 14) in revised version of the manuscript

9. Did you only use two of the baobab genomes for Orthofinder? Why?

Response: In the revised manuscript, we show all five genomes for which we have long reads (**Response Fig. 5**). However, for the identification of baobab single-copy genes and the additional time-calibrated phylogenetic tree analysis (**Response Fig. 1**), we focused specifically on the reference genome (Ad77271a and the sibling genome). This decision was made to ensure consistency and accuracy in our comparative genomic analyses, as the reference genome provides the most comprehensive and well-annotated dataset for these specific analyses.

Response Fig. 7: An UpSet plot showing orthogroups across 20 plant species: *A. digitata* (Ad77271a, Ad77271b, AdKB, AdOHT), *A. za* (Aza135), *A. calamus*, *A. gramineus*, *A. thaliana*, *A. trichopoda*, *B. ceiba*, *D. zibethinus*, *G. biloba*, *G. hirsutum*, *G. raimondii*, *K. fedtschenkoii*, *P. trichocarpa*, *Q. rubra*, *S. grande*, *T. cacao*, *V. vinifera*. The number of species-specific orthogroups and shared orthogroups is shown on the main bar y-axis; red bars correspond to the baobabs. The x-axis bars correspond to the number of orthogroups containing species.

10. The cells in your suppl. table 7, column E are being formatted to dates from row 96 onwards.

Response: Revised, thanks.

11. Please add the URL of the Baobab database <https://resources.michael.salk.edu/baobab/index.html> to the manuscript. The link in the text is not working.

Response: We have updated it and ensured it works properly (Response Fig. 11).

Baobab Genome Database

Response Fig. 11: Baobab genome portal: Access the baobab genome portal at <https://resources.michael.salk.edu/baobab/index.html> for comprehensive information and resources on baobab genomes.

12. Page 13, line 28: (population 3; top left; green circles). You need to refer to the figure. Fig 6a, I assume.

Response: Yes, you are correct. The sentence now reads: "population 3; top left; green circles (Fig. 6a)." We have made the necessary corrections.

13. "Genes in baobab were predicted via the Funannotate (v1.8.2) pipeline with modifications" What modifications? And were all the baobab genomes annotated like this?

Response: We have added the specific annotation parameters used in the Funannotate pipeline in the revised version of the manuscript; see the comment to reviewer#1. Yes, all the baobab genomes were annotated using the same pipeline to ensure consistency across our analyses.

14. Would it be possible to name the chromosomes chr1 instead of just "1"?

Response: We have updated the manuscript to name the chromosomes as "chr1" instead of just "1."

Reviewer #3 (Remarks to the Author):

I was happy to read this manuscript presenting very interesting research and very much needed if the domestication of African baobab should be started. The introduction summarizes the state of our knowledge of African baobab but in the end, I miss much more clearly explained problem statement and also the general and specific objectives should be expressed in better way. E.g. why we need to know the ploidy of *A. digitata*, and how it can help us in further domestication of this species.

Response: Thank you for your suggestions and encouraging feedback. We have revised the introduction to more clearly articulate the problem statement and explicitly state the general and specific objectives of the study, including the importance of understanding the ploidy of *A. digitata*, and its implications for domestication efforts.

The supplemented words in the "Introduction" section are as follows:

Page 3 lines 1~17

Baobabs are some of the oldest and largest non-clonal living organisms on Earth with trees that can live over 2,400 years with canopy sizes of greater than 500 m³ and trunks reaching diameters of up to 10.8 meters (35 feet)⁸. However, baobabs are unlike most large and long-lived trees; they are succulents characterized by parenchyma-rich tissues that efficiently store

water and therefore, do not form “growth rings” or true wood⁹. Achieving maturity in the wild presents a considerable challenge for baobab trees since seedlings face predation from caterpillars, goats, and cattle¹⁰. Despite having bisexual flowers, baobabs are mostly self-incompatible, depending on external pollinators for successful fertilization^{11,12}. In natural populations, *A. digitata* is primarily pollinated by bats^{13,14}, with occasional visits by bushbabies¹⁵, and hawkmoths in southern Africa¹². Since *A. digitata* is an obligate outcrosser, the populations harbor a high level of diversity, which is observed as heterozygosity in the genome (Fig. 1a; Table 1)^{12,16}. Understanding the ploidy of *A. digitata* is essential for breeding and genetic improvement programs, as polyploidy affects traits like growth rate and environmental adaptability. This understanding can help overcome the species' slow maturation period of 8 to 23 years and aid in its domestication. Additionally, measuring genome size across different baobab species provides valuable evolutionary insights and aids in designing genomic sequencing projects^{17,18,19}.

Page 3 line 42 to Page 4 line 17: (see the comment to reviewer#2)

The results from this research should be of the interest of wider scientific community, however, sometimes I felt lost such a high number presented results in Table and six figures. Would it be possible to consolidate result explain them specially in much understandable manner for wider audience even without the deep experience in this specific topic. I just feel overwhelmed with such a high number of highly specific data results. However the implication of the results are very well enlighten in the discussion part

We have revised the results to make them clearer and more accessible to a wider audience, while still including detailed information for experts. Some of the novel discoveries are illustrated in graphical format (**Response Fig. 5**).

Response Fig. 5: Insights into transposable elements (TEs) in baobab. **a** Barplot comparison of TEs classification and sizes in five baobab genomes: Ad77271a, Ad77271b, Aza135, AdKB, and AdOHT. Aza135 is highlighted in the center to emphasize TE divergence. Except for Aza135 (*A. za*), the other four genomes belong to *A. digitata*. Distinct colors in the legend denote TE classes. The TE proportion to genome size (Mb) and percentage are displayed on the left and right y-axes, respectively. **b** Density plots of intact TEs, displaying the distribution of different types of TEs. Mutator transposons showed a burst around 11 million years ago. **c** Accumulation of DNA mutators in the centromeric regions of the Ad77271a chromosome 3. Four mutator Terminal Inverted Repeat (TIR) transposons (TE_00000631: Mutator631, TE_00000927: Mutator927, TE_00000845: Mutator845, and TE_00000967: Mutator967) with a potential role in centromere creation or repositioning are displayed. **d** High solo to intact LTR ratio in Ad77271a chromosomes shows an aggressive purging mechanism following WGD, contributing to a smaller genome size (659 Mb).

Reference:

Klein, S. J. & O'Neill, R. J. Transposable elements: genome innovation, chromosome diversity, and centromere conflict. *Chromosome Res.* **26**, 5–23 (2018).

#####

#####